# IBNorm: Information-Bottleneck Inspired Normalization for Representation Learning

## Abstract

Normalization is fundamental to deep learning, but existing approaches such as BatchNorm, LayerNorm, and RMSNorm are variance-centric by enforcing zero mean and unit variance, stabilizing training without controlling how representations capture task-relevant information. We propose IB-Inspired Normalization (IBNorm), a simple yet powerful family of methods grounded in the Information Bottleneck principle. IBNorm introduces bounded compression operations that encourage embeddings to preserve predictive information while suppressing nuisance variability, yielding more informative representations while retaining the stability and compatibility of standard normalization. Theoretically, we prove that IBNorm achieves a higher IB value and tighter generalization bounds than variance-centric methods. Empirically, IBNorm outperforms BatchNorm, Layer-Norm, and RMSNorm across large-scale language models (LLaMA, GPT-2) and vision models (ResNet, ViT), with mutual information analysis confirming superior information bottleneck behavior. Code will be released publicly.

## 1 Introduction

Normalization has become a cornerstone of deep learning, credited for stabilizing and accelerating training across domains. Techniques such as Batch Normalization (BN) (Ioffe & Szegedy, 2015), Layer Normalization (LN) (Ba et al., 2016), and RMSNorm (Zhang & Sennrich, 2019) are now standard in architectures spanning vision and language (Vaswani et al., 2017; Dosovitskiy et al., 2020; He et al., 2016a), underscoring normalization as a key driver of training stability and acceleration.

Despite this success, existing normalization methods share a fundamental limitation: they are inherently *variance-centric*. By enforcing zero mean and unit variance, followed by rescaling and shifting, they improve the conditioning of the optimization problem (Huang et al., 2023; 2018a; He et al., 2016b; Desjardins et al., 2015). However, this process operates purely on first- and second-order statistics, without offering any guidance on (intermediate) representation learning. Indeed, two representations may share identical mean and variance, but can encode drastically different amounts of task-relevant information. This raises a key question: can normalization be re-designed not only to stabilize training but also to shape representations toward sufficiency and generalization?

Early attempts have explored this direction. For instance, Eftekhari & Papyan (2025) encouraged Gaussian-like features via power transforms and additive noise, arguing that Gaussianity aids representation and noise acts as implicit regularization. However, such approaches lack a principled information-theoretic foundation and fail to explicitly balance preserving predictive information against suppressing nuisance factors, often yielding suboptimal embeddings for downstream tasks.

**Contributions.** In this work, we move beyond variance-centric normalization and introduce **IBNorm**, a simple yet powerful normalization method grounded by the Information Bottleneck (IB) principle (Shamir et al., 2010; Tishby & Zaslavsky, 2015; Shwartz-Ziv & Tishby, 2017). Moreover, we also theoretically justify the superiority of IBNorm and its better generalization performance.

First, we introduce an IB perspective to normalization, and propose IBNorm for generalization improvement. The IB principle seeks representations that preserve task-relevant information while discarding nuisances, thereby connecting normalization to generalization. Within this framework, we decompose existing normalization methods into three steps: (i) grouping features (e.g., across batch, channel, or dimension), (ii) standardization, and (iii) recovery via re-scaling and shifting. Our

analysis reveals that the normalization operation itself is the key bottleneck governing information flow. Building on this insight, IBNorm (IBNorm-S, IBNorm-L, IBNorm-T) introduces compression operators that selectively compress activations toward their means, providing explicit control over sufficiency–redundancy tradeoffs while remaining drop-in compatible with existing architectures.

Second, we provide theoretical guarantees. Under the IB framework, we prove that IBNorm achieves a strictly larger IB value than variance-centric methods like LN and BN, thereby better balancing predictive sufficiency with nuisance suppression. Moreover, we establish a provably tighter generalization bound, explaining why IBNorm outperforms standard normalization in practice.

Extensive experiments show the effectiveness of IBNorm across architectures and modalities. In language modeling, integrating IBNorm into LLaMA (Touvron et al., 2023b) and GPT-2 (Radford et al., 2019) outperforms LN, RMSNorm, and NormalNorm on LLM Leaderboard I/II by up to 8.75%. In vision, applying IBNorm to ResNet (He et al., 2016a) and ViT (Dosovitskiy et al., 2020) yields substantial gains, e.g., 3.98% improvement on ImageNet (Deng et al., 2009) (ResNet-50), and up to 8.17% improvement on ImageNet (ViT).

## 2 RELATED WORK

### 2.1 NORMALIZATION

Normalization plays a central role in modern deep learning, enabling faster convergence and improved generalization across natural language processing (NLP) (Devlin et al., 2019; Vaswani et al., 2017; Touvron et al., 2023a;b; Dubey et al., 2024; Bai et al., 2023; Liu et al., 2024) and computer vision (CV) (He et al., 2016a; Dosovitskiy et al., 2020; Liu et al., 2022).

Batch Normalization (BN) (Ioffe & Szegedy, 2015) normalizes activations across mini-batches to stabilize training, with numerous extensions such as Mean-only BN (Salimans & Kingma, 2016), $L^p$-Norm BN (Liao et al., 2016), Conditional BN (De Vries et al., 2017), and Whitening BN (Siarohin et al., 2018; Huang et al., 2018b; 2019). Layer Normalization (LN) (Ba et al., 2016) was proposed for sequential and small-batch settings and has inspired variants including Dynamic LN (Kim et al., 2017), RMSNorm (Zhang & Sennrich, 2019), and Adaptive LN (Xu et al., 2019), which are now standard in Transformers (Vaswani et al., 2017; Dosovitskiy et al., 2020). Group Normalization (GN) (Wu & He, 2018) and Batch Group Normalization (BGN) (Summers & Dinneen, 2019) extend this idea by combining grouping across features, channels, or batches. Collectively, these methods underscore normalization as a structural tool for scaling depth and stabilizing optimization (Huang et al., 2023; Xu et al., 2019; Joudaki et al., 2023).

Beyond variance-centric approaches, some works attempt to explicitly shape representation distributions. For example, Eftekhari & Papyan (2025) encourage Gaussian-like features via power transforms with additive noise, arguing that Gaussianity enhances representational capacity while noise acts as implicit regularization. However, this approach faces key limitations: it does not optimize mutual information with targets, incurs overhead in estimating power parameters, and introduces stochasticity that can destabilize training (see Appendix C).

Broadly, most normalization methods rely on re-centering and rescaling to mitigate internal covariate shift, yet they overlook the information-theoretic properties of representations, such as sufficiency and redundancy. This gap motivates our work. We propose **IBNorm**, which augments conventional normalization with compression operation designed to regulate information flow, guiding activations toward IB-optimal representations that enhance both generalization and robustness.

### 2.2 REPRESENTATION LEARNING WITH EXPLICIT INFORMATION BOTTLENECK OBJECTIVES

The Information Bottleneck (IB) principle (Tishby et al., 2000; Shamir et al., 2010; Tishby & Zaslavsky, 2015) formulates representation learning as a trade-off between sufficiency (preserving target-related information) and compression (compressing target-irrelevant noise). Prior representation learning works (Tishby & Zaslavsky, 2015; Alemi et al., 2016; Kolchinsky et al., 2019; Belghazi et al., 2018; Hu et al., 2024) have explored this principle primarily through empirical analysis and leveraging explicit IB objectives during deep learning model training.

Methods such as deep variational information bottleneck (VIB) (Alemi et al., 2016), nonlinear information bottleneck (Kolchinsky et al., 2019), and mutual information neural estimation (MINE) (Belghazi et al., 2018) leverage the IB principle by explicitly estimating mutual information through auxiliary neural MI estimators and incorporating these estimates as IB-regularization terms during training. These approaches typically rely on encoder–decoder architectures (e.g., VAEs), require large datasets to train accurate MI estimators prior to training the primary model, and introduce additional MI-based loss functions that cause both computational and hyperparameter overhead. The reliance on auxiliary estimators also imposes architectural constraints and significantly increases the computational burden, a limitation that becomes particularly pronounced in autoregressive LLMs, where the sequential decoding process complicates the design and increases the inference cost of MI estimators. Furthermore, as demonstrated by MINE (Belghazi et al., 2018), accurate MI estimation requires substantial sample complexity, making these methods expensive and often unstable when applied to long-context or large-scale models. Recent advanced techniques such as structured probabilistic coding (SPC) (Hu et al., 2024) incorporate IB-inspired coding mechanisms but are confined to specific encoder architectures and inherit similar computational and optimization challenges.

Thus, representation learning methods based on explicit IB objectives face substantial practical limitations: they impose heavy computational overhead due to auxiliary MI-estimator networks, introduce architectural constraints, require large sample sizes for accurate mutual information estimation, and may introduce optimization instability. These challenges are particularly pronounced in modern deep learning systems, such as large-scale autoregressive LLMs.

To effectively integrate the IB principle into modern deep learning models, IBNorm introduces the first framework to internalize the IB principle directly within the normalization operation. By redesigning the normalization layer to act as an information filter, IBNorm achieves the benefits of IB-guided compression—improved generalization and robustness—without the need for computationally expensive auxiliary estimators, external loss terms, or architectural constraints. This fills a critical gap in the literature by providing the first theoretical and practical framework for normalization design grounded in the Information Bottleneck principle. Appendix B.1 offers a comprehensive discussion of task-irrelevant information compression approaches in representation learning, while Appendix B.2 provides a further analysis of training pipelines based on explicit IB objectives.

## 3 PRELIMINARY

Here we introduce information bottleneck (IB) framework (Tishby et al., 2000; Shamir et al., 2010; Tishby & Zaslavsky, 2015) which forms the basis of our approach in Sec. 4.

Consider two continuous random variables $X$ and $Y$ with joint probability densities $p(x, y)$ over the space $\mathcal{X} \times \mathcal{Y}$. Then, we can define the mutual information between $X$ and $Y$:

$$I(X;Y) = \int_{x \in \mathcal{X}} \int_{y \in \mathcal{Y}} p(x, y) \log \frac{p(x, y)}{p(x)\, p(y)} \, dx \, dy. \tag{1}$$

Mutual information measures dependencies between random variables, and can be understood as how much knowing $X$ reduces the uncertainty in $Y$ or vice versa.

Building on this, the IB framework formulates representation learning as a trade-off between two goals: 1) minimality which compresses $X$ by reducing $I(X;T)$; and 2) sufficiency which preserves information about $Y$ by maximizing $I(Y;T)$. This is captured by the IB objective (Tishby & Zaslavsky, 2015):

$$\max_{T} \Big[ I(Y;T) - \beta I(X;T) \Big]. \tag{2}$$

where $\beta > 0$. Here, $T$ must satisfy the Markov chain $T - X - Y$, and the optimal representation $T^*$ is given by the conditional distribution $p_{T^*|X}(t|x)$ with marginal $p_{T^*}(t) = \int p_{T^*|X}(t|x)p_X(x)dx$ govern by self-consistency equations (Tishby et al., 2000; Shamir et al., 2010).

Intuitively, maximizing $I(Y;T)$ ensures that the representation $T$ preserving all information relevant for predicting the target variable $Y$, thereby enhancing predictive power and interpretability. Concurrently, penalizing $I(X;T)$ enforces compression, which eliminates irrelevant variability in the representation and, in turn, enhances generalization and robustness of the representation (Kawaguchi et al., 2023; Alemi et al., 2016; Goldfeld & Polyanskiy, 2020; Saxe et al., 2019; Federici et al., 2020).

## 4 METHODOLOGY

Inspired by the IB principle introduced in Sec. 3, we now present how it motivates a novel and more effective normalization strategy — an essential component of modern deep learning architectures.

### 4.1 NORMALIZATION DESIGN VIA IB PRINCIPLE

**Normalization decomposition and limitations.** Consider a data space $\mathcal{X}$ and label space $\mathcal{Y}$ with a fixed joint distribution $\mathbb{P}(X, Y) \in \mathcal{P}(\mathcal{X} \times \mathcal{Y})$. A feedforward neural network $f(\cdot; \theta)$, parameterized by $\theta$, maps an input $X \in \mathcal{X}$ to a prediction $\hat{Y} = f(X; \theta)$, aiming to approximate the ground-truth label $Y$. Here, $f(\cdot; \theta)$ can represent various architectures, such as convolutional neural networks (e.g., ResNet (He et al., 2016a)) and transformer-based networks (e.g., large language models like (Touvron et al., 2023b) or vision transformer (Dosovitskiy et al., 2020)).

For an $L$-layer network equipped with normalization, we write the intermediate representation after the $l$-th normalization as

$$T_l := \Phi_l \circ h_l \circ \cdots \circ \Phi_1 \circ h_1(X) \in \mathcal{T}_l, \quad l \in \{1, \cdots, L\}, \tag{3}$$

where $\mathcal{T}_l$ is a measurable space, $h_i$ is a transformation block with parameter $\theta_i$, and $\Phi_i$ is the normalization layer, e.g., layer normalization. We use symbol $\circ$ to represent the composition of functions. Following (Huang et al., 2023), any normalization layer $\Phi$ can be decomposed into three steps:

$$\Phi(\cdot) \equiv \eta \circ \psi \circ \zeta, \tag{4}$$

where $\zeta$ defines the *normalization area partitioning* (NAP), i.e., how features are grouped (e.g., batch-level in BN, feature-level in LN, or group-level in GN); $\psi$ is the *normalization operation* (NOP), i.e., the standardization operation; $\eta$ is the *normalization representation recovery* (NRR), typically an affine re-scaling and shifting.

This decomposition unifies existing normalization methods. For instance, BN (Ioffe & Szegedy, 2015), LN (Ba et al., 2016), and GN (Wu & He, 2018) differ mainly in their choice of NAP ($\zeta$): BN normalizes across the batch dimension, LN across the feature dimension, and GN across groups of features. In all cases, the NOP enforces zero-mean and unit-variance statistics, while the NRR rescales and shifts the result. Crucially, among them, NAP ($\zeta$) is often dictated by the network architecture or task domain (e.g., batch-based normalization favored by CNNs, feature-based normalization for Transformers). Likewise, NRR ($\eta$) is linear and invertible, and therefore only reparameterizes activations without altering their information content. This leaves the NOP ($\psi$) as the main degree of freedom: it governs how information is filtered and thus determines whether normalization merely stabilizes optimization or also enhances the quality of learned representations.

However, most existing NOP ($\psi$) operations are fundamentally variance-centric, enforcing zero-mean and unit-variance scaling of activations. While stabilizing and accelerating training, it overlooks whether the resulting representations retain task-relevant information. From the IB perspective in Sec. 3, it reveals a fundamental limitation: variance normalization alone cannot guarantee the balance between sufficiency (retaining predictive information) and minimality (removing redundancy). As a result, relying solely on variance-centric NOP ($\psi$) can produce representations with superior generalization and robustness, since higher IB values are empirically and theoretically linked to better generalization and robustness (Kawaguchi et al., 2023; Alemi et al., 2016; Goldfeld & Polyanskiy, 2020; Saxe et al., 2019; Achille & Soatto, 2018).

**IB-guided normalization objective.** The IB framework provides a principled way to rethink normalization. An ideal intermediate representation $T_l$ should satisfy two criteria: 1) preserve sufficiency, namely, maximizing the predictive information $I(Y; T_l)$ about the target and thus representation $T_l$ can easily predict the target $Y$; and 2) promote compression, i.e., minimizing task-nuisance information $I(T_{l-1}; T_l)$ carried over from the previous layer and thereby removing the potential unimportant noises or patterns. This intuition leads to the following multi-layer IB objective:

$$\max_{\{T_l\}_{l=1}^L} \sum_{l=1}^L \Big( I(Y; T_l) - \beta I(T_{l-1}; T_l) \Big), \tag{5}$$

where $\{T_l\}_{l=1}^L \in \mathcal{T}_1 \times \cdots \times \mathcal{T}_L \equiv \mathcal{T}^{\otimes L}$, $T_0 = X$ and $\beta > 0$ controls the trade-off between sufficiency and compression. Prior works (Kawaguchi et al., 2023; Saxe et al., 2019; Alemi et al., 2016;

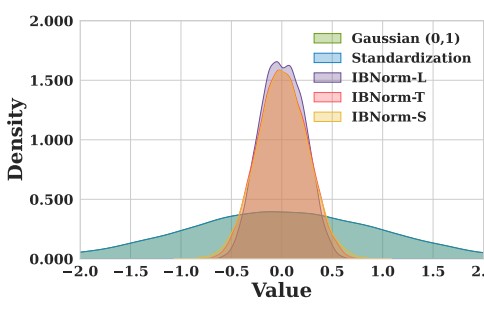 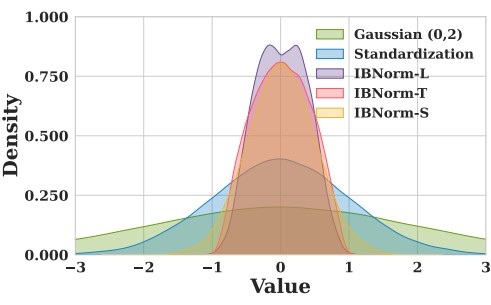

(a) Under Gaussian (0,1) input.      (b) Gaussian (0,2) input.

Figure 1: Comparison of kernel density estimation for Gaussian inputs (mean 0, varying variance) under different compression operations: Standardization, IBNorm-L, IBNorm-T, and IBNorm-S ($\lambda = 4$). Compression operations in IBNorm compress the tail of activations while adjusting higher-order statistics. IBNorm-L, IBNorm-T, and IBNorm-S, with the same $\lambda$, exhibit different abilities to compress the tails of activations. See more examples in Appendix D.5.

Goldfeld & Polyanskiy, 2020; Achille & Soatto, 2018) has shown that both generalization error and robustness positively correlate with the IB value in Eqn. (5), implying that aligning intermediate representations with the IB principle can improve generalization and robustness.

Building on this insight, we aim to enhance conventional variance-centric NOP ($\psi$) operator with an IB-inspired compression operation. Rather than treating normalization as a mere stabilizer, we reinterpret it as an information filter that produces representations that are both compact and predictive, bridging optimization stability and information-theoretic optimality. Directly optimizing Eqn. (5) as an additional IB-based training loss is infeasible due to two main reasons: (1) the computational cost of estimating mutual information using auxiliary MI neural estimators over the unknown joint distribution $p_{X,Y}$, and (2) the significant training overhead and sample complexity associated with such MI neural estimators (see detailed discussion in Appendix B.2). However, it provides a guiding principle: normalization should go beyond variance standardization to encourage information-preserving activations. We introduce our IB-inspired normalization in Sec.4.2 and theoretically justify its superiority and improved generalization in Sec. 4.3.

## 4.2 IB-INSPIRED NORMALIZATION

In the previous subsection, we identified a key limitation of existing normalization methods: their variance-centric design does not guarantee that intermediate representations preserve task-relevant information. Guided by the IB principle, our goal is to design a normalization operator that explicitly reshapes activations into information-preserving forms, rather than merely controlling their variance.

To this end, we propose an IB-inspired normalization. Unlike conventional variance-centric approaches, which manipulate only first- and second-order activation statistics via standardizing activations to have zero mean and unit variance, our approach introduces an extra compression operation into the normalization operation (NOP) $\psi$ that acts on higher-order statistics. Specifically, $\psi$ compresses activations toward their mean in a controlled manner, thereby increasing local kurtosis and inducing sparsity in activations across instances. Prior work has shown that mean-centered and sparse representations, where most activations concentrate around the mean, often exhibit stronger generalization since redundant and task-unrelated information are removed (Olshausen & Field, 1997; Ranzato et al., 2007; Bengio et al., 2013; Zhang et al., 2018; Guo et al., 2019). Motivated by this insight, our compression operation suppresses variability in activation tails, reshaping the higher-order distribution of activations rather than merely rescaling and shifting activations. More importantly, we find this design aligns normalization with the IB principle, preserving target-relevant information $I(Y; T_l)$ while suppressing irrelevant task-nuisance information $I(T_{l-1}; T_l)$.

**Compression operation in IBNorm.** Here we introduce the normalization operation (NOP) $\psi$ in our IBNorm. Let $X$ be hidden activations from a given layer, and let $\boldsymbol{x} = \{x_i\}_{i=1}^{H}$ represent a sample of $H$-dimensional activations. We define the compression operation $s_\lambda(\cdot; \lambda) : \mathbb{R}^d \to \mathbb{R}^d$ as

$$s_\lambda(x_i; \lambda) = \mu + \text{sign}(x_i - \mu) \cdot f_\lambda(|x_i - \mu|), \tag{6}$$

where $\mu = \frac{1}{H}\sum_{i=1}^{H} x_i$ is the mean activation, and $f_\lambda(|x_i - \mu|)$ is a measurable and strictly monotone non-decreasing function with a hyper-parameter $\lambda$ satisfying the *bounded compression property*:

$$0 \le f_\lambda(r) \le \alpha_\lambda r, \quad \forall r \ge 0, \quad \alpha_\lambda \in [0, 1]. \tag{7}$$

The property (7) reduces the spread of the activation distribution and increases local kurtosis and sparsity in activations by pushing more activations toward the mean. This sparsity effectively suppresses variability in the tail regions, which often encode task-irrelevant fluctuations in the high-dimensional activations (Olshausen & Field, 1997; Hyvärinen & Oja, 2000). As a result, the compression operation decreases task-nuisance information with the input, $I(T_{l-1}; T_l)$, while preserving the main mass of the distribution that carries task-relevant information, $I(Y; T_l)$. In other words, by promoting sparse and mean-centered activations, compression encourages representations that retain task relevant information for the downstream task while attenuating task-nuisance information, consistent with the IB principle, which is also empirically validated in Sec. 4.3.

We instantiate three representative forms of the compression function $f_\lambda$: IBNorm-S, IBNorm-L, and IBNorm-T. These functionals yield different compression behaviors:

$$f_\lambda(|x_i - \mu|) = |x_i - \mu|/\lambda, \qquad \textbf{(IBNorm-S}, \text{ linear compression)} \tag{8}$$
$$f_\lambda(|x_i - \mu|) = \ln(1 + |x_i - \mu|/\lambda), \qquad \textbf{(IBNorm-L}, \text{ logarithmic compression)} \tag{9}$$
$$f_\lambda(|x_i - \mu|) = \tanh(|x_i - \mu|/\lambda). \qquad \textbf{(IBNorm-T}, \text{ hyperbolic tangent compression)} \tag{10}$$

For these functionals, the compression ratio $\alpha_\lambda$ in Eqn. (7) becomes $\alpha_\lambda = 1/\lambda$, guaranteeing the bounded compression property whenever $\lambda \ge 1$. So $\lambda$ controls the compression strength: larger $\lambda$ enforces stronger compression toward the mean $\mu$, whereas smaller $\lambda$ lead to broader activations. By suppressing the tails of the activation distribution, this reduces task-nuisance information while preserving task-relevant information. See more rationale about compression operation in Appendix D.5.

**IBNorm.** Building on the foundation of LayerNorm, we introduce *IBNorm*, which integrates an explicit compression operation into the normalization pipeline. This design preserves the well-known benefits of LayerNorm—training stability and acceleration and architectural compatibility—while also incorporating the information-theoretic advantages of the IB principle. Finally, we derive three varaints: IBNorm-S, IBNorm-L, and IBNorm-T.

Formally, given an activation input $\boldsymbol{x}$, IBNorm applies the following sequence of operations: first, the NAP operator $\zeta$ partitions features in the same way as LayerNorm; next, a compression operator $s_\lambda$ reduces nuisance variability; then, the NOP operator $\psi$ standardizes activations; and finally, the NRR operator $\eta$ rescales and shifts them:

$$\text{IBNorm}(\boldsymbol{x}; \lambda) \equiv \eta \circ \psi \circ s_\lambda \circ \zeta. \tag{11}$$

This hybrid procedure, summarized in Algorithm 1 of Appendix B.3, yields a normalization strategy explicitly guided by the IB principle. We highlight that this differs from the function of nonlinear activations (see Appendix B.1). By construction, IBNorm goes beyond variance normalization: it acts as an *information filter*, enhancing sufficiency by retaining predictive information while improving minimality through compression of irrelevant information in the input. In this way, IBNorm bridges the gap between practical optimization benefits and information-theoretic optimality.

### 4.3 THEORETICAL JUSTIFICATION OF IBNORM

Here we first show that IBNorm achieves a larger IB value in Eqn. (5) than standard normalization methods, and then prove its superior generalization performance.

**Information-theoretic superiority of IBNorm.** To analyze the impact of normalization, we consider a simplified setting with two one-layer neural networks, $f_S = \Phi_s \circ h$ and $f_{IB} = \Phi_{IB} \circ h$, that share the same feature extractor $h$ but differ in their normalization layers: $\Phi_s \equiv \eta \circ \psi \circ \zeta$ in the baseline network $f_s$ and $\Phi_{IB} \equiv \eta \circ \psi \circ s_\lambda \circ \zeta$ in the network $f_{IB}$ with IBNorm.

Given the sample dataset $S = \{x_i, y_i\}_{i=1}^{M} \sim \mathbb{P}(X, Y)$ containing $M < \infty$ samples, we define the empirical IB value as $\widehat{\text{IB}}_S(T_1) := \hat{I}_S(Y; T_1) - \beta \hat{I}_S(X; T_1)$ based on Eqn. (5). Because $\eta$ is linear and invertible, it does not affect the mutual information terms; therefore, it suffices to analyze the IB value of intermediate features $\tilde{T}_s := \psi \circ \zeta \circ h(X)$ and $\tilde{T}_{IB} := \psi \circ s_\lambda \circ \zeta \circ h(X)$.

Based on the bounded compression property of IBNorm, we can establish the following relationship.

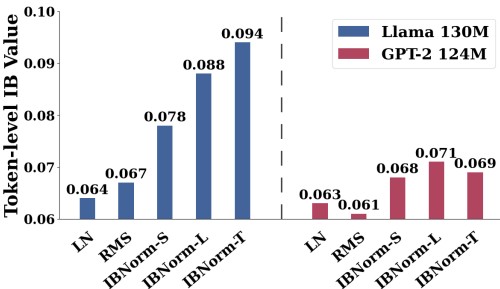 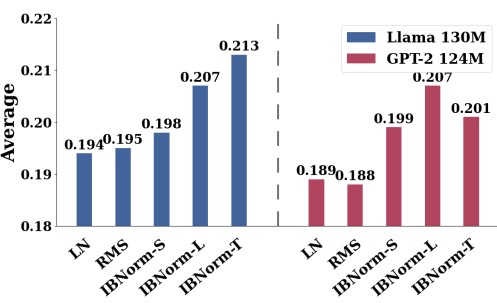

(a) Token-level IB values of Llama-130M and GPT-2 small trained on C4 and OpenWebText, respectively.

(b) Test performance of Llama-130M and GPT-2 small evaluated on the LLM Leaderboard II.

Figure 2: Evaluation of different normalization methods on Llama-130M and GPT-2 small. (a) shows token-level IB values evaluated at the test dataset when training only the normalization layers, and (b) reports test performance on the LLM Leaderboard II.

**Theorem 1** (IB Value). *For any hyperparameter $\beta \in [0,1]$ and the sample dataset $S \sim \mathbb{P}(X, Y)$ of size $M$, we have*

$$\widehat{IB}_S(\tilde{T}_{IB}) \geq \widehat{IB}_S(\tilde{T}_s) \quad \text{almost surely .}$$

The proof is provided in Appendix F.1. Theorem 1 establishes the theoretical advantage of IBNorm: its compressed features yield higher IB values than those from normalization with standardization, such as LN and BN. Intuitively, IBNorm better preserves label-relevant information $I(Y; \tilde{T}_{IB})$ while suppressing task-nuisance information $I(X; \tilde{T}_{IB})$, leading to more efficient and predictive representations. Crucially, this benefit holds uniformly across all trade-off parameters $\beta > 0$, underscoring the robustness of IBNorm in balancing compression and informativeness.

This result aligns with our empirical observation. We train Llama-130M (Touvron et al., 2023b) and GPT-2 small (Radford et al., 2019) with frozen parameters except the normalization layers on C4 (Raffel et al., 2020) and OpenWebText (Gokaslan & Cohen, 2019), respectively. As shown in Fig. 2a, IBNorm consistently achieves higher token-level IB values ($\beta = 1$) compared to variance-centric normalizations, confirming that the compression operation directly improves information-theoretic optimality. See details in token-level IB value calculation in Appendix H.

**Generalization superiority of IBNorm.** We next extend the analysis to deep multi-layer networks. Based on the one-layer case, we expect IBNorm to increase IB values layer by layer, thereby enhancing both expressiveness and compression throughout the hierarchy. To formalize this, we study the generalization gap which measures difference between the expected and empirical-training loss:

$$\text{gen}(S) := \mathbb{E}_{X,Y}[\mathcal{L}(f(X), Y)] - \frac{1}{M} \sum_{i=1}^{M} \mathcal{L}(f(x_i), y_i), \tag{12}$$

where $\mathcal{L}$ denotes a per-sample loss. As shown in prior works (Zhang et al., 2016; Neyshabur et al., 2017; Hoffer et al., 2017; Jakubovitz et al., 2019), a smaller gen($S$) indicates improved generalization performance and robustness both empirically and theoretically.

**Corollary 2** (Generalization Bound). *With probability at least $1 - \delta$ over training set $S$ of size $M < \infty$, the generalization error of a $L$-layer network $f_o$ satisfies*

$$\text{gen}(S; f_o) \leq U_o := \sum_{l=1}^{L} U_l^o, \quad \text{with} \quad U_l^o = \sqrt{\frac{I(X; \tilde{T}_l^o | Y) + C(S, f_l^o) + \log(\frac{2}{\delta})}{M}},$$

*where $\Phi_l^o$ is the hypothesis space associated with the $l$-th layer $f_l^o$, $\tilde{T}_l^o$ is the intermediate representation at layer $l$ before normalization representation recovery $\eta$, and $C(S, f_l^o)$ is a term depending on the cardinality of the hypothesis space $\Phi_l^o$ and the training set $S$. Here $f_o$ can be the network $f_S$ using standard normalization like LN and BN, or our network $f_{IB}$ with our IBNorm.*

See Appendix F.4 for the proof. Corollary 2 establishes that the overall generalization gap depends on the cumulative conditional mutual information across layers. This quantity,

$$I(X; \tilde{T}_l^o | Y) = I(X; \tilde{T}_l^o) - I(Y; \tilde{T}_l^o) = -\text{IB}(\tilde{T}_l^o) \quad (\beta = 1),$$

measures the amount of task-irrelevant input information preserved in the representations.

Table 1: Results of Llama models evaluated on LLM Leaderboard II and I. Due to space limitation, we defer the results of each task on Leadboard I to Tab. 6 of Appendix D.1.

| Model | Normalization | Leaderboard II | | | | | | | Leaderboard I |
| | | IFEval | BBH | MATH | GPQA | MUSR | MMLU-PRO | AVG | AVG |
|---|---|---|---|---|---|---|---|---|---|
| Llama 60M | LayerNorm | 0.1479 | 0.2970 | 0.0000 | 0.2576 | 0.3426 | 0.1071 | 0.1920 | 0.2912 |
| | RMSNorm | 0.1274 | 0.2951 | 0.0000 | 0.2634 | 0.3585 | 0.1117 | 0.1924 | 0.2894 |
| | NormalNorm | 0.1101 | 0.2876 | 0.0000 | 0.2676 | 0.3571 | 0.1166 | 0.1898 | 0.2825 |
| | IBNorm-S | 0.1915 | 0.2732 | 0.0000 | 0.2534 | 0.3651 | 0.1152 | 0.1997 | 0.2950 |
| | IBNorm-L | 0.1899 | 0.3001 | 0.0000 | 0.2772 | 0.3519 | 0.1137 | **0.2055** | **0.2955** |
| | IBNorm-T | 0.1754 | 0.3013 | 0.0008 | 0.2668 | 0.3598 | 0.1110 | 0.2025 | 0.2937 |
| Llama 130M | LayerNorm | 0.1507 | 0.2949 | 0.0000 | 0.2650 | 0.3417 | 0.1116 | 0.1940 | 0.2921 |
| | RMSNorm | 0.1274 | 0.3052 | 0.0008 | 0.2643 | 0.3572 | 0.1121 | 0.1945 | 0.2911 |
| | NormalNorm | 0.1385 | 0.2888 | 0.0000 | 0.2584 | 0.3611 | 0.1098 | 0.1928 | 0.2925 |
| | IBNorm-S | 0.1343 | 0.3114 | 0.0008 | 0.2601 | 0.3690 | 0.1151 | 0.1984 | 0.2933 |
| | IBNorm-L | 0.1836 | 0.3028 | 0.0000 | 0.2816 | 0.3638 | 0.1124 | 0.2074 | 0.2962 |
| | IBNorm-T | 0.1999 | 0.3053 | 0.0000 | 0.2777 | 0.3783 | 0.1165 | **0.2130** | **0.2970** |
| Llama 350M | LayerNorm | 0.1668 | 0.2927 | 0.0000 | 0.2659 | 0.3586 | 0.1178 | 0.2003 | 0.3062 |
| | RMSNorm | 0.1673 | 0.2999 | 0.0030 | 0.2667 | 0.3571 | 0.1121 | 0.2010 | 0.3025 |
| | NormalNorm | 0.1667 | 0.2940 | 0.0008 | 0.2458 | 0.3599 | 0.1103 | 0.1962 | 0.3035 |
| | IBNorm-S | 0.1447 | 0.3020 | 0.0030 | 0.2940 | 0.3585 | 0.1094 | 0.2019 | 0.3089 |
| | IBNorm-L | 0.1872 | 0.3135 | 0.0000 | 0.2968 | 0.3558 | 0.1162 | 0.2116 | **0.3101** |
| | IBNorm-T | 0.1797 | 0.3115 | 0.0008 | 0.2983 | 0.3788 | 0.1150 | **0.2140** | 0.3089 |
| Llama 1B | LayerNorm | 0.1701 | 0.3024 | 0.0038 | 0.2691 | 0.3683 | 0.1183 | 0.2053 | 0.3138 |
| | RMSNorm | 0.1693 | 0.3005 | 0.0008 | 0.2685 | 0.3585 | 0.1139 | 0.2019 | 0.3134 |
| | NormalNorm | 0.1679 | 0.2971 | 0.0030 | 0.2576 | 0.3656 | 0.1127 | 0.2006 | 0.3051 |
| | IBNorm-S | 0.1572 | 0.3118 | 0.0030 | 0.2958 | 0.3597 | 0.1130 | 0.2067 | 0.3184 |
| | IBNorm-L | 0.1875 | 0.3167 | 0.0038 | 0.2986 | 0.3717 | 0.1171 | 0.2159 | 0.3179 |
| | IBNorm-T | 0.1829 | 0.3155 | 0.0030 | 0.2994 | 0.3793 | 0.1169 | **0.2162** | **0.3199** |

By Theorem 1, IBNorm increases the IB value of the population distribution at each layer, thereby reducing $I(X; \tilde{T}_l^{\text{IB}}|Y)$ compared to $I(X; \tilde{T}_l^{\text{S}}|Y)$. Moreover, the model-complexity terms satisfy $C(S, f_l^{\text{IB}}) \equiv C(S, f_l^{\text{S}})$ for all $l$ (See Appendix F.4). Summing across layers, we obtain $U_{\text{IB}} \leq U_{\text{S}}$, which shows that the generalization bound of the IBNorm network is tighter than that of the standard network. This implies that $f_{\text{IB}}$ achieves superior generalization performance compared to $f_{\text{S}}$. In Fig. 2b, we follow the setting in Fig. 2a to train Llama-130M and GPT-2 small. When evaluated by LLM Leaderboard II (Fourrier et al., 2024), using IBNorm can improve generation performance on downstream tasks, consistent with the information-theoretic analysis (Kawaguchi et al., 2023).

## 5 EXPERIMENT

We evaluate three variants—IBNorm-S, IBNorm-L, and IBNorm-T—across LLM pretraining and vision classification, comparing against LN (Ba et al., 2016), BN (Ioffe & Szegedy, 2015), RMSNorm (Zhang & Sennrich, 2019), and NormalNorm (Eftekhari & Papyan, 2025).

**LLM pretraining.** We pretrain the LLaMA series (Touvron et al., 2023b) (60M-1B) on C4 (Raffel et al., 2020), following (Lialin et al., 2023; Zhao et al., 2024), and GPT-2 Small (124M) and Medium (355M) on OpenWebText (Gokaslan & Cohen, 2019), following the Sophia setup (Liu et al., 2023) with the nanoGPT implementation (Karpathy, 2022). See details in Appendix G.1. We report performance on LLM Leaderboards I & II (Beeching et al., 2023; Fourrier et al., 2024).

**Vision training.** We train ResNet-18 on CIFAR-10 (Krizhevsky et al., 2009) and ResNet-50 and ViT on ImageNet (Deng et al., 2009), following the setup (Eftekhari & Papyan, 2025). Both top-1 and top-5 accuracies are reported. Detailed settings are provided in Appendix G.2. Extended results of ViT-S/16 and ViT-B/16 (Dosovitskiy et al., 2020) on ImageNet can be found in Appendix D.4.

### 5.1 MAIN RESULTS

**Results on LLM.** Considering the space limitation and the fact that LLM Leaderboard II benchmark can more effectively evaluate the performance and is more widely used now, we defer the results of each task in Leadboard I benchmark to Appendix D.1, and only report its task average result.

As shown in Tab. 1, IBNorm outperforms variance-centric normalization methods such as LN and RMSNorm. On LLaMA-60M, IBNorm-L achieves average scores of 0.2955 (Leaderboard I) and 0.2055 (Leaderboard II), improving over LN by 1.47% and 7.03%, and over RMSNorm by 2.11%

Table 2: Results of GPT-2 models evaluated on LLM Leaderboard II and I. Due to space limitation, we defer the results of each task on Leadboard I to Tab. 7 of Appendix D.1.

| Model | Normalization | Leaderboard II | | | | | | | Leaderboard I |
|---|---|---|---|---|---|---|---|---|---|
| | | IFEval | BBH | MATH | GPQA | MUSR | MMLU-PRO | AVG | AVG |
| GPT-2 124M | LayerNorm | 0.1010 | 0.2809 | 0.0000 | 0.2718 | 0.3624 | 0.1158 | 0.1886 | 0.2887 |
| | RMSNorm | 0.0966 | 0.2772 | 0.0008 | 0.2651 | 0.3730 | 0.1154 | 0.1880 | 0.2860 |
| | NormalNorm | 0.1499 | 0.2913 | 0.0000 | 0.2693 | 0.3690 | 0.1119 | 0.1986 | 0.2927 |
| | IBNorm-S | 0.1860 | 0.2793 | 0.0000 | 0.2619 | 0.3540 | 0.1149 | 0.1994 | 0.2934 |
| | IBNorm-L | 0.2207 | 0.2850 | 0.0000 | 0.2650 | 0.3558 | 0.1134 | **0.2067** | **0.2970** |
| | IBNorm-T | 0.1836 | 0.2810 | 0.0000 | 0.2697 | 0.3571 | 0.1158 | 0.2012 | 0.2952 |
| GPT-2 355M | LayerNorm | 0.1128 | 0.2835 | 0.0000 | 0.2799 | 0.3661 | 0.1106 | 0.1922 | 0.2983 |
| | RMSNorm | 0.1395 | 0.2796 | 0.0000 | 0.2694 | 0.3707 | 0.1090 | 0.1947 | 0.2950 |
| | NormalNorm | 0.1425 | 0.2899 | 0.0000 | 0.2702 | 0.3708 | 0.1187 | 0.1987 | 0.2955 |
| | IBNorm-S | 0.2069 | 0.2802 | 0.0000 | 0.2716 | 0.3692 | 0.1176 | 0.2076 | 0.3011 |
| | IBNorm-L | 0.2258 | 0.2867 | 0.0000 | 0.2771 | 0.3687 | 0.1163 | **0.2124** | **0.3035** |
| | IBNorm-T | 0.1964 | 0.2824 | 0.0008 | 0.2780 | 0.3719 | 0.1183 | 0.2080 | **0.3035** |

Table 3: Accuracy (%) on image classification tasks. We report BatchNorm's results for CNNs (ResNet18/50) since it significantly outperforms LayerNorm on CNNs, and present LayerNorm's results on Transformers (ViT) because it performs much better than BatchNorm.

| Normalization | ResNet18 on CIFAR-10 | | ResNet50 on ImageNet | | ViT on ImageNet | |
|---|---|---|---|---|---|---|
| | Top-1 | Top-5 | Top-1 | Top-5 | Top-1 | Top-5 |
| BatchNorm | 93.65 | 99.85 | 76.15 | 92.86 | — | — |
| LayerNorm | — | — | — | — | 71.54 | 89.40 |
| RMSNorm | — | — | — | — | 71.03 | 89.15 |
| NormalNorm | 94.01 | 99.86 | 77.29 | 94.04 | 75.25 | **92.23** |
| IBNorm-S | 95.49 | 99.85 | 78.85 | 94.03 | 75.82 | 91.97 |
| IBNorm-L | **95.57** | **99.87** | **79.18** | **94.82** | **76.83** | 92.22 |
| IBNorm-T | 95.51 | 99.83 | 78.61 | 93.79 | 76.46 | 92.19 |

and 6.81%. On LLaMA-350M, IBNorm-T reaches 0.3101 (Leaderboard I) and 0.2140 (Leaderboard II), *i.e.* 1.27% and 6.84% over LN, and 2.51% and 6.46% over RMSNorm. On LLaMA-1B, IBNorm-T achieves the highest scores: 0.3199 on Leaderboard I and 0.2162 on Leaderboard II. Similarly, Tab. 2 also shows that on GPT-2 (355M), IBNorm-L attains 0.3035 (Leaderboard I) and 0.2124 (Leaderboard II), yielding improvements of 2.88% and 9.95% over RMSNorm.

These results demonstrate that IBNorm enhances representation quality in large language models, yielding more predictive and generalizable representations. Further analysis of mutual information and token-level IB values in Appendix D.6 confirms that IBNorm promotes representations that retain task-relevant information while suppressing nuisances, explaining the observed empirical gains. We find that IBNorm shows some feailure case in some tasks and defer the discussion to Appendix D.1.

**Results for vision classification.** Tab. 3 shows that IBNorm also improves performance across vision models and datasets. On CIFAR-10, ResNet18 with IBNorm-L achieves 95.57% top-1 accuracy, surpassing BN and NormalNorm by 1.92% and 1.56%. On ImageNet, ResNet50 with IBNorm-L improves over BN and NormalNorm by 3.03% and 1.89%. For ViT, IBNorm-L yields top-1 accuracy gains of 5.29%, 5.80%, and 1.58% over LN, RMSNorm, and NormalNorm, respectively.

## 5.2 ABLATION STUDY

**Effects of compression hyper-parameter** $\lambda$. We analyze the effect of the hyperparameter $\lambda$ in our proposed IBNorm, which controls the strength of the compression on intermediate activations. Specifically, we consider $\lambda = 0.5, 4, 8$, corresponding to mild, moderate, and strong compression, respectively. As shown in Tab. 4, the results show that moderate compression ($\lambda = 4$) achieves the best performance, striking a balance between preserving task-relevant information and suppressing irrelevant variability. For instance, IBNorm-S achieves the best scores of 0.2952 and 0.1987. We find that overly mild and strong compression lead to suboptimal performance, excessively reducing the representation diversity.

Table 4: Ablation study of Llama 60M with normalization using different $\lambda$ evaluated on LLM Leaderboard II. We defer the results of each task on Leadboard I to Tab. 11 of Appendix D.1.

| Normalization | $\lambda$ | Leaderboard II | | | | | | | Leaderboard I |
| | | IFEval | BBH | MATH | GPQA | MUSR | MMLU-PRO | AVG | AVG |
|---|---|---|---|---|---|---|---|---|---|
| IBNorm-S | 0.5 | 0.1687 | 0.2894 | 0.0008 | 0.2592 | 0.3519 | 0.1162 | 0.1977 | 0.2898 |
| | 4 | 0.1781 | 0.2977 | 0.0000 | 0.2501 | 0.3571 | 0.1091 | **0.1987** | **0.2952** |
| | 8 | 0.1623 | 0.2947 | 0.0008 | 0.2606 | 0.3515 | 0.1185 | 0.1981 | 0.2918 |
| IBNorm-L | 0.5 | 0.1577 | 0.2968 | 0.0000 | 0.2710 | 0.3690 | 0.1162 | 0.2018 | 0.2863 |
| | 4 | 0.1899 | 0.3001 | 0.0000 | 0.2772 | 0.3519 | 0.1137 | **0.2055** | **0.2955** |
| | 8 | 0.1364 | 0.2888 | 0.0000 | 0.2450 | 0.3452 | 0.1132 | 0.1881 | 0.2921 |
| IBNorm-T | 0.5 | 0.1761 | 0.3024 | 0.0000 | 0.2492 | 0.3469 | 0.1158 | 0.1984 | 0.2881 |
| | 4 | 0.1754 | 0.3013 | 0.0008 | 0.2668 | 0.3598 | 0.1110 | **0.2025** | **0.2937** |
| | 8 | 0.1753 | 0.2933 | 0.0000 | 0.2559 | 0.3631 | 0.1156 | 0.2005 | 0.2898 |

Table 5: Ablation study of Llama 60M evaluated on LLM Leaderboard II. Due to space limitation, we defer the results of each task on Leadboard I to Tab. 10 of Appendix D.1.

| Normalization | Leaderboard II | | | | | | | Leaderboard I |
| | IFEval | BBH | MATH | GPQA | MUSR | MMLU-PRO | AVG | AVG |
|---|---|---|---|---|---|---|---|---|
| IBNorm-S* | 0.1626 | 0.2646 | 0.0000 | 0.2876 | 0.3609 | 0.1114 | 0.1978 | 0.2952 |
| IBNorm-S** | 0.1376 | 0.2946 | 0.0000 | 0.2776 | 0.3460 | 0.1114 | 0.1945 | 0.2881 |
| IBNorm-T* | 0.1372 | 0.2880 | 0.0000 | 0.2801 | 0.3519 | 0.1166 | 0.1956 | 0.2936 |
| IBNorm-T** | 0.1272 | 0.2980 | 0.0000 | 0.2701 | 0.3519 | 0.1166 | 0.1940 | 0.2831 |
| IBNorm-L* | 0.1374 | 0.2941 | 0.0000 | 0.2793 | 0.3505 | 0.1190 | 0.1967 | 0.2905 |
| IBNorm-L** | 0.1374 | 0.2841 | 0.0000 | 0.2693 | 0.3505 | 0.1190 | 0.1934 | 0.2859 |

**Effects of compression operation and linear affine reparameterization.** To further clarify the design choices underlying our proposed IBNorm, we perform ablation studies on two key components: the compression operation and the standardization step. As reported in Tab. 5, the variant of IBNorm marked with an asterisk (*) applies standardization before the compression operation, namely, $\text{IBNorm}(\boldsymbol{x}; \lambda) \equiv \eta \circ s_\lambda \circ \psi \circ \zeta$. This modification leads to only minor fluctuations in performance compared to the baseline: for example, the baseline achieves average scores of 0.2950 and 0.1997, while IBNorm-S* obtains 0.2952 and 0.1978 on Leaderboard I and II, respectively. These results indicate that applying compression operation prior to standardization is preferable for optimal performance. In addition, the variant denoted by a double asterisk (**) removes the linear affine component, which results in a substantial performance drop across benchmarks. For instance, IBNorm-L achieves 0.2955 and 0.2055, whereas IBNorm-L** drops to 0.2859 and 0.1934 on Leaderboard I and II. This pronounced degradation highlights the critical role of the affine reparameterization ($\eta$) in the IBNorm design.

## 6 CONCLUSION

We revisited normalization through the lens of IB principle and proposed a principled framework for designing normalization layers. Then, we introduced **IBNorm**, a family of nonlinear transforms that compress activations toward their mean, yielding more information-rich representations. Without relying on explicit IB objectives during training, IBNorm remains model-agnostic and incurs no heavy computational overhead. Theoretically, we showed that IBNorm achieves stronger information-theoretic optimality—attaining higher IB values and tighter generalization bounds—than variance-centric methods. Empirically, IBNorm outperforms BN, LN, and RMSNorm across both vision and language domains.

**Limitations.** Our experiments are limited to medium-scale LLMs due to computational constraints. Extending the evaluation to larger foundation models will be addressed when sufficient computational resources are available.

## ETHICS STATEMENT

The large language models employed in this research may reflect biases or generate sensitive or potentially offensive content, intended solely for academic and scientific purposes. The opinions expressed within generated outputs do not represent the views of the authors. We remain committed to fostering the development of AI technologies which align with ethical standards and reflect societal values.

## REPRODUCIBILITY STATEMENT

We detail our work in the Methodology section (Sec. 4) and describe implementation details in Sec. 5 and Appendix G.

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

CONTENTS OF APPENDIX

## A  DECLARATION OF LLM USAGE

LLMs were used solely for polishing and formatting text. All research ideas, methods, and experimental results presented are original contributions of the authors.

# B  IBNORM

## B.1  COMPARISON WITH TASK-IRRELEVANT INFORMATION COMPRESSION METHODS IN REPRESENTATION LEARNING

Prior works (Tishby et al., 2000; Tishby & Zaslavsky, 2015; Saxe et al., 2019; Hua et al., 2021; Weng et al., 2023; Hu et al., 2025) explore the IB theory and representation learning methods, such as whitening, and empirically shows how hidden layers with activation nonlinearities and whitening modules aiming to decorrelate features and stabilize training can compress task-irrelevant information.

While these findings provide valuable empirical insights, there are two notable gaps unaddressed: (1) they do not examine how normalization layers influence the decomposition of task-relevant and task-irrelevant mutual information within neural representations, and (2) they do not provide a theoretical framework for how to design the normalization following the IB framework to facilitate more informative and robust representations. Importantly, activation and normalization are conceptually and structurally distinct: In modern LLMs (e.g., LLaMA, GPT), nonlinear activations occur only in the gated branch of the MLP, and do not influence the representations within each transformer block. In these architectures, inserting tanh as an "activation" would not propagate compression throughout the model, since the gating branch represents only one component of the block.

In contrast, IBNorm applies compression directly the normalization layer, which is invoked throughout the model—after each attention block and MLP block in LLMs. This placement ensures that compression systematically modulates the IB value of the representation at every layer, producing effects that cannot be achieved by modifying the activation function alone. Thus, even if tanh exhibits compression as an activation function, it does not serve the same purpose, does not affect the same parts of the network, and does not shape representation statistics in the same way as normalization-based compression in IBNorm. Moreover, unlike tanh, which is only empirically shown to induce compression, our proposed IBNorm is explicitly derived from an IB-theoretic analysis of normalization. Our analysis demonstrates how different IB-motivated compression operations shape higher-order statistics and directly align normalization behavior with the IB principle.

## B.2  COMPARISON WITH TRAINING PIPELINES BASED ON EXPLICIT INFORMATION BOTTLENECK OBJECTIVES

Methods (Kolchinsky et al., 2019; Alemi et al., 2016; Belghazi et al., 2018) focus on estimating mutual information by parameterizing the IB principle with pretrained neural MI estimators and then guiding the training of deep learning models using the explicit estimated IB objective. These approaches typically rely on encoder–decoder architectures (e.g., VAEs), require large training datasets to train accurate and lightweight MI estimators before training the deep learning models, and introduce additional estimated MI-regularization loss functions during training the deep learning models. These components introduce extra hyperparameters, increase computational cost, and may lead to unstable optimization. Specifically, these approaches face two key limitations when applied to deep learning models, such as autoregressive LLMs, in practice:

**MI Neural Estimator Architecture and Computational Overhead:** These methods require additional MI neural estimators with architectural constraints alongside the main network, significantly increasing heavy inference computational cost and complicating optimization. In addition, designing such neural estimators is particularly difficult for autoregressive LLMs due to their sequential decoding characteristics.

**Sample Complexity:** Accurate MI estimation, as noted in MINE (Belghazi et al., 2018), requires large sample sizes, which is challenging for models with long-context inputs and requires heavy training cost.

IBNorm fills this gap by providing the first theoretical analysis of normalization under the IB principle. Specifically, IBNorm incorporates the IB principle within the normalization operation. It requires no auxiliary MI estimators, no additional IB-based training losses, and no architectural constraints. The compression operation introduced in IBNorm is not an external module but a theoretically grounded reformulation of normalization that increases the IB value of intermediate representations while adding minimal computational overhead.

We highlight the key advantages of IBNorm here:

**Systematic Theoretical Grounding:** We provide a theoretical analysis of widely used variance-centric normalization methods. This analysis motivates a principled mechanism for designing normalization layers that follow IB principle, as shown in Section 4.3.

**Computational Efficiency:** Unlike (Kolchinsky et al., 2019; Alemi et al., 2016; Belghazi et al., 2018), IBNorm does not rely on computationally heavy MI neural estimators or explicit IB regularizers, making it more efficient and stable during training. We include the comparison of training time and VRAM across IBNorm and baselines in Appendix D.7.

**Compatibility:** Probabilistic coding method like Structured Probabilistic Coding (SPC) (Hu et al., 2024) introduces an IB-inspired coding mechanism, but it is limited to encoder-only models and does not examine normalization layers. IBNorm is model-agnostic and can be directly integrated into Transformers (e.g., LLaMA, GPT-2) and CNNs (e.g., ResNet) without modifying the training pipeline, as demonstrated in Section 5.

### B.3 IBNorm Algorithm

Formally, given an activation input $\boldsymbol{x}$, IBNorm applies the following sequence of operations: first, the NAP operator $\zeta$ partitions features in the same way as LayerNorm; next, a compression operator $s_\lambda$ reduces nuisance variability; then, the NOP operator $\psi$ standardizes activations; and finally, the NRR operator $\eta$ rescales and shifts them:

$$\text{IBNorm}(\boldsymbol{x}; \lambda) \equiv \eta \circ \psi \circ s_\lambda \circ \zeta. \tag{13}$$

Here we summarize the IBNorm algorithmic steps in Algorithm 1. By construction, IBNorm goes beyond variance normalization: it acts as an *information filter*, enhancing sufficiency by retaining predictive information while improving minimality through compression of irrelevant information in the input. In this way, IBNorm bridges the gap between practical optimization benefits and information-theoretic optimality.

---

**Algorithm 1** IB-inspired normalization (IBNorm)

---

1: **Input:** Activations $\boldsymbol{x} \in \mathbb{R}^{d \times m \times h \times w}$, hyperparameter $\lambda$
2: **Output:** Normalized activations $\boldsymbol{y} \in \mathbb{R}^{d \times m \times h \times w}$
3: Normalization area partitioning (NAP): $\hat{\boldsymbol{x}} \leftarrow \zeta(\boldsymbol{x})$
4: Compression operation: $\tilde{\boldsymbol{x}} \leftarrow s_\lambda(\hat{\boldsymbol{x}})$
5: Normalization operation (NOP): $\bar{\boldsymbol{x}} \leftarrow \psi(\tilde{\boldsymbol{x}})$
6: Normalization representation recovery (NRR): $\hat{\boldsymbol{y}} \leftarrow \eta(\bar{\boldsymbol{x}})$
7: Reshape back: $\boldsymbol{y} \leftarrow \zeta^{-1}(\hat{\boldsymbol{y}})$

---

## C  NormalNorm

We discuss the difference of our proposed IBNorm and NormalNorm here. We first provide the specific algorithm of NormalNorm in Algorithm 2.

**Power Transform.** Consider a random variable $X$ from which a sample $\boldsymbol{x} = \{x_i\}_{i=1}^H$ is obtained. The power transform $g(\cdot; \lambda)$ gaussianizes $\boldsymbol{x}$ by applying the following function for each $x_i$:

$$g(x_i; \lambda) = \begin{cases} \frac{(x_i+1)^\lambda - 1}{\lambda}, & x_i \geq 0, \ \lambda \neq 0 \\ \log(x_i + 1), & x_i \geq 0, \ \lambda = 0 \\ -\frac{(-x_i+1)^{2-\lambda} - 1}{2-\lambda}, & x_i < 0, \ \lambda \neq 2 \\ -\log(-x_i + 1), & x_i < 0, \ \lambda = 2, \end{cases} \tag{14}$$

where $\lambda$ is a transformation parameter typically estimated from the data to make the transformed values approximately normally distributed.

The key design difference lies in how NormalNorm and IBNorm shape hidden activation distributions. NormalNorm aims to maximize the entropy of hidden activations by applying a power

transform after standardization. In contrast, our proposed IBNorm follows the IB principle and shapes the activation distributions via a *compression operation* applied *before* standardization.

While NormalNorm also attempts to explicitly shape activation distributions, encouraging Gaussian-like features through the power transform with additive noise, this approach has several limitations: 1) In practice, we find that dynamically varying $\lambda$ not only increases computational cost (see Appendix D.7), but can also lead to training instability due to numerical issues, especially in the setting of lower precision like bfloat16. 2) Furthermore, because the power transform modifies both the first- and second-order statistics of the hidden activations, the resulting outputs after the NOP no longer have zero mean and unit variance. As a result, the network cannot fully benefit from traditional normalization techniques, such as improved training stability and scale invariance.

---

**Algorithm 2** Normal Normalization

---

**Input:** $\boldsymbol{u} = \{u_i\}_{i=1}^{N}$
**Input:** $\boldsymbol{y} = \{y_i\}_{i=1}^{N}$
**Learnable Parameters:** $\gamma$, $\beta$
**Noise Factor:** $\xi \geq 0$
**Standardization:**
$\hat{u} \leftarrow \frac{1}{N} \sum_{i=1}^{N} u_i$
$\hat{\sigma}^2 \leftarrow \frac{1}{N} \sum_{i=1}^{N} (u_i - \hat{u})^2$
$c_i \leftarrow \frac{u_i - \hat{\mu}}{\sqrt{\hat{\sigma}^2 + \epsilon}}$
**Power Transform and Scaled Additive Noise:**
$x_i \leftarrow g(c_i; \hat{\lambda})$
with gradient tracking disabled:
$\bar{x} = \frac{1}{N} \sum_{i=1}^{N} x_i$
$s = \frac{1}{N} \sum_{i=1}^{N} |x_i - \bar{x}|$
sample $z_i \sim \mathcal{N}(0, 1)$
$v_i = x_i + z_i \cdot s \cdot \xi$
**Affine Transform:**
$y_i \leftarrow \gamma \cdot v_i + \beta$

---

# D   EXTENDED EMPIRICAL RESULTS

## D.1   RESULTS ON LLM LEADERBOARD I

Table 6: Results of Llama models evaluated on LLM Leaderboard I.

| Model | Normalization | ARC | HellaSwag | MMLU | TruthfulQA | Winogrande | GSM8K | AVG |
|---|---|---|---|---|---|---|---|---|
| Llama 60M | LayerNorm | 0.2150 | 0.2725 | 0.2557 | 0.4935 | 0.5075 | 0.0031 | 0.2912 |
| | RMSNorm | 0.2244 | 0.2783 | 0.2496 | 0.4761 | 0.5059 | 0.0023 | 0.2894 |
| | NormalNorm | 0.2270 | 0.2504 | 0.2295 | 0.4925 | 0.4957 | 0.0000 | 0.2825 |
| | IBNorm-S | 0.2631 | 0.2616 | 0.2526 | 0.4899 | 0.5028 | 0.0000 | 0.2950 |
| | IBNorm-L | 0.2568 | 0.2651 | 0.2689 | 0.4966 | 0.4854 | 0.0000 | **0.2955** |
| | IBNorm-T | 0.2270 | 0.2758 | 0.2620 | 0.4827 | 0.5099 | 0.0049 | 0.2937 |
| Llama 130M | LayerNorm | 0.2312 | 0.2795 | 0.2562 | 0.4754 | 0.5059 | 0.0042 | 0.2921 |
| | RMSNorm | 0.2210 | 0.2950 | 0.2494 | 0.4639 | 0.5154 | 0.0019 | 0.2911 |
| | NormalNorm | 0.2244 | 0.2684 | 0.2621 | 0.4797 | 0.5178 | 0.0027 | 0.2925 |
| | IBNorm-S | 0.2295 | 0.2950 | 0.2616 | 0.4698 | 0.5012 | 0.0027 | 0.2933 |
| | IBNorm-L | 0.2577 | 0.2697 | 0.2697 | 0.4892 | 0.4891 | 0.0019 | 0.2962 |
| | IBNorm-T | 0.2267 | 0.2900 | 0.2619 | 0.4949 | 0.5043 | 0.0042 | **0.2970** |
| Llama 350M | LayerNorm | 0.2355 | 0.3286 | 0.2579 | 0.4889 | 0.5225 | 0.0038 | 0.3062 |
| | RMSNorm | 0.2432 | 0.3155 | 0.2513 | 0.5201 | 0.4808 | 0.0038 | 0.3025 |
| | NormalNorm | 0.2398 | 0.3031 | 0.2618 | 0.4998 | 0.5107 | 0.0057 | 0.3035 |
| | IBNorm-S | 0.2363 | 0.3463 | 0.2620 | 0.4917 | 0.5149 | 0.0023 | 0.3089 |
| | IBNorm-L | 0.2619 | 0.3342 | 0.2692 | 0.4966 | 0.4949 | 0.0038 | **0.3101** |
| | IBNorm-T | 0.2598 | 0.3346 | 0.2623 | 0.4957 | 0.4970 | 0.0038 | 0.3089 |
| Llama 1B | LayerNorm | 0.2364 | 0.3711 | 0.2580 | 0.4882 | 0.5226 | 0.0064 | 0.3138 |
| | RMSNorm | 0.2517 | 0.3593 | 0.2522 | 0.5035 | 0.5093 | 0.0045 | 0.3134 |
| | NormalNorm | 0.2244 | 0.3183 | 0.2713 | 0.4961 | 0.5170 | 0.0038 | 0.3051 |
| | IBNorm-S | 0.2551 | 0.3749 | 0.2636 | 0.4923 | 0.5188 | 0.0057 | 0.3184 |
| | IBNorm-L | 0.2632 | 0.3725 | 0.2691 | 0.4945 | 0.4980 | 0.0099 | 0.3179 |
| | IBNorm-T | 0.2654 | 0.3809 | 0.2663 | 0.4952 | 0.5075 | 0.0042 | **0.3199** |

Table 7: Results of GPT-2 models evaluated on LLM Leaderboard I.

| Model | Normalization | ARC | HellaSwag | MMLU | TruthfulQA | Winogrande | GSM8K | AVG |
|---|---|---|---|---|---|---|---|---|
| GPT-2 124M | LayerNorm | 0.2483 | 0.2667 | 0.2371 | 0.4868 | 0.4933 | 0.0000 | 0.2887 |
| | RMSNorm | 0.2551 | 0.2540 | 0.2373 | 0.4848 | 0.4838 | 0.0008 | 0.2860 |
| | NormalNorm | 0.2645 | 0.0000 | 0.2632 | 0.2457 | 0.4890 | 0.4941 | 0.2927 |
| | IBNorm-S | 0.2690 | 0.2558 | 0.2458 | 0.4846 | 0.5053 | 0.0000 | 0.2934 |
| | IBNorm-L | 0.2756 | 0.2577 | 0.2689 | 0.4859 | 0.4933 | 0.0008 | **0.2970** |
| | IBNorm-T | 0.2734 | 0.2569 | 0.2394 | 0.4835 | 0.5178 | 0.0004 | 0.2952 |
| GPT-2 355M | LayerNorm | 0.2637 | 0.2772 | 0.2485 | 0.4945 | 0.5051 | 0.0011 | 0.2983 |
| | RMSNorm | 0.2643 | 0.2673 | 0.2518 | 0.4951 | 0.4917 | 0.0000 | 0.2950 |
| | NormalNorm | 0.2601 | 0.2683 | 0.2549 | 0.4948 | 0.4949 | 0.0000 | 0.2955 |
| | IBNorm-S | 0.2781 | 0.2679 | 0.2581 | 0.4955 | 0.5072 | 0.0000 | 0.3011 |
| | IBNorm-L | 0.2813 | 0.2681 | 0.2693 | 0.4964 | 0.5058 | 0.0000 | **0.3035** |
| | IBNorm-T | 0.2819 | 0.2693 | 0.2532 | 0.4972 | 0.5185 | 0.0011 | **0.3035** |

Table 8: Results of Llama models evaluated on LLM Leaderboard I and II.

| Model | Normalization | Leaderboard I | Leaderboard II |
|---|---|---|---|
| Llama 60M | LayerNorm | $0.2910 \pm 0.0020$ | $0.1922 \pm 0.0033$ |
| | RMSNorm | $0.2895 \pm 0.0027$ | $0.1927 \pm 0.0038$ |
| | NormalNorm | $0.2823 \pm 0.0029$ | $0.1895 \pm 0.0053$ |
| | IBNorm-L | $\mathbf{0.2955 \pm 0.0022}$ | $\mathbf{0.2055 \pm 0.0031}$ |
| Llama 130M | LayerNorm | $0.2919 \pm 0.0021$ | $0.1942 \pm 0.0035$ |
| | RMSNorm | $0.2913 \pm 0.0027$ | $0.1940 \pm 0.0039$ |
| | NormalNorm | $0.2918 \pm 0.0031$ | $0.1927 \pm 0.0055$ |
| | IBNorm-T | $\mathbf{0.2973 \pm 0.0024}$ | $\mathbf{0.2133 \pm 0.0037}$ |
| Llama 350M | LayerNorm | $0.3062 \pm 0.0020$ | $0.2003 \pm 0.0037$ |
| | RMSNorm | $0.3025 \pm 0.0028$ | $0.2010 \pm 0.0041$ |
| | NormalNorm | $0.3035 \pm 0.0033$ | $0.1962 \pm 0.0058$ |
| | IBNorm-L | $\mathbf{0.3101 \pm 0.0021}$ | $\mathbf{0.2116 \pm 0.0047}$ |
| | IBNorm-T | $\mathbf{0.3089 \pm 0.0022}$ | $\mathbf{0.2140 \pm 0.0050}$ |

Table 9: Results of GPT-2 models evaluated on LLM Leaderboard I and II.

| Model | Normalization | Leaderboard I | Leaderboard II |
|---|---|---|---|
| GPT-2 124M | LayerNorm | $0.2888 \pm 0.0017$ | $0.1889 \pm 0.0021$ |
| | RMSNorm | $0.2861 \pm 0.0019$ | $0.1883 \pm 0.0025$ |
| | NormalNorm | $0.2923 \pm 0.0027$ | $0.1983 \pm 0.0031$ |
| | IBNorm-L | $\mathbf{0.2969 \pm 0.0022}$ | $\mathbf{0.2068 \pm 0.0025}$ |

**Discussion of Failure Cases.** We acknowledge that IBNorm does not consistently outperform all baselines across every task in LLM Leaderboard I and II. Upon our careful examination, the tasks where IBNorm underperforms, such as BBH, GPQA, and MMLU-PRO, are highly challenging knowledge-intensive benchmarks, with questions crafted by domain experts across diverse fields. This contrasts with tasks such as factual retrieval, where IBNorm consistently improves performance.

IBNorm is designed to enhance representation expressiveness via the information bottleneck principle, improving generalization and robustness. However, performance on reasoning-intensive tasks may not be directly correlated with the information bottleneck value. Importantly, IBNorm still contributes to the model's generalization capabilities by shaping more informative hidden representations across diverse tasks.

## D.2 ABLATION STUDY ON NORMALIZATION STRUCTURE

Table 10: Ablation study of Llama 60M evaluated on LLM Leaderboard I.

| Normalization | ARC | HellaSwag | MMLU | TruthfulQA | Winogrande | GSM8K | AVG |
|---|---|---|---|---|---|---|---|
| IBNorm-S* | 0.2661 | 0.2698 | 0.2542 | 0.4788 | 0.5012 | 0.0011 | 0.2952 |
| IBNorm-S** | 0.2587 | 0.2570 | 0.2535 | 0.4750 | 0.4822 | 0.0023 | 0.2881 |
| IBNorm-T* | 0.2518 | 0.2658 | 0.2570 | 0.4879 | 0.4988 | 0.0004 | 0.2936 |
| IBNorm-T** | 0.2261 | 0.2562 | 0.2472 | 0.4740 | 0.4941 | 0.0008 | 0.2831 |
| IBNorm-L* | 0.2483 | 0.2615 | 0.2389 | 0.5029 | 0.4886 | 0.0030 | 0.2905 |
| IBNorm-L** | 0.2435 | 0.2606 | 0.2407 | 0.4743 | 0.4957 | 0.0008 | 0.2859 |

### D.3 ABLATION STUDY ON $\lambda$

Table 11: Ablation study of Llama 60M with normalization using different $\lambda$ evaluated on LLM Leaderboard I.

| Normalization | $\lambda$ | ARC | HellaSwag | MMLU | TruthfulQA | Winogrande | GSM8K | AVG |
|---|---|---|---|---|---|---|---|---|
| IBNorm-S | 0.5 | 0.2227 | 0.2641 | 0.2478 | 0.4948 | 0.5046 | 0.0045 | 0.2898 |
| | 4 | 0.2393 | 0.2665 | 0.2496 | 0.5033 | 0.5078 | 0.0049 | **0.2952** |
| | 8 | 0.2270 | 0.2647 | 0.2579 | 0.4984 | 0.4958 | 0.0072 | 0.2918 |
| IBNorm-L | 0.5 | 0.2193 | 0.2781 | 0.2501 | 0.4892 | 0.4767 | 0.0045 | 0.2863 |
| | 4 | 0.2568 | 0.2651 | 0.2689 | 0.4966 | 0.4854 | 0.0000 | **0.2955** |
| | 8 | 0.2722 | 0.2692 | 0.2293 | 0.4853 | 0.4964 | 0.0000 | 0.2921 |
| IBNorm-T | 0.5 | 0.2243 | 0.2644 | 0.2692 | 0.4846 | 0.4862 | 0.0000 | 0.2881 |
| | 4 | 0.2270 | 0.2758 | 0.2620 | 0.4827 | 0.5099 | 0.0049 | **0.2937** |
| | 8 | 0.2274 | 0.2645 | 0.2519 | 0.4923 | 0.5028 | 0.0000 | 0.2898 |

Table 12: Ablation study of GPT-2 124M with normalization using different $\lambda$ evaluated on LLM Leaderboard I.

| Normalization | $\lambda$ | ARC | HellaSwag | MMLU | TruthfulQA | Winogrande | GSM8K | AVG |
|---|---|---|---|---|---|---|---|---|
| IBNorm-T | 0.5 | 0.2652 | 0.2634 | 0.2512 | 0.4896 | 0.5092 | 0.0000 | 0.2964 |
| | 4 | 0.2819 | 0.2693 | 0.2532 | 0.4972 | 0.5185 | 0.0011 | **0.3035** |
| | 8 | 0.2794 | 0.2660 | 0.2522 | 0.4903 | 0.5037 | 0.0000 | 0.2986 |

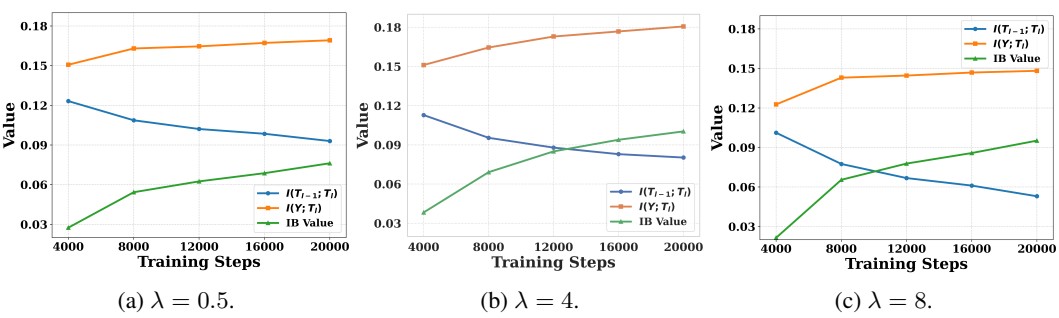

(a) $\lambda = 0.5$.       (b) $\lambda = 4$.       (c) $\lambda = 8$.

Figure 3: Evolution of predictive information $\hat{I}(Y; T_l)$, task–nuisance information $\hat{I}(T_{l-1}; T_l)$, and the token-level IB value during training of Llama-130M on C4, using IBNorm-T across different hyperparameters $\lambda = 0.5, 4, 8$. Results are reported on the test set.

As shown in Figure 3, we find: 1) Under-compression ($\lambda = 0.5$) fails to sufficiently suppress task-irrelevant information, leading to suboptimal IB tradeoffs. 2) Over-compression ($\lambda = 8$) starts to suppress task-relevant information, harming predictive performance. 3) Moderate compression ($\lambda = 4$) achieves the best IB value, effectively removing nuisance factors while preserving task-useful information.

### D.4 EXTENDED RESULTS ON IMAGENET

In addition to the results for ViT in Tab. 3, we conduct ViT experiments following the ViT-S/16 and ViT-B/16 settings (Yuan et al., 2021), with details provided in Appendix G.2.

Table 13: Accuracy (%) of ViT-S/16 and ViT-B/16 on image classification tasks. We present Layer-Norm's results on Vision Transformer (ViT) because it performs much better than BatchNorm.

| | ViT-S/16 on ImageNet | | ViT-B/16 on ImageNet | |
| Normalization | Top-1 | Top-5 | Top-1 | Top-5 |
|---|---|---|---|---|
| BatchNorm | — | — | — | — |
| LayerNorm | 78.25 | 94.12 | 81.09 | 95.57 |
| RMSNorm | 78.19 | 93.98 | 81.06 | 95.50 |
| NormalNorm | 78.77 | 94.21 | 81.03 | 95.46 |
| IBNorm-S | 80.23 | 94.94 | 82.53 | 95.97 |
| IBNorm-L | 80.57 | 95.12 | 83.69 | 96.43 |
| IBNorm-T | **80.76** | **95.21** | **83.71** | **96.49** |

Table 14: Accuracy (%) on CIFAR-10 with ResNet18. We report BatchNorm's results since it significantly outperforms LayerNorm on CNNs.

| Normalization | Top-1 | Top-5 |
|---|---|---|
| BatchNorm | 93.65 | 99.85 |
| IterNorm | 94.48 | 99.86 |
| LayerNorm | — | — |
| RMSNorm | — | — |
| NormalNorm | 94.01 | 99.86 |
| IBNorm-S | 95.49 | 99.85 |
| IBNorm-L | **95.57** | **99.87** |
| IBNorm-T | 95.51 | 99.83 |

## D.5 THE EFFECT OF COMPRESSION OPERATION IN IBNORM

We instantiate three representative forms of the compression function $f_\lambda$: linear (IBNorm-S), logarithmic (IBNorm-L), and hyperbolic tangent (IBNorm-T).

$$f_\lambda(|x_i - \mu|) = |x_i - \mu|/\lambda, \qquad \text{(\textbf{IBNorm-S}, linear compression)}$$
$$f_\lambda(|x_i - \mu|) = \ln(1 + |x_i - \mu|/\lambda), \qquad \text{(\textbf{IBNorm-L}, logarithmic compression)}$$
$$f_\lambda(|x_i - \mu|) = \tanh(|x_i - \mu|/\lambda). \qquad \text{(\textbf{IBNorm-T}, hyperbolic tangent compression)}$$

These choices capture complementary compression behaviors. Specifically, IB-S provides a linear baseline, IB-L yields progressively stronger suppression of large activations, and IB-T enforces a hard saturation effect on extreme values. Together, they span the spectrum from mild to strong tail suppression while sharing the desirable properties of boundedness and monotonicity. While other monotonic compression functions are possible, we provide these three forms to demonstrate the range of behaviors and provide a principled comparison.

The bounded compression property requires

$$0 \le f_\lambda(r) \le \alpha_\lambda r, \quad \forall r \ge 0, \quad \alpha_\lambda \in [0, 1].$$

To determine $\alpha_\lambda$, we analyze the ratio $f_\lambda(r)/r$.

**Linear.** For $f_\lambda(r) = r/\lambda$,

$$\frac{f_\lambda(r)}{r} = \frac{1}{\lambda}, \quad \forall r > 0,$$

thus $\alpha_\lambda = 1/\lambda$.

**Logarithmic.** For $f_\lambda(r) = \log(1 + r/\lambda)$,

$$\frac{f_\lambda(r)}{r} = \frac{\log(1 + r/\lambda)}{r},$$

which is monotonically decreasing in $r$ and attains its maximum at $r \to 0$:

$$\lim_{r \to 0} \frac{\log(1 + r/\lambda)}{r} = \frac{1}{\lambda}.$$

Hence $\alpha_\lambda = 1/\lambda$.

**Hyperbolic tangent.** For $f_\lambda(r) = \tanh(r/\lambda)$, the ratio

$$\frac{f_\lambda(r)}{r}$$

is maximized as $r \to 0$, giving

$$\lim_{r \to 0} \frac{\tanh(r/\lambda)}{r} = \frac{1}{\lambda}.$$

Thus $\alpha_\lambda = 1/\lambda$.

In conclusion, for these functionals, the compression ratio $\alpha_\lambda$ in Eqn. (7) is given by $\alpha_\lambda = 1/\lambda$, which guarantees the bounded compression property whenever $\lambda \geq 1$.

Compression operations in IBNorm compress the tail of activations while adjusting higher-order statistics. IBNorm-L, IBNorm-T, and IBNorm-S exhibit different abilities to compress the tails of activations.

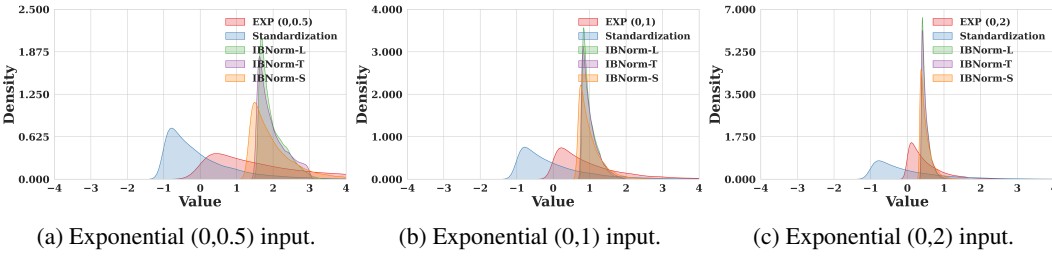

(a) Exponential (0,0.5) input.      (b) Exponential (0,1) input.      (c) Exponential (0,2) input.

Figure 4: Comparison of kernel density estimation across different compression operation (Standardization, IBNorm-L ($\lambda = 4$), IBNorm-T ($\lambda = 4$), IBNorm-S ($\lambda = 3$)) given Exponential distribution inputs with mean 0 and different lambdas. Compression operations in IBNorm compress the tail in the activations and adjust the higher-order statistics.

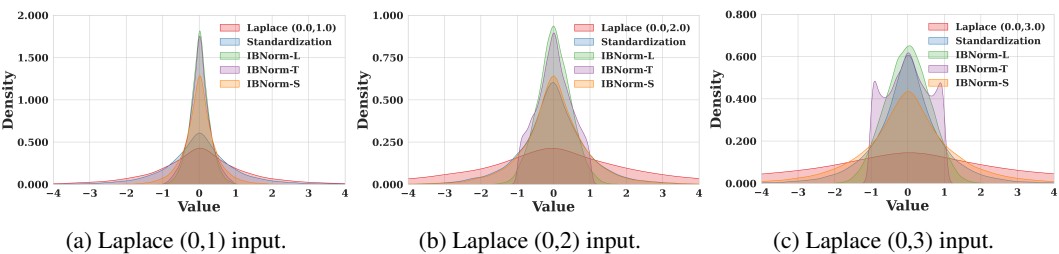

(a) Laplace (0,1) input.      (b) Laplace (0,2) input.      (c) Laplace (0,3) input.

Figure 5: Comparison of kernel density estimation across different compression operation (Standardization, IBNorm-L ($\lambda = 4$), IBNorm-T ($\lambda = 4$), IBNorm-S ($\lambda = 3$)) given Laplace distribution inputs with mean 0 and different scales. Compression operations in IBNorm compress the tail in the activations and adjust the higher-order statistics.

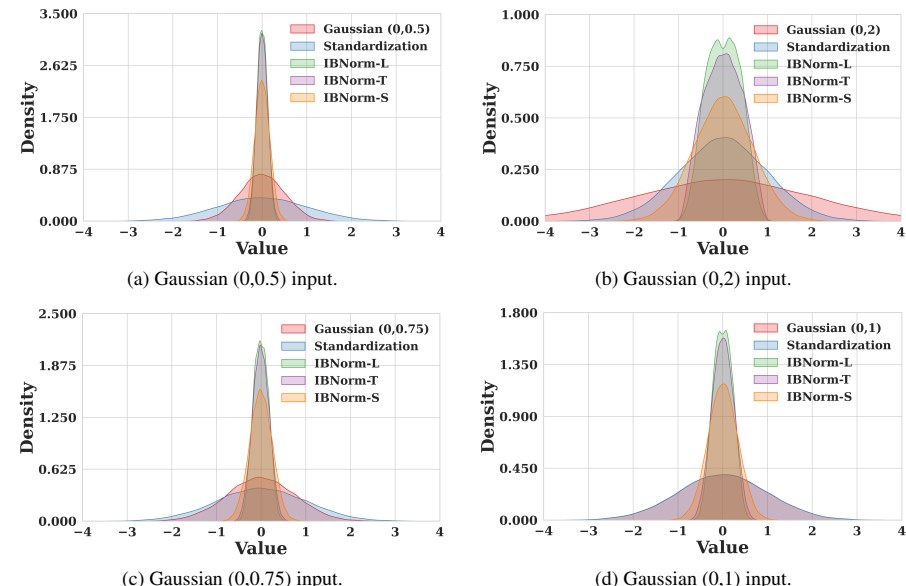

(a) Gaussian (0,0.5) input.

(b) Gaussian (0,2) input.

(c) Gaussian (0,0.75) input.

(d) Gaussian (0,1) input.

Figure 6: Comparison of kernel density estimation across different compression operation (Standardization, IBNorm-L ($\lambda = 4$), IBNorm-T ($\lambda = 4$), IBNorm-S ($\lambda = 3$)) given Gaussian distribution inputs with mean 0 and different variance. Compression operations in IBNorm compress the tail in the activations and adjust the higher-order statistics.

**Distinction of IBNorm-S, IBNorm-L, and IBNorm-T** IBNorm-S, IBNorm-L, and IBNorm-T are three types of IBNorm, which satisfy the compression property. Compression operations in IBNorm compress the tail of activations while adjusting higher-order statistics. IBNorm-L, IBNorm-T, and IBNorm-S exhibit different abilities to compress the tails of activations. These differences manifest in the induced entropy of the output distribution and in how aggressively each function compresses activation tails. Because all three satisfy IB constraints, their IB curves (predictive MI, nuisance MI, token-level IB value) follow similar trends during training, making it difficult to distinguish their own characteristics purely from the IB framework. For this reason, we provide an additional quantitative analysis using kernel density entropy, which directly reflects how much information each compression function filters.

Specifically, to illustrate the distinct compression behaviors of IBNorm-S, IBNorm-L, and IBNorm-T, we compute the entropy of the kernel density estimate (KDE) of activations after applying each variant (with the same hyperparameter $\lambda = 4$ under different input distributions. For an input Gaussian $\mathcal{N}(0, 2)$ with entropy 1.768, the resulting KDE entropies are 1.695 for IBNorm-S, 1.530 for IBNorm-L, and 1.302 for IBNorm-T. Similarly, for an input Gaussian $\mathcal{N}(0, 1)$ with entropy 1.418, the corresponding output entropies are 1.352 for IBNorm-S, 1.263 for IBNorm-L, and 1.192 for IBNorm-T. These results indicate that IBNorm-S performs the mildest compression, IBNorm-L achieves stronger compression, and IBNorm-T is the most aggressive in reducing entropy. This analysis highlights the distinct statistical effects of the three variants on latent representations.

Beyond compression strength, we observe differences in robustness. As shown in Tables 4 and 11, IBNorm-S is less sensitive to variations in the hyperparameter $\lambda$ compared to IBNorm-L and IBNorm-T. Similarly, ablation studies on the normalization structure shown in Tables 5 and 10 indicate that IBNorm-S is more robust to the ordering of the IB compression and standardization operations.

At the task level, performance differences align with these compression characteristics. Across diverse medium-scale LLMs (Llama 130M–1B and GPT 350M), IBNorm-S generally achieves the worst performance among the three IBNorm variants on evaluation tasks in Tables 1 and 6, reflecting its mildest compression. For challenging tasks, such as BBH and GPQA, the stronger compression applied by IBNorm-L can lead to improved performance, suggesting that mild compression can better support reasoning and generalization in difficult tasks.

### D.6 ESTIMATED MUTUAL INFORMATION QUANTITIES AND TOKEN-LEVEL IB VALUE ACROSS TRAINING

We track the evolution of three mutual information quantities during training: predictive information $\hat{I}(Y; T_l)$, task–nuisance information $\hat{I}(T_{l-1}; T_l)$, and their token-level IB value $\hat{I}(Y; T_l) - \hat{I}(T_{l-1}; T_l)$. These quantities are measured throughout training on test dataset for Llama-130M and GPT-2 (124M) on C4 and OpenWebText, respectively. The details in token-level IB value calculation are in Appendix H.

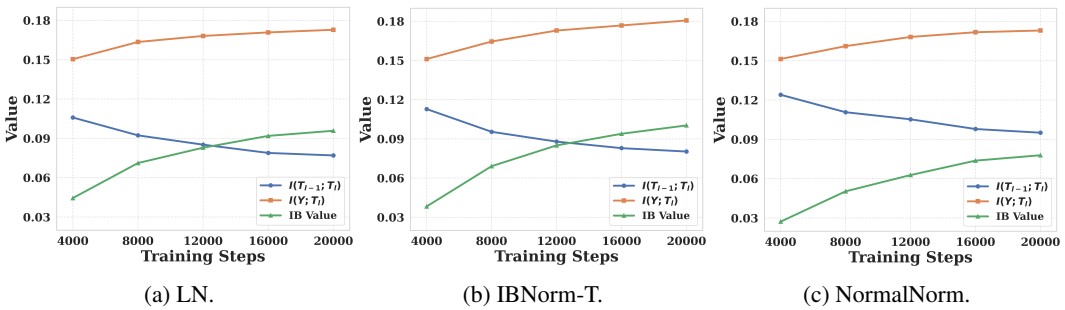

(a) LN.     (b) IBNorm-T.     (c) NormalNorm.

Figure 7: Evolution of predictive information $\hat{I}(Y; T_l)$, task–nuisance information $\hat{I}(T_{l-1}; T_l)$, and the token-level IB value during training of Llama-130M on C4, using LN, IBNorm-T, and Normal-Norm. Results are reported on the test set.

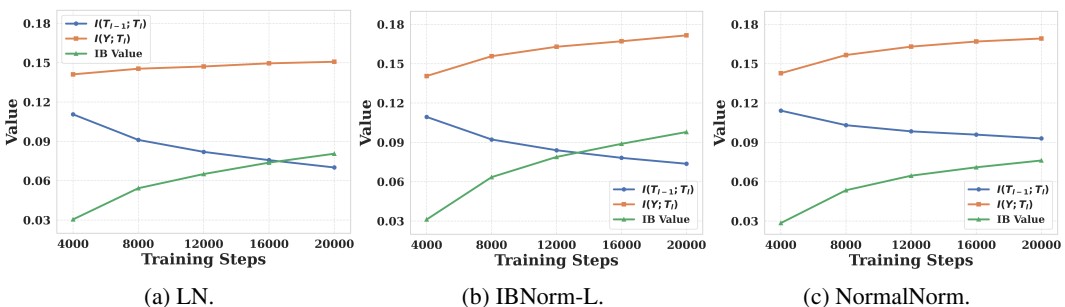

(a) LN.     (b) IBNorm-L.     (c) NormalNorm.

Figure 8: Evolution of predictive information $\hat{I}(Y; T_l)$, task–nuisance information $\hat{I}(T_{l-1}; T_l)$, and the token-level IB value during training of GPT-2 (124M) on OpenWebText, using LN, IBNorm-L, and NormalNorm. Results are reported on the test set.

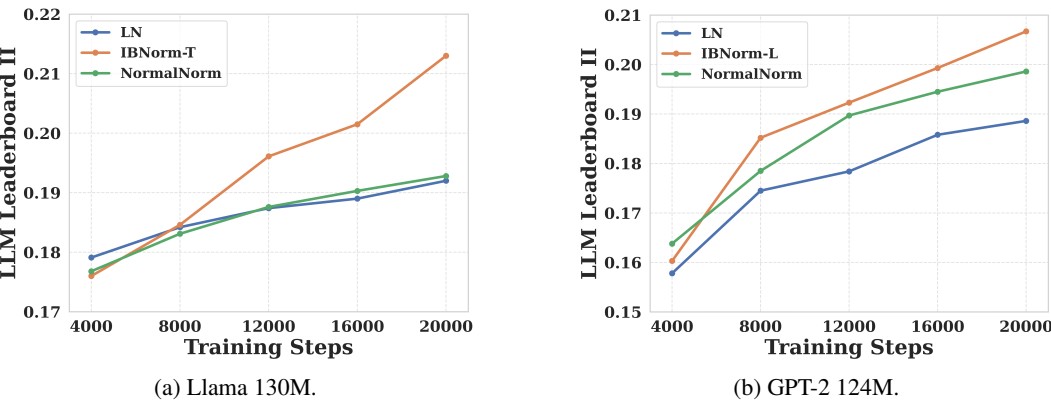

(a) Llama 130M.     (b) GPT-2 124M.

Figure 9: Evaluation of Llama 130M and GPT-2 124M trained with LN, IBNorm-L, NormalNorm on LLM Leaderboard II across training.

As shown in Figs. 7, 8, and 9, these results demonstrate that IBNorm encourages representations that retain task-relevant information while suppressing irrelevant task-nuisance information, providing a explanation for the observed empirical gains.

### D.7 DISCUSSION ON TRAINING TIME AND COMPUTATIONAL OVERHEAD

IBNorm introduces only a single compression operation per normalization layer. Consequently, it does not incur significant computational overhead or cause numerical stability issues, even for larger models. Training time and memory usage of IBNorm remain comparable to standard variance-centric normalization methods such as LayerNorm or RMSNorm.

We report the training time and total VRAM usage for LLaMA-60M under the same global batch size 512 and local batch size 64, trained on $4\times$ L40-S GPUs with 20000 training steps:

Table 15: Training time and VRAM usage for different normalization methods on LLaMA-60M.

| Normalization | Training Time | Total VRAM |
|---|---|---|
| LayerNorm | 1h 54min | 95,944 MiB |
| RMSNorm | 2h 15min | 102,601 MiB |
| NormalNorm | 3h 38min | 139,207 MiB |
| IBNorm-S | 2h 14min | 102,460 MiB |
| IBNorm-L | 2h 17min | 106,371 MiB |
| IBNorm-T | 2h 14min | 104,853 MiB |

We report the training time and total VRAM usage for LLaMA-1B under the same global batch size 512 and local batch size 8, trained on $4\times$ L40-S GPUs with 100000 training steps:

Table 16: Training time and VRAM usage for different normalization methods on LLaMA-1B.

| Normalization | Training Time | Total VRAM |
|---|---|---|
| LayerNorm | 176h 02min | 99,832 MiB |
| RMSNorm | 190h 58min | 110,574 MiB |
| NormalNorm | 246h 06min | 162,088 MiB |
| IBNorm-S | 194h 29min | 115,920 MiB |
| IBNorm-L | 190h 02min | 106,484 MiB |
| IBNorm-T | 188h 41min | 106,481 MiB |

## E DETAILS OF OTHER TERMS

We have
$$G(\sigma_{H_1|X}, \sigma_{H_1|Y}, \sigma_{p_1|X}, \sigma_{p_1|Y}, \mu_{p_1|X}, \mu_{p_1|Y})$$

$$= C_1\sqrt{\frac{\sigma_{H_1|X}}{M}} + C_1\sqrt{\frac{\sigma_{H_1|Y}}{M}} + \sum_y p(y) \int_{\mathcal{T}_1} \phi(\sqrt{\sigma_{p_1|Y}})\, dt + \beta \sum_x p(x) \int_{\tilde{\mathcal{T}}_1} \phi(\sqrt{\sigma_{p_1|X}})\, dt$$

$$+ (\beta + 2) \int_{\tilde{\mathcal{T}}_1} \phi\left(C_1\sqrt{\frac{\sigma_{H_1|X}}{M}}\right) p_{\tilde{T}_1}(t)\, dt + \sum_y p(y) \int_{\tilde{\mathcal{T}}_1} \phi(\mu_{p_1|Y})\, dt + \beta \sum_x p(x) \int_{\tilde{\mathcal{T}}_1} \phi(\mu_{p_1|X})\, dt$$

$$+ (\beta + 1) \int_{\tilde{\mathcal{T}}_1} \sum_x p(x)[\phi(\sqrt{\sigma_{p_1|X}}) + \phi(\mu_{p_1|X})]\, dt,$$

where
$$\sigma_{H_1|X} := V(H(\tilde{T}_1|X)) = |\mathcal{X}|\mathbb{V}(H(\tilde{T}_1|X)), \quad \sigma_{H_1|Y} := V(H(\tilde{T}_1|Y)) = |\mathcal{Y}|\mathbb{V}(H(\tilde{T}_1|Y)),$$
$$\sigma_{p_1|X} := V(p(\tilde{T}_1|X)) = |\mathcal{X}|\mathbb{V}(p(\tilde{T}_1|X)), \quad \sigma_{p_1|Y} := V(p(\tilde{T}_1|Y)) = |\mathcal{Y}|\mathbb{V}(p(\tilde{T}_1|Y)),$$
$$\mu_{p_1|X} := \mathbb{E}[p(\tilde{T}_1|X)], \qquad\qquad \mu_{p_1|Y} := \mathbb{E}[p(\tilde{T}_1|Y)],$$

and

$$C_1 = 2 + \sqrt{2\log((|\mathcal{Y}| + 2)/\delta)}.$$

In addition, we have the optimum–dependent term:

$$Q(\tilde{T}_1^*) = (2 + \sqrt{2\log((|\mathcal{Y}| + 2)/\delta)})\sqrt{\frac{V(H(\tilde{T}_1^*|Y))}{M}}$$

$$+ \beta(2 + \sqrt{2\log((|\mathcal{Y}| + 2)/\delta)})\sqrt{\frac{V(H(\tilde{T}_1^*|X))}{M}}$$

$$+ (\beta + 2)\int_{\tilde{\mathcal{T}}_1^*} \phi\left((2 + \sqrt{2\log((|\mathcal{Y}| + 2)/\delta)})\sqrt{\frac{V(p(\tilde{T}_1^*|X))}{M}}\right) p_{\tilde{T}_1^*}(t)\, dt$$

$$+ \sum_y p(y)\int_{\tilde{\mathcal{T}}_1} \phi(p_{\tilde{T}_1^*|Y}(t|y))\, dt + \beta\sum_x p(x)\int_{\tilde{\mathcal{T}}_1} \phi(p_{\tilde{T}_1^*|X}(t|x))\, dt + (\beta + 1)\int_{\tilde{\mathcal{T}}_1} \phi(p(\tilde{T}_1^*))\, dt$$

$$+ (\beta + 1)\int_{\tilde{\mathcal{T}}_1} \sum_x p(x)\phi(p_{\tilde{T}_1^*|X}(t|x))\, dt$$

# F  PROOF

## F.1  PROOF FOR THEOREM 1

**Theorem.** *For any hyperparameter $\beta \in [0, 1]$ and the sample set $S \sim \mathbb{P}(X, Y)$ of size $M$, we have*

$$\widehat{IB}_S(\tilde{T}_{IB}) \geq \widehat{IB}_S(\tilde{T}_s) \quad \text{almost surely}.$$

*Proof.* Given the dataset $S \sim \mathbb{P}_{X,Y}$ with size $M$ and one-layer neural networks, $f_1 = \Phi_1 \circ h$. We analyze the IB value of intermediate features $\tilde{T}_1 := \psi \circ \zeta \circ h(X)$. First, we define the empirical gap $\Delta_S(\tilde{T}_1; \tilde{T}_1^*)$ between the IB bottleneck derived from arbitrary representation $\tilde{T}_1$ and the optimum representation $\tilde{T}_1^*$, *i.e.*

$$\Delta_S(\tilde{T}_1; \tilde{T}_1^*) = |\left(\hat{I}(Y; \tilde{T}_1) - \beta\hat{I}(\tilde{T}_0; \tilde{T}_1)\right) - \left(\hat{I}(Y; \tilde{T}_1^*) - \beta\hat{I}(\tilde{T}_0^*; \tilde{T}_1^*)\right)|, \tag{15}$$

where $\tilde{T}_0 = \tilde{T}_0^* = X$.

We prove $\widehat{IB}_1 \geq \widehat{IB}_2$ by showing that $\Delta_S(\tilde{T}_{IB}; \tilde{T}_1^*) \leq \Delta_S(\tilde{T}_s; \tilde{T}_1^*)$.

**Lemma 3.** *Given a fixed trade-off hyperparameter $\beta > 0$ and a confidence parameter $\delta \in (0, 1)$, the following holds with probability at least $1 - \delta$ over any sample set $S \sim \mathbb{P}(X, Y)$ with size $M$: for any representation $\tilde{T}_1 \in \tilde{\mathcal{T}}_1$,*

$$\Delta_S(\tilde{T}_1; \tilde{T}_1^*) \leq G(\sigma_{H_1|X}, \sigma_{H_1|Y}, \sigma_{p_1|X}, \sigma_{p_1|Y}, \mu_{p_1|X}, \mu_{p_1|Y}) + Q(T_1^*). \tag{16}$$

*where $G(\cdot)$ is a monotonically increasing function with its arguments. Specifically, $\sigma_{H_1|X} := \mathbb{V}(H(\tilde{T}_1|X)), \sigma_{H_1|Y} := \mathbb{V}(H(\tilde{T}_1|Y)), \sigma_{p_1|X} := \mathbb{V}(p(\tilde{T}_1|X)), \sigma_{p_1|Y} := \mathbb{V}(p(\tilde{T}_1|Y)), \mu_{p_1|X} := \mathbb{E}[p(\tilde{T}_1|X)], \mu_{p_1|Y} := \mathbb{E}[p(\tilde{T}_1|Y)]$ and $\mathbb{V}$ is the variance function.*

*The formulas of concrete form of $G$ and $Q$ are given in Appendix E. The proof is given in Appendix F.1.*

We now analyze the gap $\Delta_S(\tilde{T}_{IB}; \tilde{T}_1^*)$ and $\Delta_S(\tilde{T}_s; \tilde{T}_1^*)$ by analyzing the $\tilde{T}_1$ relevant terms. We first note that we have the following property for the IBNorm.

**Proposition 4** (Entropy Reduction). *For any random vector $Z \in \mathbb{R}^d$ with distribution $\mathbb{P}_Z$, the compression operation $s_\lambda$ in IBNorm satisfies*

$$H(s_\lambda(Z)) \leq H(Z), \tag{17}$$

*where $H(\cdot)$ denotes differential entropy. The detailed proof is given in Appendix F.3.*

Based on Proposition 4, we have the following lemma.

**Lemma 5.** *1) Let $\tilde{T}_{IB} = s_\lambda(\tilde{T}_s)$, where $s_\lambda$ is a bounded-compression mapping with Lipschitz constant $L_1 = \prod_{i=1}^{d} \frac{1}{\lambda}$. For any small $\delta > 0$, the following holds:*

$$\mathbb{P}(\mathbb{V}(H(\tilde{T}_{IB}|X)) \leq \mathbb{V}(H(\tilde{T}_s|X))) \geq 1 - \delta, \quad \mathbb{P}(\mathbb{V}(H(\tilde{T}_{IB}|Y)) \leq \mathbb{V}(H(\tilde{T}_s|Y))) \geq 1 - \delta.$$

*2) In addition, let $\tilde{T}_1, \tilde{T}_1' \in \tilde{\mathcal{T}}_1$ be two arbitrary elements in the representation space. Then, for any $\delta \in (0, 1)$, the following holds with probability at least $1 - \delta$:*

$$|\mathbb{V}(p(\tilde{T}_1|X)) - \mathbb{V}(p(\tilde{T}_1'|X))| \leq \delta,$$
$$|\mathbb{V}(p(\tilde{T}_1|Y)) - \mathbb{V}(p(\tilde{T}_1'|Y))| \leq \delta,$$
$$|\mathbb{E}[p(\tilde{T}_1|X)] - \mathbb{E}[p(\tilde{T}_1'|X)]| \leq \delta,$$
$$|\mathbb{E}[p(\tilde{T}_1|Y)] - \mathbb{E}[p(\tilde{T}_1'|Y)]| \leq \delta,$$

*where the expectation and variance are taken with respect to the Lebesgue measure on $\mathcal{X} \times \mathcal{Y}$.*

*The detailed proof is given in Appendix F.4.*

Since $G(\cdot)$ is monotone in its variance arguments, Lemma 5 implies that, for a fixed trade-off hyperparameter $\beta \in [0, 1]$ and confidence parameter $\delta \in (0, 1)$, with probability at least $1 - \delta$ over the sample set $S \sim \mathbb{P}(X, Y)$ with size $M$, the following inequality holds:

$$\Delta_S(\tilde{T}_{IB}; \tilde{T}_1^*) \leq \Delta_S(\tilde{T}_s; \tilde{T}_1^*).$$

Thus, given the hyperparameter $\beta$ and the sample set $S \sim \mathbb{P}(X, Y)$ with size $M$, we have

$$\widehat{\text{IB}}_S(\tilde{T}_{IB}) \geq \widehat{\text{IB}}_S(\tilde{T}_s) \quad \text{almost surely .}$$

$\square$

## F.2 PROOF FOR LEMMA 3

**Lemma.** *Given a fixed trade-off hyperparameter $\beta > 0$ and a confidence parameter $\delta \in (0, 1)$, the following holds with probability at least $1 - \delta$ over the sample set $S$: for any representation $\tilde{T}_1 \in \tilde{\mathcal{T}}_1$,*

$$\Delta_S(\tilde{T}_1; \tilde{T}_1^*) \leq G(\sigma_{H_1|X}, \sigma_{H_1|Y}, \sigma_{p_1|X}, \sigma_{p_1|Y}, \mu_{p_1|X}, \mu_{p_1|Y}) + Q(T_1^*). \tag{18}$$

*where $G(\cdot)$ is a monotonically increasing function with its arguments. Specifically, $\sigma_{H_1|X} := \mathbb{V}(H(\tilde{T}_1|X)), \sigma_{H_1|Y} := \mathbb{V}(H(\tilde{T}_1|Y)), \sigma_{p_1|X} := \mathbb{V}(p(\tilde{T}_1|X)), \sigma_{p_1|Y} := \mathbb{V}(p(\tilde{T}_1|Y)), \mu_{p_1|X} := \mathbb{E}[p(\tilde{T}_1|X)], \mu_{p_1|Y} := \mathbb{E}[p(\tilde{T}_1|Y)]$ and $\mathbb{V}$ is the variance function.*

*The formulas of concrete form of $G$ and $Q$ are given in Appendix E.*

*Proof.* Let $\tilde{T}_1$ be the embeddings derived from a neural network based on the sample set $S$.

We define the population IB value as $\text{IB}(T) := I(Y; T) - \beta I(X; T)$ and empirical IB value as $\widehat{\text{IB}}(T) := \hat{I}(Y; T) - \beta \hat{I}(X; T)$ for brevity. We denote the optimum IB representation achieving the maximum population IB value by $\tilde{T}_1^*$.

We define the optimum $\tilde{T}_1^*$ of one-layer IB objective as:

$$\tilde{T}_1^* \in \arg\max_{\tilde{T}_1} (I(Y; \tilde{T}_1) - \beta I(\tilde{T}_0; \tilde{T}_1)), \tag{19}$$

where $\tilde{T}_0 = X$.

Given the dataset $S \sim \mathbb{P}_{X,Y}$ with size $M$, we consider the empirical gap $\Delta_S(\tilde{T}_1; \tilde{T}_1^*)$ between the IB bottleneck derived from arbitrary representation $\tilde{T}_1$ and the optimum representation $\tilde{T}_1^*$, *i.e.*

$$\Delta_S(\tilde{T}_1; \tilde{T}_1^*) = |\left(\hat{I}(Y; \tilde{T}_1) - \beta \hat{I}(\tilde{T}_0; \tilde{T}_1)\right) - \left(\hat{I}(Y; \tilde{T}_1^*) - \beta \hat{I}(\tilde{T}_0^*; \tilde{T}_1^*)\right)|, \tag{20}$$

where $\tilde{T}_0 = \tilde{T}_0^* = X$. We have:

$$\Delta_S(\tilde{T}_1; \tilde{T}_1^*) = \left| \left( \hat{I}(Y; \tilde{T}_1) - \beta \hat{I}(X; \tilde{T}_1) \right) - \left( \hat{I}(Y; \tilde{T}_1^*) - \beta \hat{I}(X; \tilde{T}_1^*) \right) \right| \tag{21}$$

$$\leq \underbrace{\left| \left( \hat{I}(Y; \tilde{T}_1) - \hat{I}(Y; \tilde{T}_1^*) \right) \right|}_{I} + \beta \underbrace{\left| \left( \hat{I}(X; \tilde{T}_1) - \hat{I}(X; \tilde{T}_1^*) \right) \right|}_{II} \tag{22}$$

For any real-valued vector $\mathbf{a} = (a_1, \cdots, a_n)$, we define the function $V(\mathbf{a}) = \sum_{i=1}^n (a_i - \frac{1}{n} \sum_{j=1}^n a_j)^2$. Note that $\frac{1}{n} V(\mathbf{a})$ is simply the variance of the elements of $\mathbf{a}$.

In addition, we define an auxiliary real-valued function

$$\phi(x) = \begin{cases} -\frac{1}{e}, & x < -\frac{1}{e}, \\ -x \log \frac{1}{-x}, & -\frac{1}{e} < x \leq 0, \\ 0, & x = 0, \\ x \log \frac{1}{x}, & 0 < x \leq \frac{1}{e}, \\ \frac{1}{e}, & x > \frac{1}{e}. \end{cases}$$

Note that $\phi$ is a continuous, and concave function, and that $\lim_{x \to 0} \phi(x) = 0$. In addition, we have the following properties for the auxiliary function $\phi(\cdot)$ (Shamir et al., 2010):

**Lemma 6.** *For any $a, b \in [0, 1]$, $|a \log(a) - b \log(b)| \leq \phi(a - b)$.*

**Lemma 7.** *For any real numbers $a$ and $b$, we have $\phi(a + b) \leq \phi(|a|) + \phi(|b|)$.*

**I. Bound on** $\left| \left( \hat{I}(Y; \tilde{T}_1) - \hat{I}(Y; \tilde{T}_1^*) \right) \right|$.

Without loss of generality and reduce the complexity, we denote $T \equiv \tilde{T}_1$ and $T^* \equiv \tilde{T}_1^*$. By the triangle inequality, we have:

$$|\hat{I}(Y; T) - \hat{I}(Y; T^*)| \leq \underbrace{|I(Y; T) - \hat{I}(Y; T)|}_{A} + \underbrace{|I(Y; T^*) - \hat{I}(Y; T^*)|}_{B} + \underbrace{|I(Y; T) - I(Y; T^*)|}_{C}.$$

To bound the terms $A'$ and $B'$, we extend the results of Section 6.2 in (Shamir et al., 2010) to derive a general bound $|I(Y; T) - \hat{I}(Y; T)|$ under the setting where $T$ is a continuous random variable. Firstly, using the relationship $I(Y; T) = H(T) - H(T|Y)$, we apply the triangle inequality:

$$|I(Y; T) - \hat{I}(Y; T)| \leq |H(T) - \hat{H}(T)| + |H(T|Y) - \hat{H}(T|Y)| \tag{23}$$

STEP 1: BOUNDING $|H(T) - \hat{H}(T)|$

By Lemma 6, for any two densities $p(t)$ and $\hat{p}(t)$:

$$|H(T) - \hat{H}(T)| = \left| \int_{\mathcal{T}} p(t) \log p(t) \, dt - \int_{\mathcal{T}} \hat{p}(t) \log \hat{p}(t) \, dt \right| \leq \int_{\mathcal{T}} \phi(|p(t) - \hat{p}(t)|) \, dt \tag{24}$$

The difference in densities is:

$$|p(t) - \hat{p}(t)| = \left| \sum_x p(t|x)(p(x) - \hat{p}(x)) \right| \tag{25}$$

Using the Cauchy-Schwarz inequality and the definition of $V(p(t|x))$:

$$|p(t) - \hat{p}(t)| \leq ||p(x) - \hat{p}(x)||_2 \sqrt{V(p(T = t|x))} \tag{26}$$

Substituting this back into the integral:

$$|H(T) - \hat{H}(T)| \leq \int_{\mathcal{T}} \phi \left( ||p(x) - \hat{p}(x)||_2 \sqrt{V(p(T = t|x))} \right) \, dt \tag{27}$$

STEP 2: BOUNDING $|H(T|Y) - \hat{H}(T|Y)|$

We decompose the error into two terms:

$$|H(T|Y) - \hat{H}(T|Y)| \leq \left| \sum_y p(y)(H(T|y) - \hat{H}(T|y)) \right| + \left| \sum_y (p(y) - \hat{p}(y))\hat{H}(T|y) \right| \quad (28)$$

We have the following bound for the first term:

$$\left| \sum_y p(y)(H(T|y) - \hat{H}(T|y)) \right| \leq \sum_y p(y) \int_{\mathcal{T}} \phi(|p(t|y) - \hat{p}(t|y)|) \, dt \quad (29)$$

$$\leq \sum_y p(y) \int_{\mathcal{T}} \phi\left( ||\hat{p}(x|y) - p(x|y)||_2 \sqrt{V(p(T=t|x))} \right) dt \quad (30)$$

We have the following bound for the second term:

$$\left| \sum_y (p(y) - \hat{p}(y))\hat{H}(T|y) \right| \leq ||p(y) - \hat{p}(y)||_2 \sqrt{V(\hat{H}(T|Y))} \quad (31)$$

STEP 3: FINAL COMBINATION

Combining the bounds for $H(T)$ and $H(T|Y)$ in Equations 27 and 31 and extending the results in Section 6 (Shamir et al., 2010), we have:

$$|I(Y;T) - \hat{I}(Y;T)| \leq (2 + \sqrt{2\log((|\mathcal{Y}| + 2)/\delta)}) \sqrt{\frac{V(\hat{H}(T|Y))}{M}}$$

$$+ 2 \int_{\mathcal{T}} \phi\left( (2 + \sqrt{2\log((|\mathcal{Y}| + 2)/\delta)}) \sqrt{\frac{V(p(T|X))}{M}} \right) p_T(t) \, dt$$

$$\leq (2 + \sqrt{2\log((|\mathcal{Y}| + 2)/\delta)}) \sqrt{\frac{V(H(T|Y))}{M}}$$

$$+ 2 \int_{\mathcal{T}} \phi\left( (2 + \sqrt{2\log((|\mathcal{Y}| + 2)/\delta)}) \sqrt{\frac{V(p(T|X))}{M}} \right) p_T(t) \, dt \quad \text{(Law of Large Numbers)}$$

Without loss of generality, by applying the same bounding technique in Equations, we have:

A)

$$|I(Y;T) - \hat{I}(Y;T)| \leq (2 + \sqrt{2\log((|\mathcal{Y}| + 2)/\delta)}) \sqrt{\frac{V(\hat{H}(T|Y))}{M}}$$

$$+ 2 \int_{\mathcal{T}} \phi\left( (2 + \sqrt{2\log((|\mathcal{Y}| + 2)/\delta)}) \sqrt{\frac{V(p(T|X))}{M}} \right) p_T(t) \, dt$$

$$\leq (2 + \sqrt{2\log((|\mathcal{Y}| + 2)/\delta)}) \sqrt{\frac{V(H(T|Y))}{M}}$$

$$+ 2 \int_{\mathcal{T}} \phi\left( (2 + \sqrt{2\log((|\mathcal{Y}| + 2)/\delta)}) \sqrt{\frac{V(p(T|X))}{M}} \right) p_T(t) \, dt$$

B)

$$|I(Y;T^*) - \hat{I}(Y;T^*)| \leq (2 + \sqrt{2\log((|\mathcal{Y}| + 2)/\delta)})\sqrt{\frac{V(\hat{H}(T^*|Y))}{M}}$$

$$+ 2\int_{\mathcal{T}_*} \phi\left((2 + \sqrt{2\log((|\mathcal{Y}| + 2)/\delta)})\sqrt{\frac{V(p(T^*|X))}{M}}\right) p_{T^*}(t)\, dt$$

$$\leq (2 + \sqrt{2\log((|\mathcal{Y}| + 2)/\delta)})\sqrt{\frac{V(H(T^*|Y))}{M}}$$

$$+ 2\int_{\mathcal{T}_*} \phi\left((2 + \sqrt{2\log((|\mathcal{Y}| + 2)/\delta)})\sqrt{\frac{V(p(T^*|X))}{M}}\right) p_{T^*}(t)\, dt$$

For term C, we have

C)

$$|I(Y;T) - I(Y;T^*)| \leq |H(T|Y) - H(T^*|Y)| + |H(T) - H(T^*)|$$

i)

$$|H(T|Y) - H(T^*|Y)| = |\sum_y p(y)\big(H(T|y) - H(T^*|y)\big)|$$

$$= |\sum_y p(y)\int_{\mathcal{T}} p_{T|Y}(t|y)\log(p_{T|Y}(t|y))dt - \sum_y p(y)\int_{\mathcal{T}_*} p_{T^*|Y}(t|y)\log(p_{T^*|Y}(t|y))dt|$$

$$\leq \sum_y p(y)|\int_{\mathcal{T}} p_{T|Y}(t|y)\log(p_{T|Y}(t|y)) - p_{T^*|Y}(t|y)\log(p_{T^*|Y}(t|y))dt|$$

$$\leq \sum_y p(y)\int_{\mathcal{T}} |p_{T|Y}(t|y)\log(p_{T|Y}(t|y)) - p_{T^*|Y}(t|y)\log(p_{T^*|Y}(t|y))|dt$$

$$\leq \sum_y p(y)\int_{\mathcal{T}} \phi(p_{T|Y}(t|y) - p_{T^*|Y}(t|y))dt \quad \text{(By Lemma 6)}$$

$$= \sum_y p(y)\int_{\mathcal{T}} \phi(p_{T|Y}(t|y) - \mathbb{E}[p_{T|Y}(t|y)] + \mathbb{E}[p_{T|Y}(t|y)] - p_{T^*|Y}(t|y))dt$$

$$\leq \sum_y p(y)\int_{\mathcal{T}} \phi(|p_{T|Y}(t|y) - \mathbb{E}[p_{T|Y}(t|y)]|) + \phi(|\mathbb{E}[p_{T|Y}(t|y)]|) + \phi(|p_{T^*|Y}(t|y)|)dt \quad \text{(By Lemma 7)}$$

$$\leq \sum_y p(y)\int_{\mathcal{T}} \phi(\sqrt{V(p_{T|Y}(t|y))}) + \phi(\mathbb{E}[p_{T|Y}(t|y)]) + \phi(p_{T^*|Y}(t|y))dt$$

$$\tag{32}$$

ii)

$$|H(T) - H(T^*)| = |\int_{\mathcal{T}} p_T(t)\log(p_T(t))dt - \int_{\mathcal{T}_*} p_{T^*}(t)\log(p_{T^*}(t))dt|$$

$$\leq |\int_{\mathcal{T}} p_T(t)\log(p_T(t)) - p_{T^*}(t)\log(p_{T^*}(t))dt|$$

$$\leq \int_{\mathcal{T}} \sum_x p(x)|p_{T|X}(t|x)\log(p_{T|X}(t|x)) - p_{T^*|X}(t|x)\log(p_{T^*|X}(t|x))|dt \quad \text{(By Lemma 6)} \tag{33}$$

$$\leq \int_{\mathcal{T}} \sum_x p(x)\phi(p_{T|X}(t|x) - p_{T^*|X}(t|x))dt$$

$$\leq \int_{\mathcal{T}} \sum_x p(x)[\phi(\sqrt{V(p_{T|X}(t|x))}) + \phi(\mathbb{E}[p_{T|X}(t|x)]) + \phi(p_{T^*|X}(t|x))]dt$$

Combining i) and ii), we have

$$
|\hat{I}(Y;T) - \hat{I}(Y;T^*)| \leq \sum_y p(y) \int_{\mathcal{T}} \phi(\sqrt{V(p_{T|Y}(t|y))}) + \phi(\mathbb{E}[p_{T|Y}(t|y)]) + \phi(p_{T^*|Y}(t|y)) dt
$$
$$
+ \int_{\mathcal{T}} \sum_x p(x)[\phi(\sqrt{V(p_{T|X}(t|x))}) + \phi(\mathbb{E}[p_{T|X}(t|x)]) + \phi(p_{T^*|X}(t|x))] dt \tag{34}
$$

Thus, in our setting, we have:

$$
\left|\left(\hat{I}(Y;\tilde{T}_1) - \hat{I}(Y;\tilde{T}_1^*)\right)\right| \leq (2 + \sqrt{2\log((|\mathcal{Y}| + 2)/\delta)})\sqrt{\frac{V(H(\tilde{T}_1|Y))}{M}}
$$
$$
+ 2\int_{\tilde{T}_1} \phi\left((2 + \sqrt{2\log((|\mathcal{Y}| + 2)/\delta)})\sqrt{\frac{V(p(\tilde{T}_1|X))}{M}}\right) p_{\tilde{T}_1}(t)\, dt
$$
$$
+ (2 + \sqrt{2\log((|\mathcal{Y}| + 2)/\delta)})\sqrt{\frac{V(H(\tilde{T}_1^*|Y))}{M}}
$$
$$
+ 2\int_{\tilde{T}_1^*} \phi\left((2 + \sqrt{2\log((|\mathcal{Y}| + 2)/\delta)})\sqrt{\frac{V(p(\tilde{T}_1^*|X))}{M}}\right) p_{\tilde{T}_1^*}(t)\, dt \tag{35}
$$
$$
+ \sum_y p(y) \int_{\tilde{T}_1} \phi(\sqrt{V(p_{\tilde{T}_1|Y}(t|y))}) + \phi(\mathbb{E}[p_{\tilde{T}_1|Y}(t|y)]) + \phi(p_{\tilde{T}_1^*|Y}(t|y))\, dt
$$
$$
+ \int_{\tilde{T}_1} \sum_x p(x)[\phi(\sqrt{V(p_{\tilde{T}_1|X}(t|x))}) + \phi(\mathbb{E}[p_{\tilde{T}_1|X}(t|x)]) + \phi(p_{\tilde{T}_1^*|X}(t|x))] dt
$$

**II. Bound on** $\beta \left|\left(\hat{I}(X;\tilde{T}_1) - \hat{I}(X;\tilde{T}_1^*)\right)\right|$.

Without loss of generality and reduce the complexity, we denote $T \equiv \tilde{T}_1$ and $T^* \equiv \tilde{T}_1^*$. By the triangle inequality, we have:

$$
|\hat{I}(X;T) - \hat{I}(X;T^*)| \leq \underbrace{|I(X;T) - \hat{I}(X;T)|}_{A'} + \underbrace{|I(X;T^*) - \hat{I}(X;T^*)|}_{B'} + \underbrace{|I(X;T) - I(X;T^*)|}_{C'}.
$$

To bound the terms $A'$ and $B'$, we extend the results of Section 6.2 in (Shamir et al., 2010) to derive the corresponding bounds. Consequently, by applying the same bounding technique stated in I, we obtain:

A')
$$
|I(X;T) - \hat{I}(X;T)| \leq (2 + \sqrt{2\log((|\mathcal{Y}| + 2)/\delta)})\sqrt{\frac{V(H(T|X))}{M}}
$$
$$
+ \int_{\mathcal{T}} \phi\left((2 + \sqrt{2\log((|\mathcal{Y}| + 2)/\delta)})\sqrt{\frac{V(p(T|X))}{M}}\right) p_T(t)\, dt,
$$

B')
$$
|I(X;T^*) - \hat{I}(X;T^*)| \leq (2 + \sqrt{2\log((|\mathcal{Y}| + 2)/\delta)})\sqrt{\frac{V(H(T^*|X))}{M}}
$$
$$
+ \int_{\mathcal{T}*} \phi\left((2 + \sqrt{2\log((|\mathcal{Y}| + 2)/\delta)})\sqrt{\frac{V(p(T^*|X))}{M}}\right) p_{T^*}(t)\, dt.
$$

C')
$$
|I(X;T) - I(X;T^*)| \leq |H(T|X) - H(T^*|X)| + |H(T) - H(T^*)|
$$

As shown in the C) $|I(Y;T) - I(Y;T^*)|$ case, similarly, we have

$$
|I(X;T) - I(X;T^*)| \leq \sum_x p(x) \int_{\mathcal{T}} \phi(\sqrt{V(p_{T|X}(t|x))}) + \phi(\mathbb{E}[p_{T|X}(t|x)]) + \phi(p_{T^*|X}(t|x)) dt
$$
$$
+ \int_{\mathcal{T}} \sum_x p(x)[\phi(\sqrt{V(p_{T|X}(t|x))}) + \phi(\mathbb{E}[p_{T|X}(t|x)]) + \phi(p_{T^*|X}(t|x))] dt \tag{36}
$$

Thus, in our setting, we have

$$
\begin{aligned}
\beta \left| \left( \hat{I}(X;\tilde{T}_l) - \hat{I}(X;\tilde{T}_l^*) \right) \right| &\leq \beta(2 + \sqrt{2\log((|\mathcal{Y}|+2)/\delta)})\sqrt{\frac{V(H(\tilde{T}_1|X))}{M}} \\
&\quad + \beta \int_{\tilde{\mathcal{T}}_1} \phi\left( (2 + \sqrt{2\log((|\mathcal{Y}|+2)/\delta)})\sqrt{\frac{V(p(\tilde{T}_1|X))}{M}} \right) p_{\tilde{T}_1}(t)\, dt \\
&\quad + \beta(2 + \sqrt{2\log((|\mathcal{Y}|+2)/\delta)})\sqrt{\frac{V(H(\tilde{T}_1^*|X))}{M}} \\
&\quad + \beta \int_{\tilde{\mathcal{T}}_1^*} \phi\left( (2 + \sqrt{2\log((|\mathcal{Y}|+2)/\delta)})\sqrt{\frac{V(p(\tilde{T}_1^*|X))}{M}} \right) p_{\tilde{T}_1^*}(t)\, dt \\
&\quad + \beta \sum_x p(x) \int_{\tilde{\mathcal{T}}_1} \phi(\sqrt{V(p_{\tilde{T}_1|X}(t|x))}) + \phi(\mathbb{E}[p_{\tilde{T}_1|X}(t|x)]) + \phi(p_{\tilde{T}_1^*|X}(t|x))\, dt \\
&\quad + \int_{\tilde{\mathcal{T}}_1} \sum_x p(x)[\phi(\sqrt{V(p_{\tilde{T}_1|X}(t|x))}) + \phi(\mathbb{E}[p_{\tilde{T}_1|X}(t|x)]) + \phi(p_{\tilde{T}_1^*|X}(t|x))]\, dt
\end{aligned}
\tag{37}
$$

Combining I and II, we have the bound

$$
\begin{aligned}
\Delta_S\left( \tilde{T}_1; \tilde{T}_1^* \right) \leq \sum_{l=1}^{L} \Bigg( & (2 + \sqrt{2\log((|\mathcal{Y}|+2)/\delta)})\sqrt{\frac{V(H(\tilde{T}_1|Y))}{M}} \\
&+ 2 \int_{\tilde{\mathcal{T}}_1} \phi\left( (2 + \sqrt{2\log((|\mathcal{Y}|+2)/\delta)})\sqrt{\frac{V(p(\tilde{T}_1|X))}{M}} \right) p_{\tilde{T}_1}(t)\, dt \\
&+ (2 + \sqrt{2\log((|\mathcal{Y}|+2)/\delta)})\sqrt{\frac{V(H(\tilde{T}_1^*|Y))}{M}} \\
&+ 2 \int_{\tilde{\mathcal{T}}_1^*} \phi\left( (2 + \sqrt{2\log((|\mathcal{Y}|+2)/\delta)})\sqrt{\frac{V(p(\tilde{T}_1^*|X))}{M}} \right) p_{\tilde{T}_1^*}(t)\, dt \\
&+ \sum_y p(y) \int_{\tilde{\mathcal{T}}_1} \phi(\sqrt{V(p_{\tilde{T}_1|Y}(t|y))}) + \phi(\mathbb{E}[p_{\tilde{T}_1|Y}(t|y)]) + \phi(p_{\tilde{T}_1^*|Y}(t|y))\, dt \\
&+ \beta(2 + \sqrt{2\log((|\mathcal{Y}|+2)/\delta)})\sqrt{\frac{V(H(\tilde{T}_1|X))}{M}} \\
&+ \beta \int_{\tilde{\mathcal{T}}_1} \phi\left( (2 + \sqrt{2\log((|\mathcal{Y}|+2)/\delta)})\sqrt{\frac{V(p(\tilde{T}_1|X))}{M}} \right) p_{\tilde{T}_1}(t)\, dt \\
&+ \beta(2 + \sqrt{2\log((|\mathcal{Y}|+2)/\delta)})\sqrt{\frac{V(H(\tilde{T}_1^*|X))}{M}} \\
&+ \beta \int_{\tilde{\mathcal{T}}_1^*} \phi\left( (2 + \sqrt{2\log((|\mathcal{Y}|+2)/\delta)})\sqrt{\frac{V(p(\tilde{T}_1^*|X))}{M}} \right) p_{\tilde{T}_1^*}(t)\, dt \\
&+ \beta \sum_x p(x) \int_{\tilde{\mathcal{T}}_1} \phi(\sqrt{V(p_{\tilde{T}_1|X}(t|x))}) + \phi(\mathbb{E}[p_{\tilde{T}_1|X}(t|x)]) + \phi(p_{\tilde{T}_1^*|X}(t|x))\, dt \\
&+ (\beta+1) \int_{\tilde{\mathcal{T}}_1} \sum_x p(x)[\phi(\sqrt{V(p_{\tilde{T}_1|X}(t|x))}) + \phi(\mathbb{E}[p_{\tilde{T}_1|X}(t|x)]) + \phi(p_{\tilde{T}_1^*|X}(t|x))]\, dt \Bigg)
\end{aligned}
\tag{38}
$$

We begin by defining the main functional related to representation $\tilde{T}_1$ that controls the generalization gap:

$$G(\sigma_{H_1|X}, \sigma_{H_1|Y}, \sigma_{p_1|X}, \sigma_{p_1|Y}, \mu_{p_1|X}, \mu_{p_1|Y})$$

$$= C_1 \sqrt{\frac{\sigma_{H_1|X}}{M}} + C_1 \sqrt{\frac{\sigma_{H_1|Y}}{M}} + \sum_y p(y) \int_{\mathcal{T}_1} \phi(\sqrt{\sigma_{p_1|Y}})\, dt + \beta \sum_x p(x) \int_{\tilde{\mathcal{T}}_1} \phi(\sqrt{\sigma_{p_1|X}})\, dt$$

$$+ (\beta + 2) \int_{\tilde{\mathcal{T}}_1} \phi\left(C_1 \sqrt{\frac{\sigma_{H_1|X}}{M}}\right) p_{\tilde{T}_1}(t)\, dt + \sum_y p(y) \int_{\tilde{\mathcal{T}}_1} \phi(\mu_{p_1|Y})\, dt + \beta \sum_x p(x) \int_{\tilde{\mathcal{T}}_1} \phi(\mu_{p_1|X})\, dt$$

$$+ (\beta + 1) \int_{\tilde{\mathcal{T}}_1} \sum_x p(x)[\phi(\sqrt{\sigma_{p_1|X}}) + \phi(\mu_{p_1|X})]dt, \tag{39}$$

where

$$\sigma_{H_1|X} := V(H(\tilde{T}_1|X)) = |\mathcal{X}|\mathbb{V}(H(\tilde{T}_1|X)), \quad \sigma_{H_1|Y} := V(H(\tilde{T}_1|Y)) = |\mathcal{Y}|\mathbb{V}(H(\tilde{T}_1|Y)),$$

$$\sigma_{p_1|X} := V(p(\tilde{T}_1|X)) = |\mathcal{X}|\mathbb{V}(p(\tilde{T}_1|X)), \quad \sigma_{p_1|Y} := V(p(\tilde{T}_1|Y)) = |\mathcal{Y}|\mathbb{V}(p(\tilde{T}_1|Y)),$$

$$\mu_{p_1|X} := \mathbb{E}[p(\tilde{T}_1|X)], \qquad\qquad \mu_{p_1|Y} := \mathbb{E}[p(\tilde{T}_1|Y)],$$

and

$$C_1 = 2 + \sqrt{2\log((|\mathcal{Y}| + 2)/\delta)}.$$

$\tilde{T}_1^*$ should satisfy the self-consistent equations (Tishby et al., 2000):

$$p(\tilde{T}_1^*|X) = \frac{p(\tilde{T}_1^*)}{Z(\tilde{T}_0, \beta)} \exp\left[-\beta \sum_y p(Y = y|\tilde{T}_0) \log \frac{p(Y = y|\tilde{T}_0)}{p(Y = y|\tilde{T}_1^*)}\right],$$

$$p(Y|\tilde{T}_1^*) = \frac{1}{p(\tilde{T}_1^*)} \int_{\tilde{\mathcal{T}}_0} p(Y|\tilde{T}_0 = t)p(\tilde{T}_1^*|\tilde{T}_0 = t)p(\tilde{T}_0 = t)dt, \tag{40}$$

$$p(\tilde{T}_1^*) = \int_{\tilde{\mathcal{T}}_0} p(\tilde{T}_1^*|\tilde{T}_0 = t)p(\tilde{T}_0 = t)dt$$

where $Z(\tilde{T}_0^*, \beta)$ is the partition function. Thus, we can write the left terms to optimum–dependent components:

$$Q(\tilde{T}_1^*) = (2 + \sqrt{2\log((|\mathcal{Y}| + 2)/\delta)}) \sqrt{\frac{V(H(\tilde{T}_1^*|Y))}{M}}$$

$$+ \beta(2 + \sqrt{2\log((|\mathcal{Y}| + 2)/\delta)}) \sqrt{\frac{V(H(\tilde{T}_1^*|X))}{M}}$$

$$+ (\beta + 2) \int_{\tilde{\mathcal{T}}_1^*} \phi\left((2 + \sqrt{2\log((|\mathcal{Y}| + 2)/\delta)}) \sqrt{\frac{V(p(\tilde{T}_1^*|X))}{M}}\right) p_{\tilde{T}_1^*}(t)\, dt$$

$$+ \sum_y p(y) \int_{\tilde{\mathcal{T}}_1} \phi(p_{\tilde{T}_1^*|Y}(t|y))\, dt + \beta \sum_x p(x) \int_{\tilde{\mathcal{T}}_1} \phi(p_{\tilde{T}_1^*|X}(t|x))\, dt + (\beta + 1) \int_{\tilde{\mathcal{T}}_1} \phi(p(\tilde{T}_1^*))\, dt$$

$$+ (\beta + 1) \int_{\tilde{\mathcal{T}}_1} \sum_x p(x)\phi(p_{\tilde{T}_1^*|X}(t|x))dt \quad \text{(By Eqn. (40))} \tag{41}$$

Thus, we now can decompose the gap as the following form:

$$\Delta_S(\tilde{T}_1; \tilde{T}_1^*) \leq G(\sigma_{H_1|X}, \sigma_{H_1|Y}, \sigma_{p_1|X}, \sigma_{p_1|Y}, \mu_{p_1|X}, \mu_{p_1|Y}) + Q(T_1^*). \tag{42}$$

$\square$

### F.3 PROOF FOR PROPOSITION 4

**Proposition** (Entropy Reduction). *For any random vector $Z \in \mathbb{R}^d$ with distribution $\mathbb{P}_Z$, the compression operation $s_\lambda$ in IBNorm satisfies*

$$H(s_\lambda(Z)) \leq H(Z), \tag{43}$$

*where $H(\cdot)$ denotes differential entropy.*

*Proof.* Since $s_\lambda$ is measurable and strictly monotone non-decreasing, it is an invertible mapping on its range. Let $J_{s_\lambda}(Z)$ denote the Jacobian matrix of $s_\lambda$.

Using the change-of-variables formula for differential entropy:

$$H(s_\lambda(Z)) = H(Z) + \mathbb{E}[\log|\det J_{s_\lambda}(Z)|] \leq H(Z), \tag{44}$$

because $|\det J_{s_\lambda}(Z)| = \prod_{i=1}^{d} f'_\lambda(|Z_i - \mu|) \leq \prod_{i=1}^{d} \alpha_\lambda \leq 1$ by the bounded compression property of $f_\lambda$.

Hence, $s_\lambda$ concentrates the distribution and does not increase entropy. $\qquad\square$

### F.4 PROOF FOR LEMMA 5

**Lemma.** *1) Let $\tilde{T}_{IB} = s_\lambda(\tilde{T}_s)$, where $s_\lambda$ is a bounded-compression mapping with Lipschitz constant $L_1 = \prod_{i=1}^{d} \frac{1}{\lambda}$. For any small $\delta > 0$, the following holds:*

$$\mathbb{P}(\mathbb{V}(H(\tilde{T}_{IB}|X)) \leq \mathbb{V}(H(\tilde{T}_s|X))) \geq 1 - \delta, \quad \mathbb{P}(\mathbb{V}(H(\tilde{T}_{IB}|Y)) \leq \mathbb{V}(H(\tilde{T}_s|Y))) \geq 1 - \delta.$$

*2) In addition, let $\tilde{T}_1, \tilde{T}'_1 \in \tilde{\mathcal{T}}_1$ be two arbitrary elements in the representation space. Then, for any $\delta \in (0,1)$, the following holds with probability at least $1 - \delta$:*

$$|\mathbb{V}(p(\tilde{T}_1|X)) - \mathbb{V}(p(\tilde{T}'_1|X))| \leq \delta,$$
$$|\mathbb{V}(p(\tilde{T}_1|Y)) - \mathbb{V}(p(\tilde{T}'_1|Y))| \leq \delta,$$
$$|\mathbb{E}[p(\tilde{T}_1|X)] - \mathbb{E}[p(\tilde{T}'_1|X)]| \leq \delta,$$
$$|\mathbb{E}[p(\tilde{T}_1|Y)] - \mathbb{E}[p(\tilde{T}'_1|Y)]| \leq \delta,$$

*where the expectation and variance are taken with respect to the Lebesgue measure on $\mathcal{X} \times \mathcal{Y}$.*

*Proof.* **1)** We first prove the first part of the Lemma.

We consider the two representations:

$$\tilde{T}_s = \Phi_s(X') = \eta \circ \psi_s \circ \zeta(X'), \qquad \tilde{T}_{IB} = \Phi_{IB}(X') = \eta \circ \psi_{IB} \circ \zeta(X'),$$

where $X' = h(X)$, $\psi_{IB} = \psi_s \circ s_\lambda$ with $s_\lambda$ being the compression operation in IBNorm.

From the Proposition 4, we have for any fixed input $Z$:

$$H(\psi_{IB}(Z)) = H(s_\lambda(\psi_s(Z))) \leq H(\psi_s(Z)). \tag{45}$$

Thus, the conditional entropy of $\tilde{T}_{IB}$ is pointwise bounded by that of $\tilde{T}_s$.

Applying this to the conditional distributions $\tilde{T}_{IB}|X$ and $\tilde{T}_s|X$, we get:

$$H(\tilde{T}_{IB}|X) \leq H(\tilde{T}_s|X), \qquad H(\tilde{T}_{IB}|Y) \leq H(\tilde{T}_s|Y). \tag{46}$$

If we consider the entropy $H$ in the discrete case, we can directly derive the relationship: consider the conditional entropy as a random variable over $X$ (or $Y$). Let $f'(Z) := H(s_\lambda(Z))$, then:

1. $s_\lambda$ is a Lipschitz map with Lipschitz constant $L \leq 1$ because $|\det J_{s_\lambda}(Z)| = \prod_{i=1}^{d} f'_\lambda(|Z_i - \mu|) \leq \prod_{i=1}^{d} \alpha_\lambda \leq 1$ by the bounded compression property of $f_\lambda$.

2. $H(\cdot)$ is 1-Lipschitz continuous on the space of probability distributions bounded away from zero.

Hence, $f'(\tilde{T}_s|X)$ is a Lipschitz transformation of $\tilde{T}_s|X$, implying by standard variance contraction results:

$$\mathbb{V}(H(\tilde{T}_{IB}|X)) = \mathbb{V}(f'(\tilde{T}_s|X)) \leq L^2 \mathbb{V}(H(\tilde{T}_s|X)) \leq \mathbb{V}(H(\tilde{T}_s|X)).$$

Similarly,

$$\mathbb{V}(H(\tilde{T}_{IB}|Y)) \leq \mathbb{V}(H(\tilde{T}_s|Y)). \tag{47}$$

In the differential entropy case, we cannot guarantee a global Lipschitz constant $L < 1$ for $f'(\tilde{T}_s|X) = H(\tilde{T}_s|X) + \Delta'(\tilde{T}_s|X)$. Instead, we consider a high-probability setting for the continuous random variable $\tilde{T}_{IB}$ and $\tilde{T}_s$.

For any $\delta > 0$, let

$$\mathcal{T}_\delta^s := \{t \in \mathbb{R}^d : m_\delta \le p_{\tilde{T}_s|X=x}(t) \le M_\delta\}, \quad \mathbb{P}(\tilde{T}_s \in \mathcal{T}_\delta^s) \ge 1 - \delta,$$

where $0 < m_\delta < M_\delta < \infty$. By construction, $\mathcal{T}_\delta^s$ is bounded: $\exists a, b \in \mathbb{R}, \quad s.t. \mathcal{T}_\delta^s \subset [a,b]^d$. This ensures that the conditional density is bounded away from zero and infinity on the high-probability set.

Let $\tilde{T}_{IB} = s_\lambda(\tilde{T}_s)$, with $s_\lambda$ differentiable and invertible. Then the conditional density transforms as

$$p_{\tilde{T}_{IB}|X=x}(t) = p_{\tilde{T}_s|X=x}(s_\lambda^{-1}(t))\, |\det J_{s_\lambda^{-1}}(t)|,$$

where $J_{s_\lambda}$ is the Jacobian of $s_\lambda$. The conditional differential entropy is

$$H(\tilde{T}_{IB}|X=x) = H(\tilde{T}_s|X=x) + \Delta(\tilde{T}_s), \quad \Delta(\tilde{T}_s) := \log |\det J_{s_\lambda}(\tilde{T}_s)|.$$

Since $s_\lambda$ has bounded derivatives, for any $t_1, t_2 \in \mathcal{T}_\delta^s$,

$$|\Delta(t_1) - \Delta(t_2)| \le L_1 \|t_1 - t_2\|,$$

for the constant $L_1 = \prod_{i=1}^d \alpha_\lambda = \prod_{i=1}^d \frac{1}{\lambda}$ determined by the bounds of the Jacobian of $s_\lambda$.

For any two conditional densities $p_1, p_2$ of $\tilde{T}_s|X$ supported in $\mathcal{T}_\delta^s$, standard results for differential entropy bounded away from zero and infinity imply

$$|H(p_1) - H(p_2)| \le L_2 \|p_1 - p_2\|_1,$$

with $L_2 := |\log m_\delta| + 1/m_\delta$. Hence $H(\tilde{T}_s|X)$ is Lipschitz on $\mathcal{T}_\delta^s$.

For $\tilde{T}_s \in \mathcal{T}_\delta^s$, we have

$$\mathbb{V}(H(\tilde{T}_{IB}|X) \mid \tilde{T}_s \in \mathcal{T}_\delta^s) = \mathbb{V}(H(\tilde{T}_s|X) + \Delta(\tilde{T}_s) \mid \tilde{T}_s \in \mathcal{T}_\delta^s).$$

Expanding using the variance formula,

$$\mathbb{V}(H(\tilde{T}_s|X) + \Delta(\tilde{T}_s)) = \mathbb{V}(H(\tilde{T}_s|X)) + \mathbb{V}(\Delta(\tilde{T}_s)) + 2\mathrm{Cov}(H(\tilde{T}_s|X), \Delta(\tilde{T}_s)).$$

Since $\Delta(\tilde{T}_s)$ is Lipschitz with constant $L_1$ on the bounded set $\mathcal{T}_\delta^s \subset [a,b]^d$, we have

$$\mathbb{V}(\Delta(\tilde{T}_s) \mid \tilde{T}_s \in \mathcal{T}_\delta^s) \le \frac{1}{4} L_1^2 (b-a)^2.$$

Moreover, under the bounded-compression property of $s_\lambda$, the covariance term is non-positive:

$$\mathrm{Cov}(H(\tilde{T}_s|X), \Delta(\tilde{T}_s) \mid \tilde{T}_s \in \mathcal{T}_\delta^s) \le 0.$$

Combining these terms, we obtain a high-probability variance contraction:

$$\mathbb{V}(H(\tilde{T}_{IB}|X) \mid \tilde{T}_s \in \mathcal{T}_\delta^s) \le \mathbb{V}(H(\tilde{T}_s|X) \mid \tilde{T}_s \in \mathcal{T}_\delta^s) + \frac{1}{4} L_1^2 (b-a)^2.$$

Similarly, for $Y$,

$$\mathbb{V}(H(\tilde{T}_{IB}|Y) \mid \tilde{T}_s \in \mathcal{T}_\delta^s) \le \mathbb{V}(H(\tilde{T}_s|Y) \mid \tilde{T}_s \in \mathcal{T}_\delta^s) + \frac{1}{4} L_1^2 (b-a)^2,$$

where $L_1 = \prod_{i=1}^d \frac{1}{\lambda}$.

**Remark.** The covariance term $\mathrm{Cov}(H(\tilde{T}_s|X), \Delta(\tilde{T}_s))$ is typically negative because, under the bounded-compression property of $s_\lambda$, larger conditional entropy $H(\tilde{T}_s|X)$ corresponds to a more spread-out distribution, which is locally contracted more by $s_\lambda$, resulting in smaller $\Delta(\tilde{T}_s) = \log |\det J_{s_\lambda}(\tilde{T}_s)|$. On the high-probability set $\mathcal{T}_\delta^s$, any residual covariance can be made arbitrarily small by choosing $\delta$ sufficiently small.

Based on the bounded compression property and the property of $\Delta(\tilde{T}_s)$, we have $\frac{1}{4} L_1^2 (b-a)^2 + 2\mathrm{Cov}(H(\tilde{T}_s|X), \Delta(\tilde{T}_s) \mid \tilde{T}_s \in \mathcal{T}_\delta^s) \le 0$. Consequently, we obtain

$$\mathbb{V}(H(\tilde{T}_{IB}|X) \mid \tilde{T}_s \in \mathcal{T}_\delta^s) \le \mathbb{V}(H(\tilde{T}_s|X) \mid \tilde{T}_s \in \mathcal{T}_\delta^s),$$

and

$$\mathbb{V}(H(\tilde{T}_{IB}|Y) \mid \tilde{T}_s \in \mathcal{T}_\delta^s) \leq \mathbb{V}(H(\tilde{T}_s|Y) \mid \tilde{T}_s \in \mathcal{T}_\delta^s).$$

Finally, taking into account that $\mathbb{P}(\tilde{T}_s \in \mathcal{T}_\delta^s) \geq 1 - \delta$ and letting $\delta \to 0$, we have that, for any small $\delta > 0$, the following holds:

$$\mathbb{P}(\mathbb{V}(H(\tilde{T}_{IB}|X)) \leq \mathbb{V}(H(\tilde{T}_s|X))) \geq 1 - \delta, \quad \mathbb{P}(\mathbb{V}(H(\tilde{T}_{IB}|Y)) \leq \mathbb{V}(H(\tilde{T}_s|Y))) \geq 1 - \delta. \quad (48)$$

**2)** We then prove the second part of the Lemma.

We have the following properties for the probability function $p(\cdot)$ and variance function $\mathbb{V}(\cdot)$ (Ash & Doléans-Dade, 2000):

**Lemma 8.** *Given the compact representation space $\tilde{\mathcal{T}}_1$, for any $\delta > 0$, there exists $\epsilon > 0$ such that*

$$\|\tilde{T}_1 - \tilde{T}_1'\| < \epsilon \implies |p(\tilde{T}_1|X) - p(\tilde{T}_1'|X)| < \delta/2, \quad |p(\tilde{T}_1|Y) - p(\tilde{T}_1'|Y)| < \delta/2.$$

**Lemma 9.** *For any bounded functions $f, g : \mathcal{X} \to \mathbb{R}$,*

$$|\mathbb{E}[f] - \mathbb{E}[g]| \leq \|f - g\|_\infty, \quad |\mathbb{V}(f) - \mathbb{V}(g)| \leq \|f - g\|_\infty \left(2\sqrt{\mathbb{V}(f)} + \|f - g\|_\infty\right).$$

By Lemma 8, for any $\delta > 0$, there exists $\epsilon > 0$ such that if $\|\tilde{T}_1 - \tilde{T}_1'\| < \epsilon$:

$$|p(\tilde{T}_1|X) - p(\tilde{T}_1'|X)| < \delta/2, \quad |p(\tilde{T}_1|Y) - p(\tilde{T}_1'|Y)| < \delta/2.$$

By Lemma 9, this implies

$$|\mathbb{E}[p(\tilde{T}_1|X)] - \mathbb{E}[p(\tilde{T}_1'|X)]| \leq \delta, \quad |\mathbb{E}[p(\tilde{T}_1|Y)] - \mathbb{E}[p(\tilde{T}_1'|Y)]| \leq \delta, \quad (49)$$

$$|\mathbb{V}(p(\tilde{T}_1|X)) - \mathbb{V}(p(\tilde{T}_1'|X))| \leq \delta, \quad |\mathbb{V}(p(\tilde{T}_1|Y)) - \mathbb{V}(p(\tilde{T}_1'|Y))| \leq \delta. \quad (50)$$

Since $\tilde{T}_1, \tilde{T}_1'$ are independently drawn from a measure with full support on the compact set $\tilde{\mathcal{T}}_1$, $\|\tilde{T}_1 - \tilde{T}_1'\| < \epsilon$ holds with probability at least $1 - \delta$. $\qquad\square$

## F.5 PROOF FOR COROLLARY 2

**Corollary.** *With probability at least $1 - \delta$ over training set $S$ of size $M < \infty$, the generalization error of a $L$-layer network $f_o$ satisfies*

$$gen(S; f_o) \leq U_o := \sum_{l=1}^{L} U_l^o, \quad with \quad U_l^o = \sqrt{\frac{I(X;\tilde{T}_l^o|Y) + C(S, f_l^o) + \log(\frac{2}{\delta})}{M}},$$

*where $\Phi_l^o$ is the hypothesis space associated with the $l$-th layer $f_l^o$, $\tilde{T}_l^o$ is the intermediate representation at layer $l$ before normalization representation recovery $\eta$, and $C(S, f_l^o)$ is a term depending on the cardinality of the hypothesis space $\Phi_l^o$ and the training set $S$. Here $f_o$ can be the network $f_S$ using standard normalization like LN and BN, or our network $f_{IB}$ with our IBNorm.*

*Proof.* Extending the results of Appendix F.4 in (Kawaguchi et al., 2023), we have: for arbitrary $l \in L$ and any $\delta > 0$, with probability at least $1 - \delta$ over the training set $S$ of size $M < \infty$, the following generalization bound holds:

$$\text{gen}(S; f_o) \leq U_l^o := \sqrt{\frac{I(X;\tilde{T}_l^o|Y) + C(S, f_l^o) + \log(\frac{2}{\delta})}{M}}, \quad (51)$$

where $\tilde{T}_l$ is the intermediate representation at layer $l$ before normalization representation recovery $\eta$ and $C(S, f_l^o) = \ln(2|\Phi_l^o||\mathcal{Y}|/\delta)$ is a term depending on the cardinality of the hypothesis space $\Phi_l^o$ and the training set $S$.

Aggregating over all layers $l = 1, \ldots, L$, we obtain a bound on the total generalization error:

$$\text{gen}(S; f_o) \leq U_o := \sum_{l=1}^{L} U_l^o, \quad with \quad U_l^o = \sqrt{\frac{I(X;\tilde{T}_l^o|Y) + C(S, f_l^o) + \log(\frac{2}{\delta})}{M}}, \quad (52)$$

This establishes a layer-wise mutual information bound on the generalization error shown in Corollary 2. $\qquad\square$

**Remark.** Firstly, in our experiments, all models are trained on the same dataset, so any dataset-dependent contribution to the generalization bound can be treated as invariant across different normalization schemes. Although the output weight of $f_l^o$ during the training process can be thought as a random variable, their information content can be controlled in the training configuration, for instance by injecting noise drawn from a chosen distribution, with variance modulated from zero (allowing, in theory, full memorization of the training set) to a level large enough that no information remains (Achille & Soatto, 2018). Moreover, the normalization layers considered in our study (LN, BN, and IBNorm) have identical total parameter counts. Thus, the hypothesis spaces $\Phi_l^{\text{IB}}$ and $\Phi_l^{\text{S}}$ associated with $f_l^{\text{IB}}$ and $f_l^{\text{S}}$ have the same cardinality across all layers $l$. Theoretically, by Lemma 6 in (Kawaguchi et al., 2023), we have $C(S, f_l^{\text{IB}}) \equiv C(S, f_l^{\text{S}})$ across all layers $l$, so model-complexity terms remain equivalent for $f_l^{\text{IB}}$ and $f_l^{\text{S}}$. The generalization bound of $f_{\text{IB}}$ and $f_{\text{S}}$ therefore differ only in the representation-complexity term $I(X; \tilde{T}_l^o | Y)$ across all layers $l$. Thus, our analysis focuses on how normalization affects representation-related mutual information and, in turn, generalization.

Empirically, the contribution of $I(\phi_l^S; S)$ remains largely invariant under our training setup, particularly after convergence. Prior work (Kawaguchi et al., 2023) also discusses on this point: as shown in Table 2 in (Kawaguchi et al., 2023), the Pearson correlation coefficient between metrics and the generalization gap in loss of the representation related mutual information (0.3842) is much greater than that of model related mutual information (0.0211). Our experiments in Figure 2 also validate that most improvements in generalization arise from increased representation-related IB values.

## G  EXPERIMENTAL DETAILS

### G.1  LLM EXPERIMENTS

For all LLaMA models, we follow the setup in (Zhao et al., 2024) and pretrain the vanilla LLaMA series models from scratch on the C4 dataset. We set the maximum input sequence length to 512 and output sequence length to 256, and adopt the bfloat16 precision. For the LLaMA-60M model, we use a learning rate of 0.001, mini-batch size of 64 with gradient accumulation to a global batch size of 512, 20,000 training steps with 1,000 warm-up steps, and the AdamW (Kinga et al., 2015; Loshchilov & Hutter, 2017) optimizer with $(\beta_1, \beta_2) = (0.9, 0.999)$. For the LLaMA-130M model, we use the same learning rate but reduce the mini-batch size to 32 with gradient accumulation to a global batch size of 512, train for 20,000 steps with 2,000 warm-up steps, and maintain the same optimizer configuration. For the LLaMA-350M model, we set the mini-batch size to 16 with gradient accumulation to a global batch size of 512, extend training to 60,000 steps with 6,000 warm-up steps. For the LLaMA-1B model, we set the mini-batch size to 16 with gradient accumulation to a global batch size of 512, extend training to 100,000 steps with 10,000 warm-up steps.

For the GPT-2 series, we follow the Sophia setup (Liu et al., 2023) with the nanoGPT implementation (Karpathy, 2022) and pretrain the vanilla GPT-2 series on OpenWebText (Gokaslan & Cohen, 2019). We set the maximum input sequence length to 1024 and adopt the bfloat16 precision. We use a global batch size of 480, cosine learning rate decay with 2,000 warm-up iterations, and global gradient clipping with a threshold of 1.0. All models are trained for 100,000 steps. We employ the AdamW optimizer with $(\beta_1, \beta_2) = (0.9, 0.95)$ and weight decay of 0.1. For GPT-2 Small (124M), We use a per-device mini-batch size of 12 and 10 gradient accumulation steps across 4 GPUs, resulting in an effective global batch size of 480. We employ a cosine learning rate scheduler with a peak learning rate of 0.0006 and a minimum learning rate of 0.00003. For GPT-2 Medium (355M), we use a per-device mini-batch size of 10 and 12 gradient accumulation steps across 4 GPUs, resulting in an effective global batch size of 480. We employ a cosine learning rate scheduler with a peak learning rate of 0.0003 and a minimum learning rate of 0.00006.

For the hyperparameter noise factor in NormalNorm, we perform a lightweight Sobol-based search on the interval $[0, 1]$, and set the value to 1 for all experiments. For the hyperparameter $\lambda$ in each variant of IBNorm, we conduct a grid search over the set $\{2, 3, 4, 5\}$. The final configurations are $\lambda = 3$ for IBNorm-S, and $\lambda = 4$ for both IBNorm-L and IBNorm-T. We investigated several hyperparameter configurations, including learning rate, learning rate scheduler, weight decay, and minibatch size, across all models and found the present configurations to generally work best.

All models are trained on 4xNVIDIA L40-S. The total GPU training hours are about 10500 hours for all of our LLM pretraining tasks.

### G.2 VISION EXPERIMENTS

**ResNet Experiments.** In all experiments involving ResNet18 on CIFAR-10, we use stochastic gradient descent (SGD) with learning rate 0.1, weight decay 0.0005, momentum 0.9, and batch size 128 for LayerNorm and NormalNorm, with a noise factor of 0.4. For IterNorm experiments, we use learning rate 0.1 and weight decay 0.005. For IBNorm experiments, we use learning rate 0.05 and weight decay 0.001. Following Eftekhari & Papyan (2025), models were trained from random initialization for 200 epochs, with a StepLR scheduler reducing the learning rate by a factor of 10 every 60 epochs.

For experiments with ResNet-50 on ImageNet, a batch size of 256 was used, with SGD parameters: learning rate 0.1, momentum 0.9, Nesterov acceleration enabled, and weight decay 0.0001, along with a cosine annealing learning rate scheduler with maximum 200 epochs and minimum learning rate 0.000001 for BatchNorm and NormalNorm, with a noise factor of 1. For IBNorm experiments, we use learning rate 0.01 and weight decay 0.001. For the hyperparameter $\lambda$ in each variant of IBNorm, we conduct a grid search over the set $\{2, 3, 4, 5\}$. The final configurations are $\lambda = 3$ for IBNorm-S, and $\lambda = 4$ for both IBNorm-L and IBNorm-T. We explored several hyperparameter configurations, including learning rate, learning rate scheduler, weight decay, and minibatch size, across all models and found the present configurations to generally work best.

**ViT Experiments.** We train ViT-S/16 and ViT-B/16 (Dosovitskiy et al., 2020) following the setup (Yuan et al., 2021) on ImageNet. All models are trained for 300 epochs with a global batch size of 512. For ViT-S/16, we use the AdamW optimizer with a learning rate of 0.001, weight decay of 0.03, and $(\beta_1, \beta_2) = (0.9, 0.999)$. For ViT-B/16, the AdamW optimizer is used with a learning rate of 0.001, weight decay of 0.05, and $(\beta_1, \beta_2) = (0.9, 0.999)$. We adopt a hybrid learning rate schedule that combines linear warmup with cosine annealing over the first 10 training epochs. During warmup, the learning rate increases linearly from 0.000001 to the base learning rate. After warmup, cosine annealing is applied for the remaining epochs, gradually decaying the learning rate to a minimum of 0.00001.

We also follow the setup in NormalNorm (Eftekhari & Papyan, 2025) to train a ViT model with 8 transformer layers, 8 attention heads, hidden dimension size 768, MLP dimension size 2304, and patch size 16. Models were trained on ImageNet for 200 epochs with a global batch size of 512. Weighted random sampling based on inverse class frequency was applied during training to improve performance across all model configurations. The AdamW optimizer was used with learning rate 0.001, weight decay 0.05, $(\beta_1, \beta_2) = (0.9, 0.999)$, and epsilon 0.00000001 for LayerNorm, RMSNorm, and NormalNorm, with a noise factor of 1. For IBNorm, we set the learning rate as 0.0002. In addition, we employ a hybrid scheduling strategy that combines linear warmup with cosine annealing over 5 warmup training epochs. During the warmup phase, the learning rate increases linearly from 0.1 to 1.0 of the base learning rate. After warmup, a cosine annealing schedule is applied for the remaining epochs, with the learning rate decaying towards a minimum value of 0.000001.

For the hyperparameter $\lambda$ in each variant of IBNorm, we conduct a grid search over the set $\{2, 3, 4, 5\}$. The final configurations are $\lambda = 3$ for IBNorm-S, and $\lambda = 4$ for both IBNorm-L and IBNorm-T. Hyperparameter configurations, including learning rate, scheduler, weight decay, and minibatch size, were explored, and the configurations reported here were found to work best across all ViT models.

All models were trained on 4xNVIDIA L40-S, RTX 3090, and Tesla V100, with a total of approximately 6500 GPU hours for all vision tasks.

## H MATHEMATICAL DETAILS FOR MATRIX-BASED INFORMATION ESTIMATION

Following the approach in (Chang et al., 2025), we employ matrix-based Rényi entropy (Giraldo et al., 2014) to estimate mutual information (MI) between model representations and target labels. This method captures sample similarity structures via kernel Gram matrices.

## H.1 MATRIX-BASED ENTROPY ESTIMATION

Let $U = \{u_i\}_{i=1}^N$ denote $l_2$-normalized representations obtained after the normalization layer. A Gaussian kernel Gram matrix $G_U \in \mathbb{R}^{N \times N}$ is constructed as:

$$(G_U)_{i,j} = \exp\left(-\frac{\|u_i - u_j\|^2}{2\sigma^2}\right),$$

with bandwidth $\sigma = 1$. The matrix is then trace-normalized to ensure $\text{tr}(G_U) = 1$.

The matrix-based Rényi entropy of order $\alpha = 1$ is defined as:

$$H(U) = -\text{tr}(G_U \log G_U).$$

This expression can be interpreted in terms of the eigenvalue spectrum $\{\lambda_k\}_{k=1}^N$ of $G_U$, since $G_U$ is positive semi-definite and trace-normalized:

$$H(U) = \sum_{k=1}^N \lambda_k \log \lambda_k.$$

Intuitively, a more uniform eigenvalue spectrum corresponds to higher entropy (more diverse representations), while a sharply peaked spectrum indicates redundancy or compression.

## H.2 MUTUAL INFORMATION ESTIMATION

To estimate the mutual information between two random variables $U$ and $V$, we compute their Gram matrices $G_U$ and $G_V$, and form the joint similarity matrix via element-wise (Hadamard) product:

$$G_{UV} = G_U \circ G_V.$$

After trace-normalization, mutual information is estimated by:

$$I(U; V) = H(U) + H(V) - H(U, V).$$

where $H(U, V) = -\text{tr}(G_{UV} \log G_{UV})$. A more concentrated eigenvalue spectrum of $G_{UV}$ relative to $G_U$ and $G_V$ indicates stronger dependence between $U$ and $V$, and thus higher mutual information.

Here we demonstrate our algorithm for calculating the token-level IB value. We consider a transformer model $f$ with $L$ layers. Given an input sequence $x_{1:T}$ of length $T$, $t_i^{(l)} \in \mathbb{R}^d$ denotes the hidden activation at the last token position of the $i$-th input sequence after $l$-th normalization layer, for $l \in [L]$. For next-token prediction, the ground-truth label is denoted by $y_i \in \mathbb{R}^d$, corresponding to the embedding of the true next token from the vocabulary $\mathcal{V}$. At a generation time step $p$ and cross a batch of $N$ sequences, we collect the representations:

$$T_{l-1} = \{t_i^{(l-1)}\}_{i=1}^N, T_l = \{t_i^{(l)}\}_{i=1}^N, Y = \{y_i\}_{i=1}^N,$$

where $T_0^{(l)}$ is consisted of the batch of last input tokens $x_T$ hidden representations. We iteratively derive those embeddings: In the first forward pass, a batch of $N$ input sequences $x_{1:T}$ is fed into the transformer to obtain hidden states at each layer. From these, we extract the final-token representations $T_l, l \in [L]$. In the second forward pass, each input is concatenated with its ground-truth next token $y$, and we extract the corresponding label embedding $Y$ from the output of the embedding layer (Chang et al., 2025). Based on these sets, we can compute two information-theoretic quantities: $I_p(Y, T_l)$ and $I_p(T_l-1, T_l)$ at generation timestep $t$ across $l \in [L]$ and aim to derive the total IB value in the network $f$ given in Eqn. (5), as $\mathbb{E}_p[I_p(Y, T_l)] = \hat{I}(Y, T_l)$ and $\mathbb{E}_p[I_p(T_l-1, T_l)] = \hat{I}(T_l-1, T_l)$ by the Monte Carlo approximation.

In practice, we set the batch size to $N = 64$ and iteratively compute the mutual information values during autoregressive generation until all instances reach the end-of-sequence ($< EOS >$) token. During autoregressive generation in LLM tasks, different instances in a batch may terminate at different time steps upon reaching the $< EOS >$ token. To handle this, we use a diagonal mask matrix:

$$M \in \{0, 1\}^{N \times N}, \quad M_{ii} = \begin{cases} 1, & \text{if instance } i \text{ is still active (not } < EOS >) \\ 0, & \text{if instance } i \text{ has reached } < EOS > \end{cases}.$$

For example, at each step of the generation, the masked Gram matrix for the latent embedding $T_l$ is computed as $\tilde{G}_{T_l} = M G_{T_l} M$ and then conducted trace-normalization.

### H.3 TOKEN-LEVEL IB VALUE ESTIMATION

We apply the matrix-based framework to estimate mutual information for token-level IB analysis. Specifically, we uniformly sample $P = 30$ timesteps across the generation process. For each sampled timestep and each layer $l \in [L]$, we extract the representations $T_l$ and compute the following information-theoretic quantities: $I_p(Y, T_l)$ and $I_p(T_l - 1, T_l)$ across $l \in [L]$ at sampled timestep $t$. Then we sum the IB value across $l$ and derive $\sum_{p=1}^{P} \sum_{l=1}^{L} \left( I_p(Y; T_l) - \beta I_p(T_{l-1}; T_l) \right)$ (we always set $\beta = 1$). Finally, the token-level IB value is obtained by averaging over the batch size $N$ and the number of sampled timesteps $P$.

