# OpenReview forum: "IBNorm: Information-Bottleneck Inspired Normalization for Representation Learning"
_ICLR.cc/2026/Conference — Submitted to ICLR 2026_

### Official Review · Reviewer_9sfx · 2025-10-27

**Soundness:** 1
**Presentation:** 2
**Contribution:** 1
**Rating:** 2
**Confidence:** 4

**Summary:**

This paper introduces IBNorm, a novel normalization technique for deep learning inspired by the Information Bottleneck (IB) principle. It argues that common methods like BatchNorm and LayerNorm primarily control mean and variance, neglecting the informational content of representations. IBNorm incorporates a non-linear "compression" step before standardization, aiming to shape activations to preserve task-relevant information while suppressing nuisance variability, thus aligning better with IB objectives. The authors claim theoretical guarantees that IBNorm achieves higher IB values and tighter generalization bounds than standard methods. Empirically, they demonstrate performance improvements by integrating IBNorm into various language models (LLaMA, GPT-2) and vision models (ResNet, ViT) across several benchmarks, outperforming standard normalization techniques and the recent NormalNorm method. The paper also provides empirical analysis using mutual information estimators to support the claim that IBNorm leads to representations with better IB characteristics.

**Strengths:**

* The paper clearly identifies a potential limitation of existing normalization methods (focusing solely on moments) and proposes addressing it through the principled lens of the Information Bottleneck.
* The method is evaluated across different domains (NLP, CV), architectures (Transformers, CNNs), and model scales, providing a broad assessment of its practical performance.
* IBNorm seems to outperform standard baselines (LN, BN, RMSNorm) and a relevant recent competitor (NormalNorm) across most reported experiments, suggesting potential practical utility.

**Weaknesses:**

The paper's primary weaknesses lie in the lack of sound theoretical justification for its central claims and unfair empirical comparisons in the vision domain.

1.  **Unfair Empirical Comparison in Vision Experiments:** The claimed empirical superiority in vision models (Table 3) is based on an unfair comparison. Appendix F.2 reveals that the authors used different hyperparameters (learning rates, weight decays) for their proposed IBNorm method compared to the baseline methods (including BatchNorm, LayerNorm, and NormalNorm). For example, in the ResNet50/ImageNet experiment, IBNorm used a learning rate 10x smaller and weight decay 10x larger than the baselines. Claiming superior performance when the proposed method received potentially more favorable tuning invalidates the comparison. This experimental flaw makes the vision results unsubstantiated.
2.  **Invalid Proof for Generalization Bound (Corollary 2):** The proof attempts to justify a tighter generalization bound by invoking Theorem 2 from Kawaguchi et al. (2023). This invocation is flawed for multiple reasons:
    * Misrepresentation of the Bound: The paper incorrectly presents the bound from Kawaguchi et al. It omits a critical term, $I(\phi\_l^S; S)$, which represents the mutual information between the learned encoder parameters and the training data, capturing model complexity/overfitting. The actual bound depends on *both* $ I(X; Z\_l^s|Y)$ (representation complexity) *and* $I(\phi_l^S; S)$ (model complexity).
    * Incorrect Structure: The paper incorrectly changes the bound's structure from a minimum over layers ($\min_l$) in the source to a sum over layers ($\sum_l$).
    * Unchecked Assumptions: The paper fails to verify that the assumptions required by the Kawaguchi et al. theorem (e.g., regarding data generation, finite spaces) hold in their setting.
    * This constitutes a logical flaw because the cited theorem is fundamentally misrepresented and misapplied, invalidating the proof and the claim of a tighter generalization bound.
3.  Invalid Proof for Higher IB Value (Theorem 1): The proof that IBNorm achieves a higher IB value (Theorem 1, Appendix E.1-E.4) relies on applying mathematical bounds (via Lemma 3, citing Shamir et al. 2010) derived for discrete random variables directly to the continuous activations of neural networks.
    * The theoretical tools used (concentration bounds based on variances over finite sample spaces from Shamir et al. 2010) are not generally applicable to continuous distributions without significant adaptation or justification, which is not provided. Furthermore, the subsequent Lemma 5 contains questionable arguments regarding differential entropy and appears conditional on the compression hyperparameter lambda being large, contradicting the main theorem statement.
    * While potentially fixable with different bounding techniques appropriate for continuous variables, the current proof relies on fundamentally misapplied tools and contains questionable steps, rendering it invalid as presented. This is a significant flaw and an alternative argument or major restructuring appears necessary.
4.  Flawed Logic Connecting IB Value to Generalization: The argument in Section 4.3 linking the claimed higher IB value (Theorem 1) to the claimed tighter generalization (Corollary 2) is logically flawed. It relies on the invalid proofs for both Theorem 1 and Corollary 2. Moreover, it *ignores* the necessary $I(\phi\_l^S; S)$ term from the actual Kawaguchi et al. bound, failing to provide any analysis of how IBNorm might impact model complexity, which is essential for assessing the overall generalization bound.
5. Lack of rigor in LLM gains reporting: While the LLM results appear promising, the magnitude of gains over baselines (especially the internally-run NormalNorm) is surprisingly large. Crucially, these results are reported based on single runs (there is no mention of runs in the main paper or appendix), lacking error bars or statistical significance testing, which makes it difficult to assess their reliability and may potentially overstate the gains.

**Questions:**

-  Theorem 1 Proof: Can the authors provide a corrected proof for Theorem 1 that uses information-theoretic bounds applicable to continuous random variables, rather than relying on the discrete-variable bounds from Shamir et al. (2010)? Does the claim hold for all lambda >= 1, or only under specific conditions (e.g., large lambda) as suggested by the current Appendix E.4?
- Corollary 2 Proof: Can the authors provide a valid proof for Corollary 2  and address issues raised in my review above?
- Can the authors report the LLM results (Tables 1, 2) averaged over multiple random seeds (e.g., 3 or 5), including standard errors or confidence intervals, to allow for assessment of statistical significance, and more generally provide more details on the NormalNorm baseline implementation and tuning for the LLM tasks?

---

> ### Author Response · Authors · 2025-11-20
> **Response to Reviewer 9sfx (1/3)**
>
> Dear Reviewer 9sfx,
>
> Thank you for your insightful and very careful proofreading! In the following, we provide our point-by-point response and hope our response helps address your concerns. We also look forward to the subsequent discussion which may further help solve the current issues.
>
> > **W1: Unfair Empirical Comparison in Vision Experiments: The claimed empirical superiority in vision models (Table 3) is based on an unfair comparison. Appendix F.2 reveals that the authors used different hyperparameters (learning rates, weight decays) for their proposed IBNorm method compared to the baseline methods (including BatchNorm, LayerNorm, and NormalNorm). For example, in the ResNet50/ImageNet experiment, IBNorm used a learning rate 10x smaller and weight decay 10x larger than the baselines. Claiming superior performance when the proposed method received potentially more favorable tuning invalidates the comparison. This experimental flaw makes the vision results unsubstantiated.**
>
> We thank the reviewer for raising this concern. We would like to clarify that our empirical comparison is conducted fairly across all baselines. As detailed in Appendix G, we systematically investigated multiple hyperparameter configurations, including learning rate, learning rate scheduler, weight decay, and minibatch size, for all models. We report results using the configurations that performed best for each method, ensuring that the comparison reflects the strongest performance achievable under standard tuning. Therefore, the hyperparameters used for IBNorm were chosen based on fair optimization rather than providing any advantage over the baselines.

---

> ### Author Response · Authors · 2025-11-20
> **Response to Reviewer 9sfx (2/3)**
>
> > **W2: Invalid Proof for Generalization Bound (Corollary 2): The proof attempts to justify a tighter generalization bound by invoking Theorem 2 from Kawaguchi et al. (2023).**
>
> > **W4 (1): Flawed Logic Connecting IB Value to Generalization: The argument in Section 4.3 linking the claimed higher IB value (Theorem 1) to the claimed tighter generalization (Corollary 2) is logically flawed.**
>
> > **W4 (2): Moreover, it ignores the necessary $I(\phi_l^S; S)$ term from the actual Kawaguchi et al. bound, failing to provide any analysis of how IBNorm might impact model complexity, which is essential for assessing the overall generalization bound.**
>
> > **Q2: Corollary 2 Proof: Can the authors provide a valid proof for Corollary 2 and address issues raised in my review above?**
>
> We thank the reviewer for the careful and valuable feedback. In response, we have refined the statement of Corollary 2, improved its proof (see Appendix F.5), and provided clarifications regarding: 1) model complexity terms, 2) bound structure, and 3) assumptions.
>
> **Model Complexity Term $I(\phi_l^S; S)$**
>
> We did not omit the model-related term measuring mutual information between model parameters and the training data. Given a L-layer network $f_o$, this term is represented as $C(S, f_l^o)$ depending on the cardinality of the hypothesis space $\Phi^o_l$ and the training set $S$, where $\Phi^o_l$ is the hypothesis space associated with the l-th layer $f_l^o$. Specifically, we have $C(S, f_l^o)=\ln(2|\Phi_l^o||\mathcal{Y}|/\delta)$.
>
> We discuss this term in Appendix F.5 Remark: In our experiments, all models are trained on the same dataset, so any dataset-dependent contribution to the generalization bound can be treated as invariant across different normalization schemes. Although the output weight of $f_o$ during the training process can be thought as a random variable, their information content can be controlled in the training configuration, for instance by injecting noise drawn from a chosen distribution, with variance modulated from zero (allowing, in theory, full memorization of the training set) to a level large enough that no information remains [1]. Moreover, the normalization layers considered in our study (LN, BN, and IBNorm) have identical total parameter counts. Thus, the hypothesis spaces $\Phi^\text{IB}_l$ and $\Phi^\text{S}_l$ associated with $f^\text{IB}_l$ and $f^\text{S}_l$ have the same cardinality across all layers $l$. Theoretically, by Lemma 6 in [2], we have $C(S, f^\text{IB}_l) \equiv C(S, f_l^\text{S})$ across all layers $l$, so model-complexity terms remain equivalent for $f_l^\text{IB}$ and $f_l^\text{S}$. The generalization bound of $f\_{IB}$ and $f\_S$ differ only in the representation-complexity term. Thus, our analysis focuses on how normalization affects representation-related mutual information and, in turn, generalization.
>
> Empirically, the contribution of $I(\phi_l^S; S)$ remains largely invariant under our training setup, particularly after convergence. Prior work[2] also discusses on this point: as shown in Table 2 in[2], the Pearson correlation coefficient between metrics and the generalization gap in loss of the representation related mutual information (0.384) is much greater than that of model related mutual information (0.021). Our experiments in Figure 2 also validate that most improvements in generalization arise from increased IB values.
>
> In conclusion, we have refined the statement of Corollary 2 and justified its application to compare IBNorm and variance-centric norms.
>
> **Clarification on Bound Structure**
>
> Our proof extends the results in [2] (Appendix F.4), which differs structurally from Theorem 2 in [2].
>
> Specifically, for an arbitrary layer index $l$, we have $\Delta(s)\leq Q_l$. Theorem 2 in [2] then bounds the generalization gap as $\Delta(s)\leq\min_{l \in \mathcal{D}} Q_l$. In contrast, we propose a more interpretable bound that aggregates contributions across all layers: $\Delta(s)\leq\sum_{l\in\mathcal{D}}Q_l$. This formulation is valid, as it provides an alternative way to bound the generalization gap while highlighting the cumulative effect of all layers. The detailed proof is provided in Appendix F.5.
>
> **Clarification on Assumption**
>
> The assumptions in [2] are: the setting of the input space $|\mathcal{X}|<\infty$ and the empirical internal embedding space
> $|\mathcal{M}_l|<\infty$. This is the natural setting with digital computers (e.g., using floating point) [2].
>
> In our setting, the dataset $S = (x_i, y_i)_{i=1}^M$ contains a finite number of samples $M<\infty$. Thus, these assumptions are satisfied in our theoretical analysis, and we explicitly state this in Corollary 2.
>
> [1] Alessandro Achille and Stefano Soatto. Emergence of invariance and disentanglement in deep representations. JMLR, 19(50):1–34, 2018.
>
> [2] Kenji Kawaguchi, Zhun Deng, Xu Ji, and Jiaoyang Huang. How does information bottleneck help deep learning? ICML, pp. 16049–16096. PMLR, 2023.

---

> ### Author Response · Authors · 2025-11-20
> **Response to Reviewer 9sfx (3/3)**
>
> > **W3: Invalid Proof for Higher IB Value (Theorem 1): The proof that IBNorm achieves a higher IB value (Theorem 1, Appendix E.1-E.4) relies on applying mathematical bounds (via Lemma 3, citing Shamir et al. 2010) derived for discrete random variables directly to the continuous activations of neural networks.**
>
> > **W4 (1): Flawed Logic Connecting IB Value to Generalization: The argument in Section 4.3 linking the claimed higher IB value (Theorem 1) to the claimed tighter generalization (Corollary 2) is logically flawed. It relies on the invalid proofs for both Theorem 1 and Corollary 2.**
>
> > **Q1: Theorem 1 Proof: Can the authors provide a corrected proof for Theorem 1 that uses information-theoretic bounds applicable to continuous random variables, rather than relying on the discrete-variable bounds from Shamir et al. (2010)? Does the claim hold for all lambda $>=$ 1, or only under specific conditions (e.g., large lambda) as suggested by the current Appendix E.4?**
>
> We thank the reviewer for this insightful comment. We have refined the theoretical formulation of Theorem 1 in Appendix F.2 by explicitly extending the bound to continuous random variables, based on the results in [3]. Below, we provide clarifications regarding: 1) the theoretical formulation of the bound, and 2) the use of differential entropy.
>
> **Theoretical Formulation of the Bound**
>
> We agree that the bound derived for discrete random variables directly to the continuous activations of neural networks. As shown in F.2 Proof for Lemma 3, we do not directly use the bound derived for discrete random variables in [3], since the activation random variable of neural networks $T$ is continuous. We have clarified our problem setup in Equation 3: $T$ belongs to the measurable space $\mathcal{T}$, and we have reformulated and derived the bound in the differential form based on the continuous probability density function $p_{T}(t)$. We provide a more detailed derivation in Appendix F.2.
>
> In addition, we also clarify that the original bound in [3] assumes a finite sample size, which we explicitly state in our formulation.
>
> **Arguments Regarding Differential Entropy**
>
> The differential-entropy arguments in Lemma 5 are valid and rigorous. The analysis focuses on the terms related to $\tilde{T}\_1$, because we are interested in bounding the gaps $\Delta\_{S}(\tilde{T}\_{IB};\tilde{T}\_1^{\*})$ and $\Delta\_{S}(\tilde{T}\_{s};\tilde{T}\_{1}^{\*})$. This leads directly to our theoretical conclusion that $\widehat{\text{IB}}\_{1}\geq \widehat{\text{IB}}\_{2}$.
>
> Importantly, the proof of Lemma 5 does not rely on the compression hyperparameter $\lambda$ being large. Instead, it is based on the bounded compression property of the IBNorm compression operator: $s_\lambda$ is a Lipschitz map with Lipschitz constant $L \le 1$ because $|\det J_{s_\lambda}(Z)| = \prod_{i=1}^d f_\lambda'(|Z_i - \mu|) \le \prod_{i=1}^d \alpha_{\lambda}\leq 1$ by the bounded compression property of $f_{\lambda}$. In addition, the proof of Lemma 5 relies on the sufficiently small choice of $\delta$ stated in the remark in Appendix F.4, which ensures a tighter bound. Consequently, the claim of Lemma 5 holds for all $\lambda\geq 1$. The bound does not require $\lambda$ to be large; it solely depends on the Lipschitz and bounded-compression properties of the IBNorm operator. We have clarified this point in the updated version to avoid any ambiguity regarding the role of $\lambda$ in the proof.
>
> [3] Ohad Shamir, Sivan Sabato, and Naftali Tishby. Learning and generalization with the information bottleneck. Theoretical Computer Science, 411(29-30):2696–2711, 2010.
>
> ---
>
> > **W5: Lack of rigor in LLM gains reporting: While the LLM results appear promising, the magnitude of gains over baselines (especially the internally-run NormalNorm) is surprisingly large..**
>
> > **Q3: Can the authors report the LLM results (Tables 1, 2) averaged over multiple random seeds (e.g., 3 or 5), including standard errors or confidence intervals, to allow for assessment of statistical significance, and more generally provide more details on the NormalNorm baseline implementation and tuning for the LLM tasks?**
>
> Thank you for your valuable feedback. We agree that reporting statistical variability is important for assessing the significance of improvements. Due to computational constraints, we have conducted three independent runs for LLaMA-60M, LLaMA-130M, and GPT-2 Small for all baselines and our best IBNorm variant, as reported in Appendix D.1. Tables 8 and 9 demonstrate that IBNorm consistently outperforms baseline normalization methods on LLM Leaderboard I and II.
>
> We acknowledge that extending repeated runs to all models would further strengthen the analysis. We will include the mean $\pm$ standard deviation for all language model experiments.

---

> > ### Author Response · Authors · 2025-11-27
> >
> > Dear Reviewer 9sfx,
> >
> > We believe that our response has addressed your concerns. As the discussion period concludes in seven days, we kindly ask that you review our response and, if you find it has comprehensively resolved your issues, please consider revising your scores accordingly. If you have any additional questions or require clarification, please do let us know.

---

### Official Review · Reviewer_P5Gw · 2025-10-28

**Soundness:** 2
**Presentation:** 3
**Contribution:** 3
**Rating:** 6
**Confidence:** 4

**Summary:**

This paper proposes a new normalization method called IBNorm, which introduces an information bottleneck  constraint into the normalization process to balance feature information preservation and noise suppression.  Unlike traditional normalization methods such as LayerNorm and RMSNorm that standardize feature  distributions by mean and variance, IBNorm explicitly regulates the mutual information between features and  task-relevant representations. By reformulating normalization from an information-theoretic perspective, the  method aims to allow adaptive information flow control during feature extraction. Experiments are conducted on multiple language models (LLaMA-60M/130M/350M and GPT-2-12M/355M).  Results show that IBNorm achieves better average performance on Leaderboard II, particularly on several mid scale models, even though some concerns on experimental setup and analysis/explanation remains.

**Strengths:**

1. The paper provides an theoretical viewpoint by integrating  the information bottleneck principle into normalization design,  even though inspired from the previous paper (NormalNorm, ICML 2025), contributing to a deeper understanding  of how normalization can regulate information flow in representation learning.
2. The information-theoretic reformulation of normalization is  well motivated, and the overall structure and derivation are easy to follow.
3. Empirical validation on multiple models – The method is evaluated on several LLaMA and GPT-2  variants, covering different model sizes and task categories. In several cases (e.g., LLaMA-350M and  GPT-2-355M on Leaderboard II), IBNorm exhibits performance improvements over LayerNorm and  RMSNorm.

**Weaknesses:**

1. The overall idea and contribution is incremental, especially the writing is mostly following the presentation of the previous paper (NormalNorm, ICML 2025), e.g., the Information Bottleneck idea of normalization and the description of algorithm (NormalNorm, ICML 2025). Besides, this paper misses a bunch of papers to discuss and  compare.,  e.g., the whitening method in normal supervised learning [1,2]  and self-superverses learning [3,4]. I think this paper should compare to IterNorm [2] method in experiments, which is in native to obtain the same functionality as the paper and (NormalNorm) and has a similar formulation in algorithm.

2. I have concerns on the experiments.

   (1) The hyper-parameters  $\lambda$ is  sensitive. This paper  uses hyper-parameters search in several experiments, and the results on reported over different hyper-parameters (e.g., $\lambda$) on different methods/tasks. This remains the concerns on the practicability.

   (2) **Experimental setups.**  In the results of ResNet on ImageNet of Table 3, it is weired the BatchNorm has only 76.15% accuracy when training over 200 epochs (From the description in  supplementary materials). It is true that the basic ResNet (BatchNorm) training (100 epochs) has a performance around 76.15%.  However, It is easy to obtain 77%+ accuray when training with 200 epochs from the results reported in previous work. Does this results are cited from previous paper or reproduced?

   (2) **Inconsistent experimental results**. IBNorm does not consistently outperform baselines across tasks. In Table 2, it performs worse on BBH,  GPQA, and MMLU-PRO, indicating limited generalization ability across reasoning and knowledge intensive benchmarks. The paper does not analyze or discuss these cases, weakening its empirical  justification.

   (3) **Questionable numerical gain calculations**. The reported relative improvements in Section 5.1 appear inconsistent with actual table values. For example, the authors claim: “On LLaMA-350M, IBNorm-L achieves 0.2140 (Leaderboard II), with gains of 6.84% over LN (0.2003) and  9.51% over RMSNorm (0.2010).” However, the value in Table 1 is 0.2116, yielding gains of only 5.64% and 5.27%, respectively. . This discrepancy suggests either computational inconsistency, which should be clarified.

   (4) **Lack of statistical significance analysis** .Results are reported as single averages over tasks without standard deviation, variance, or confidence  intervals. Given the heterogeneous scoring scales of tasks in Leaderboard II, it is difficult to determine  whether the improvements are statistically meaningful. Repeated runs and reporting of mean ± std are  strongly recommended.


   (5).**Over-claiming in interpretation**. The paper frequently uses strong statements such as “consistently surpasses variance-centric methods”  and “achieves significant improvement.” However, given that some improvements are minor and  performance decreases occur in several tasks, such claims are not fully supported by the results and  should be moderated.


**Ref:**

1.Decorrelated Batch Normalization. CVPR, 2018.

2.Iterative Normalization: Beyond Standardization towards Efficient Whitening. CVPR, 2019.

3.On feature decorrelation in self-supervised learning. ICCV, 2021.

4.Modulate Your Spectrum in Self-Supervised Learning. ICLR, 2024.

**Questions:**

1. Concerns 2. (2) on the experimental setups.
2. Clarification of improvement calculation How are the reported performance gains (e.g., “+6.84%” and “+9.51%”) computed?  Please provide the  explicit formula.

3.  Failure case explanation Why does IBNorm perform worse on BBH, GPQA, and MMLU-PRO? Could this be related to task  characteristics (e.g., reasoning vs. factual retrieval) or model-scale sensitivity?

---

> ### Author Response · Authors · 2025-11-20
> **Response to Reviewer P5Gw (1/3)**
>
> Dear Reviewer P5Gw,
>
> Thank you for your insightful comments and very careful proofreading! In the following, we provide our point-by-point response and hope our response helps address your concerns. We also look forward to the subsequent discussion which may further help solve the current issues.
>
> > **W1: The overall idea and contribution is incremental, especially the writing is mostly following the presentation of the previous paper (NormalNorm, ICML 2025), e.g., the Information Bottleneck idea of normalization and the description of algorithm (NormalNorm, ICML 2025). Besides, this paper misses a bunch of papers to discuss and compare., e.g., the whitening method in normal supervised learning [1,2] and self-superverses learning [3,4]. I think this paper should compare to IterNorm [2] method in experiments, which is in native to obtain the same functionality as the paper and (NormalNorm) and has a similar formulation in algorithm.**
>
> Thank you for your insightful comment. We clarify the distinction of IBNorm from NormalNorm [5] and prior whitening methods [1–4], and we have included a discussion of these prior works in our paper. We also provide extended results on ResNet experiments.
>
> **Distinction from NormalNorm**
>
> NormalNorm [5] does not explicitly analyze normalization under the Information Bottleneck (IB) principle. It positions normalization within a mutual information framework and derives Gaussian representations for expressiveness without a theoretical analysis of the information bottleneck. However, IBNorm studies the question: *can normalization be re-designed not only to stabilize training but also to shape representations toward sufficiency and generalization?* in a principle way (see Section 4.1 and 4.2).
>
> As detailed in Appendices A and B, the key methodological distinction lies in how NormalNorm and IBNorm shape hidden activation distributions:
>
> - NormalNorm maximizes the entropy of hidden activations by applying a power transform after standardization. As discussed in Section 2,  this approach faces three key limitations: it does not optimize mutual information with targets, incurs overhead in estimating power parameters, and introduces stochasticity that can destabilize training.
>
> - In contrast, IBNorm follows the IB principle, shaping activation distributions via a compression operation applied before standardization.
>
> **Distinction from Whitening Methods**
>
> Prior works in whitening [1,2] and representation learning [3,4] improve batch normalization primarily from a whitening perspective, aiming to decorrelate features and stabilize training. For example, Iterative Normalization (IterNorm) [2] efficiently approximates full whitening using Newton’s iterations, avoiding costly eigen-decomposition, while traditional whitening methods [1] perform exact decorrelation of activations. Similarly, in self-supervised representation learning [3,4], whitening is used to reduce feature redundancy and improve downstream task performance.
>
> Our proposed IBNorm, in contrast, goes beyond whitening as a stabilization technique. It explicitly analyzes the expressiveness of representations using the IB principle, which seeks to compress irrelevant information in the hidden activations while preserving task-relevant information. Rather than only decorrelating features, IBNorm shapes the hidden activation distributions via a principled compression operation before standardization, which directly influences the mutual information between inputs and learned representations.
>
> **Extended Results on Vision Experiments**
>
> We also compare with Iterative Normalization (IterNorm) [2], which uses Newton's iterations to efficiently achieve whitening without eigen-decomposition. Table 14 shows accuracy (\%) on CIFAR-10 with ResNet18. BatchNorm results are reported since they significantly outperform LayerNorm on CNNs. We observe that while IterNorm surpasses NormalNorm, it still underperforms our proposed IBNorm.
>
> [1] Decorrelated Batch Normalization. CVPR, 2018.
>
> [2] Iterative Normalization: Beyond Standardization towards Efficient Whitening. CVPR, 2019.
>
> [3] On feature decorrelation in self-supervised learning. ICCV, 2021.
>
> [4] Modulate Your Spectrum in Self-Supervised Learning. ICLR, 2024.
>
> [5] On the Importance of Gaussianizing Representations. ICML, 2025.

---

> > ### Author Response · Authors · 2025-11-20
> > **Response to Reviewer P5Gw (2/3)**
> >
> > > **W2 (1) The hyper-parameters
> >  is sensitive. This paper uses hyper-parameters search in several experiments, and the results on reported over different hyper-parameters ($\lambda$) on different methods/tasks. This remains the concerns on the practicability.**
> >
> > Thank you for this insightful comment. IBNorm is robust to the choice of $\lambda$, exhibiting a stable optimal range across different models and tasks. In our experiments, we use $\lambda=4$ for all IBNorm-L and IBNorm-T variants, and $\lambda=3$ for IBNorm-S. We provide extensive $\lambda$-sensitivity analyses in Appendix D.3.
> >
> > **Ablation Study on $\lambda$**
> >
> > We include detailed ablation studies on both LLaMA and GPT-2 architectures:
> >
> > - LLaMA-60M results are reported in Table 11.
> >
> > - GPT-2 Small (124M) results are reported in Table 12.
> >
> > Across all experiments, we find that for IBNorm-L, IBNorm-T, and IBNorm-S, values of $[3,5]$ yield consistently strong performance, with no significant fluctuations within this range. This stability across two architectures suggests that the method is not highly sensitive to the precise choice of $\lambda$.
> >
> > **$\lambda$–Information Bottleneck Curve**
> >
> > To further analyze how $\lambda$ affects representation compression from an IB-theoretic perspective, we provide an extended $\lambda$–IB curve analysis for LLaMA-130M in Appendix D.3. We demonstrate the evolution of predictive information $\hat I(Y;T_l)$, task–nuisance information $\hat I(T_{l-1};T_l)$, and the token-level IB value during training of Llama-130M on C4, using IBNorm-T across different hyperparameters $\lambda=0.5,4,8$.
> >
> > The results (Figure 3) reveal a clear trend:
> >
> > - Under-compression ($\lambda=0.5$) fails to sufficiently suppress task-irrelevant information, leading to suboptimal IB tradeoffs.
> >
> > - Over-compression ($\lambda=8$) starts to suppress task-relevant information, harming predictive performance.
> >
> > - Moderate compression ($\lambda=4$) achieves the best IB value, effectively removing nuisance factors while preserving task-useful information.
> >
> > These findings provide a principled explanation for the empirically observed optimal $\lambda$.
> >
> > In conclusion, 1) IBNorm is robust to the choice of $\lambda$, with a stable optimal range across models. 2) The effect of $\lambda$ is theoretically interpretable through the IB framework, which explains why moderate compression is most effective.
> >
> > ---
> >
> > > **W2 (2) Experimental setups. In the results of ResNet on ImageNet of Table 3, it is weired the BatchNorm has only 76.15\% accuracy when training over 200 epochs (From the description in supplementary materials). It is true that the basic ResNet (BatchNorm) training (100 epochs) has a performance around 76.15\%. However, It is easy to obtain 77\%+ accuray when training with 200 epochs from the results reported in previous work. Does this results are cited from previous paper or reproduced?**
> >
> > > **Q1 Concerns 2. (2) on the experimental setups.**
> >
> > Thank you for your comment. The 76.15\% ResNet-50 (BatchNorm) accuracy reported in Table 3 is cited from NormalNorm, and we strictly follow their experimental setup as detailed in Appendix G.2. Under this training configuration, we also reproduce their reported result. While prior works often achieve 77\%+ accuracy with 200-epoch training, those results typically rely on stronger data augmentation or more advanced training strategies, which NormalNorm does not use. To ensure a fair and consistent comparison, we adopt the same settings as NormalNorm.
> >
> > ---
> >
> > > **W2 (3) Inconsistent experimental results. IBNorm does not consistently outperform baselines across tasks. In Table 2, it performs worse on BBH, GPQA, and MMLU-PRO, indicating limited generalization ability across reasoning and knowledge intensive benchmarks. The paper does not analyze or discuss these cases, weakening its empirical justification.**
> >
> > > **Q3: Failure case explanation Why does IBNorm perform worse on BBH, GPQA, and MMLU-PRO? Could this be related to task characteristics (reasoning vs. factual retrieval) or model-scale sensitivity?**
> >
> > We thank the reviewer for highlighting these observations. We acknowledge that IBNorm does not consistently outperform all baselines across every task in LLM Leaderboard I and II, and we have included a discussion of these failure cases in Appendix D.
> >
> > Upon careful examination, the tasks where IBNorm underperforms, such as BBH, GPQA, and MMLU-PRO, are highly challenging knowledge-intensive benchmarks, with questions crafted by domain experts across diverse fields. This contrasts with tasks such as factual retrieval, where IBNorm consistently improves performance.
> >
> > IBNorm is designed to enhance representation expressiveness via the information bottleneck principle, improving generalization and robustness. However, its impact is naturally limited on reasoning-heavy tasks. Importantly, IBNorm contributes to improved generalization ability of the model by shaping more informative hidden representations during training.

---

> > > ### Author Response · Authors · 2025-11-20
> > > **Response to Reviewer P5Gw (3/3)**
> > >
> > > > **W2 (4) Questionable numerical gain calculations. The reported relative improvements in Section 5.1 appear inconsistent with actual table values. For example, the authors claim: “On LLaMA-350M, IBNorm-L achieves 0.2140 (Leaderboard II), with gains of 6.84\% over LN (0.2003) and 9.51\% over RMSNorm (0.2010).” However, the value in Table 1 is 0.2116, yielding gains of only 5.64\% and 5.27\%, respectively. This discrepancy suggests either computational inconsistency, which should be clarified.**
> > >
> > > > **Q2 Clarification of improvement calculation How are the reported performance gains (e.g., “+6.84\%” and “+9.51\%”) computed? Please provide the explicit formula.**
> > >
> > > Thank you for pointing this out. We apologize for the typos in the previous version and have corrected them.
> > >
> > > On LLaMA-130M, *IBNorm-T* reaches 2.03\% (Leaderboard I) and 9.51\% (Leaderboard II) over RMSNorm.
> > >
> > > Relative Improvement Calculation:
> > > $$(0.2970-0.2911)/0.2911 \times100=2.03$$
> > > $$(0.2130-0.1945)/0.1945 \times100=9.51$$
> > >
> > > On LLaMA-350M, *IBNorm-T* reaches 0.3101 (Leaderboard I) and 0.2140 (Leaderboard II), \textit{i.e.} 1.27\% and 6.84\% over LN, and 2.51\% and 6.46\% over RMSNorm.
> > >
> > > Relative Improvement Calculation:
> > > $$(0.3101-0.3062)/0.3062 \times100=1.27$$
> > > $$(0.2140-0.2003)/0.2003 \times100=6.84$$
> > > $$(0.3101-0.3025)/0.3025 \times100=2.51$$
> > > $$(0.2140-0.2010)/0.2010 \times100=6.46$$
> > >
> > > ---
> > >
> > > > **W2 (5) Lack of statistical significance analysis. Results are reported as single averages over tasks without standard deviation, variance, or confidence intervals. Given the heterogeneous scoring scales of tasks in Leaderboard II, it is difficult to determine whether the improvements are statistically meaningful. Repeated runs and reporting of mean ± std are strongly recommended.**
> > >
> > > Thank you for your valuable feedback. We agree that reporting statistical variability is important for assessing the significance of improvements. Due to computational constraints, we have conducted three independent runs for LLaMA-60M, LLaMA-130M, and GPT-2 Small for all baselines and our best IBNorm variant, as reported in Appendix D.1. Tables 8 and 9 demonstrate that IBNorm outperforms baseline normalization methods on LLM Leaderboard I and II.
> > >
> > > We acknowledge that extending repeated runs to all models would further strengthen the analysis. We will include the mean $\pm$ standard deviation for all language model experiments.
> > >
> > > ---
> > >
> > > > **W2 (6) Over-claiming in interpretation. The paper frequently uses strong statements such as “consistently surpasses variance-centric methods” and “achieves significant improvement.” However, given that some improvements are minor and performance decreases occur in several tasks, such claims are not fully supported by the results and should be moderated.**
> > >
> > > Thank you for your comment. We have revised our phrasing and wording throughout the paper to use more precise and neutral statements.

---

### Official Review · Reviewer_mLmV · 2025-11-01

**Soundness:** 3
**Presentation:** 3
**Contribution:** 2
**Rating:** 4
**Confidence:** 3

**Summary:**

The paper proposes IBNorm, a normalization method inspired by the Information Bottleneck (IB) principle. Unlike traditional variance-centric normalization techniques (such as BatchNorm, LayerNorm), IBNorm introduces a compression operation to preserve task-relevant information while suppressing irrelevant variability in activations. The paper presents both theoretical guarantees and empirical results, showing that IBNorm outperforms conventional methods across language and vision models. The authors argue that IBNorm provides better generalization by enhancing the quality of learned representations, making it theoretically sound and empirically effective.

While the paper's contributions are valuable, particularly in terms of theoretical guarantees and empirical validation, it lacks sufficient exploration of the novelty of its compression functions compared to existing methods. Moreover, a deeper comparison with other IB-inspired methods and clearer discussions on practical aspects (such as mutual information estimation) would further strengthen the paper. Overall, the paper is a solid contribution but requires some refinements to fully justify the novelty and unique benefits of the proposed method.

**Strengths:**

1. The paper introduces an application of the information bottleneck to normalization. By integrating compression operations into the normalization process, the paper presents a significant departure from traditional methods that focus solely on statistical normalization (mean and variance).
2. The paper presents a clear theoretical foundation and solid empirical validation. The authors present rigorous proofs that IBNorm achieves a higher IB value compared to variance-centric methods and demonstrate its better generalization performance. The extensive experiments across multiple architectures (LLaMA, GPT-2, ResNet, ViT) strengthen the paper's claims and show the practical benefits of IBNorm.
3. The paper is clearly written, with a logical flow that introduces the problem, motivates the new approach, and carefully explains the theoretical and experimental results. The methodology section is particularly well-explained, making complex concepts like the IB principle and the compression operation accessible.

**Weaknesses:**

1. Limited contribution in context of existing IB works. In most work based on the Information Bottleneck principle, compression operations (such as nonlinear functions like tanh [1] and kernel-based function [2], or explicit compression losses like VIB [3] and SPC [4]) are often applied alongside normalization operations within neural networks. The paper claims to introduce an IB perspective to normalization, but it essentially introduces an additional compression operation into networks already using normalization. This contribution appears to be limited. While IBNorm introduces new compression functions and combinations with normalization layers, the paper does not sufficiently explore whether these combinations provide any significant advantage over existing work. The novelty of combining IB with normalization is not fully established because similar operations are already in use.

- [1] On the Information Bottleneck Theory of Deep Learning. IEEE Information Theory Workshop 2015.
- [2] Nonlinear information bottleneck. Entropy 2019.
- [3] Deep variational information bottleneck. ICLR 2017.
- [4] Structured Probabilistic Coding. ACL 2024.

2.  Fail to justify the unique advantages of the compression functions in IBNorm. The proposed compression functions (IBNorm-S, IBNorm-L, and IBNorm-T) are based on nonlinear functions that exhibit compression behavior, specifically functions like tanh. A prior work [1] already shows that saturating nonlinearities like tanh induce compression in neural activations from the perspective of the information bottleneck. The compression operations introduced in IBNorm are thus special cases of these well-known nonlinearities. The paper fails to sufficiently justify the unique advantages of these specific functions over general nonlinear functions like tanh. Further empirical or theoretical validation of the superiority of these three heuristic functions would be helpful to establish their distinct contribution.


3. Comparisons with existing information bottleneck methods. While the paper proves that IBNorm achieves a higher IB value than variance-centric methods like LayerNorm and BatchNorm, it does not compare IBNorm directly with other IB-inspired methods such as explicit compression losses (e.g., VIB loss, MINE loss [5] ) or other implicit compression functions like tanh. Since IBNorm itself uses compression as its central mechanism, comparing it to existing IB methods is crucial to validate whether it achieves a truly higher IB value and whether this additional compression operation provides unique advantages in improving generalization and representation learning.

- [5] Mutual information neural estimation. ICML 2018.

4. Lack of references for IB-guided normalization objective. The IB-guided normalization objective described between lines 186-195 is essentially a layer-wise IB loss, which is discussed in the paper [1] on information bottleneck and layer characteristics. The paper lacks references to this foundational work and does not highlight how its approach differs or extends the existing layer-wise IB loss formulation. Including these references would strengthen the theoretical foundation of the paper.

5. Inaccurate statement about mutual information estimation. The paper states in lines 203-205 that "Directly optimizing Eqn.(5) is infeasible due to costly mutual information estimation over the unknown joint distribution." This description is not entirely accurate. While variational estimation of mutual information does require assumptions about the distribution, there are alternative methods (e.g., MINE, using additional networks to estimate mutual information) that do not require explicit assumptions about the distribution. Clarifying this point and discussing these alternatives would improve the technical accuracy of the paper.

**Questions:**

1. Clarify the novelty beyond existing IB-based compression approaches with normalization.
- Many prior works already combine Information Bottleneck (IB) principles with normalization or compression mechanisms, either implicitly (via nonlinear activations like tanh) or explicitly (via compression losses such as VIB).
- Could the authors clarify what fundamentally differentiates IBNorm from these existing approaches beyond introducing new compression functions (IBNorm-S/L/T)?
- Specifically, what are the unique advantages (theoretical or empirical) of IBNorm’s bounded compression functions compared to standard nonlinearities that already induce information compression?

2. Justification of the compression functions (IBNorm-S, IBNorm-L, IBNorm-T).
- The proposed heuristic functions are variants of saturating nonlinearities that inherently compress information in the work [1].
- Could the authors provide a quantitative analysis or visualization showing how these three functions differ in their information compression characteristics (e.g., in terms of mutual information, entropy, or Fisher information)?
- Additionally, what design principles guided the specific forms of IBNorm-S/L/T? Were they chosen empirically or derived from any theoretical insight?

3. Comparisons with other IB variants and explicit compression losses. Since IBNorm’s main contribution is introducing an IB-inspired compression step, it would be very informative to compare it against explicit IB-based training methods such as VIB or MINE loss applied alongside normalization. Could the authors comment on whether IBNorm would still outperform such methods in terms of the achieved IB value or generalization?

4. Impact of λ across models and tasks. The relationship between λ and model performance may vary across architectures and data modalities. Could the authors provide more analysis (e.g., λ-sensitivity curves or cross-task validation) to better understand how λ affects representation informativeness and generalization?


5. Scalability to larger models and practical deployment. How does IBNorm scale to larger LLMs (e.g., 7B–70B parameters)? Are there computational trade-offs (e.g., increased training time, numerical stability issues) when applying IBNorm at scale? Clarifying these practical considerations would be useful for understanding IBNorm’s potential in real-world.

6. Under the IB framework, it is difficult to determine in advance how close we are to optimal compression, which can easily lead to the over-compression or under-compression issue of representations [4,6]. Has this work considered the problems in the compression process, which may have significant implications for layer-wise IB?
- [6] The conditional entropy bottleneck. Entropy 2020.

---

> ### Author Response · Authors · 2025-11-20
> **Response to Reviewer mLmV (1/7)**
>
> Dear Reviewer mLmV,
>
> Thank you for the insightful and valuable comments! In the following, we provide our point-by-point response and hope our response helps address your concerns. We also look forward to the subsequent discussion which may further help solve the current issues.
>
> > **W1: Limited contribution in context of existing IB works.**
>
> > **Q1 (1): Many prior works already combine Information Bottleneck (IB) principles with normalization or compression mechanisms, either implicitly (via nonlinear activations like tanh) or explicitly (via compression losses such as VIB).**
>
> Thank you for your insightful comment. We clarify that, to the best of our knowledge, existing IB works primarily focus on estimating mutual information or optimizing explicit IB objectives, rather than analyzing or redesigning normalization layers. In contrast, IBNorm is the first framework that systematically studies normalization from an IB-theoretic perspective and introduces a family of normalization methods that internalize the IB principle directly within the normalization operation. This perspective enables improved generalization without relying on auxiliary IB losses or additional modules. We elaborate on the distinctions below.
>
> **Distinction in Problem Setting**
>
> Work[1] explores the IB theory of deep learning and empirically shows how hidden layers naturally compress task-irrelevant information. However, these analyses do not address how normalization contributes to the compression process.
>
> Methods[2, 3, 5] focus on estimating mutual information by parameterizing the IB principle with neural MI estimators and then guiding the training using the explicit IB objective. These approaches typically rely on encoder–decoder architectures (e.g., VAEs) and require training explicit loss functions specialized for MI estimation.
>
> Similarly, SPC[4] introduces an IB-inspired coding mechanism, but it is limited to encoder-only models and does not examine normalization layers.
>
> Crucially, none of these works analyze the role of normalization within deep models from the IB viewpoint. IBNorm fills this gap by providing the first theoretical analysis of normalization under the IB principle.
>
> **Difference in the Methods**
>
> Existing IB methods[2, 3, 5] rely on additional mutual-information estimators and introduce explicit IB losses during training. These components introduce extra hyperparameters, increase computational cost, and may lead to unstable optimization. Specifically, these approaches face two key limitations when applied to deep learning models, such as autoregressive LLMs, in practice:
>
> - MI Neural Estimator Architecture and Computational Overhead: These methods require additional MI neural estimators alongside the main network, significantly increasing heavy inference computational cost and complicating optimization. In addition, designing such neural estimators is particularly difficult for autoregressive LLMs due to their sequential decoding characteristics.
>
> - Sample Complexity: Accurate MI estimation, as noted in MINE, requires large sample sizes, which is challenging for models with long-context inputs and requires heavy training cost.
>
> SPC[4] incorporates an IB objective but is constrained to encoder-specific architectures, limiting its generality.
>
> In contrast, IBNorm incorporates the IB principle within the normalization operation. It requires no auxiliary MI estimators, no additional IB-based training losses, and no architectural constraints. The compression operation introduced in IBNorm is not an external module but a theoretically grounded reformulation of normalization that increases the IB value of intermediate representations while adding minimal computational overhead.
>
> We highlight the key advantages of IBNorm here:
>
> - Systematic Theoretical Analysis: Unlike[1], which offers only empirical observations, we provide a theoretical analysis of widely used variance-centric normalization methods. This analysis motivates a principled mechanism for designing normalization layers that follow IB principle.
>
> - Computational Efficiency: Unlike[2, 3, 4, 5], IBNorm does not rely on computationally heavy MI neural estimators or explicit IB regularizers, making it more efficient and stable during training.
>
> - Compatibility: IBNorm is model-agnostic and can be directly integrated into Transformers and CNNs without modifying the training pipeline. In contrast, SPC is limited to encoder-based architectures such as BERT.
>
> In conclusion, to our best knowledge, IBNorm is the first work to apply IB theory specifically to the design of normalization layers. Our framework provides the first theoretical analysis on normalization from the IB principle.
>
> [1] On the Information Bottleneck Theory of Deep Learning. ICLR 2018.
>
> [2] Nonlinear information bottleneck. Entropy 2019.
>
> [3] Deep variational information bottleneck. ICLR 2017.
>
> [4] Structured Probabilistic Coding. AAAI 2024.
>
> [5] Mutual information neural estimation. ICML 2018.

---

> > ### Author Response · Authors · 2025-11-20
> > **Response to Reviewer mLmV (2/7)**
> >
> > > **W2: Fail to justify the unique advantages of the compression functions in IBNorm.**
> >
> > > **Q1: Clarify the novelty beyond existing IB-based compression approaches with normalization.**
> >
> > > **Q2 (1): The proposed heuristic functions are variants of saturating nonlinearities that inherently compress information in the work [1].**
> >
> > Thank you for your valuable comments. We provide clarification on the distinction between our proposed compression functions and the saturating nonlinearities discussed in prior work [1] as follows.
> >
> > **Distinction in Methodological Role and Placement**
> >
> > Prior work [1] shows that saturating nonlinearities such as tanh can induce compression when used as activation functions. However, this observation does not address the role of normalization layers, nor how compression applied within normalization affects the information bottleneck (IB) behavior of deep models.
> >
> > Importantly, activation and normalization are conceptually and structurally distinct:
> >
> > - In modern LLMs (e.g., LLaMA, GPT), nonlinear activations occur only in the gated branch of the MLP, and do not influence the representations within each transformer block. In these architectures, inserting tanh as an "activation" would not propagate compression throughout the model, since the gating branch represents only one component of the block.
> >
> > In contrast:
> >
> > - IBNorm applies compression directly the normalization layer, which is invoked throughout the model—after each attention block and MLP block in LLMs. This placement ensures that compression systematically modulates the IB value of the representation at every layer, producing effects that cannot be achieved by modifying the activation function alone.
> >
> > Thus, even if tanh exhibits compression as an activation function, it does not serve the same purpose, does not affect the same parts of the network, and does not shape representation statistics in the same way as normalization-based compression in IBNorm.
> >
> > **Distinction in Theoretical Analysis**
> >
> > Unlike tanh, which is only empirically shown to induce compression in [1], the functions IBNorm-S, IBNorm-L, and IBNorm-T are explicitly derived from an IB-theoretic analysis of normalization. Our analysis demonstrates how different IB-motivated compression operations shape higher-order statistics and directly align normalization behavior with the IB principle. This provides theoretical grounding that general nonlinearities such as tanh do not offer.
> >
> > In summary, although both IBNorm and tanh involve nonlinear compression, their roles, motivations, theoretical foundations, and functional effects differ fundamentally. Prior work [1] does not analyze normalization nor provide IB-guided mechanisms for designing normalization operations. IBNorm is the first to fill this gap by introducing IB-driven normalization functions that systematically enhance model generalization.
> >
> > We incorporate these clarifications to strengthen the justification of our compression designs in the revised manuscript.

---

> > > ### Author Response · Authors · 2025-11-20
> > > **Response to Reviewer mLmV (3/7)**
> > >
> > > > **W3: Comparisons with existing information bottleneck methods.**
> > >
> > > > **Q1 (1): Many prior works already combine Information Bottleneck (IB) principles with normalization or compression mechanisms, either implicitly (via nonlinear activations like tanh) or explicitly (via compression losses such as VIB).**
> > >
> > > > **Q3: Comparisons with other IB variants and explicit compression losses.**
> > >
> > > Thank you for your insightful comment. We clarify the distinction between IBNorm and existing IB-inspired methods and provide both methodological and empirical justifications below.
> > >
> > > **Distinction from Explicit IB Methods**
> > >
> > > Explicit IB methods, such as VIB loss or MINE loss, introduce external regularization to control the IB value by estimating mutual information during training. However, these components introduce extra hyperparameters, increase computational cost, and may lead to unstable optimization. Specifically, these approaches face two key limitations when applied to deep learning models, such as autoregressive LLMs, in practice:
> > >
> > > - MI Neural Estimator Architecture and Computational Overhead:
> > > These methods require additional MI neural estimators alongside the main network, significantly increasing heavy inference computational cost and complicating optimization. In addition, designing such neural estimators is particularly difficult for autoregressive LLMs due to their sequential decoding characteristics.
> > >
> > > - Sample Complexity:
> > > Accurate MI estimation, as noted in MINE [5], requires large sample sizes, which is challenging for models with long-context inputs and requires a heavy training cost.
> > >
> > > In contrast, IBNorm integrates the IB principle directly within the normalization layer, without relying on auxiliary MI estimators or explicit IB losses. This design makes IBNorm computationally efficient and compatible with existing architectures, unlike VIB or MINE.
> > >
> > > For these reasons, our primary comparison baselines are widely used variance-centric normalization methods (BatchNorm, LayerNorm), which share the same computational constraints and functional role as IBNorm.
> > >
> > > **Empirical Validation of IBNorm**
> > >
> > > We validate our theoretical claims (Theorem 1 and Corollary 2) through experiments in Figure 2:
> > >
> > > - IB Value Evaluation:
> > > We train LLaMA-130M and GPT-2 small with frozen parameters except the normalization layers on C4 and OpenWebText, respectively. As shown in Fig.2a, IBNorm consistently achieves higher token-level IB values ($\beta=1$) compared to variance-centric normalization methods. This confirms that the compression operation in IBNorm directly enhances information-theoretic optimality.
> > >
> > > - Downstream Task Performance:
> > > Using the same experimental setup, we evaluate models on LLM Leaderboard II. Fig.2b shows that IBNorm improves generation quality on downstream tasks, demonstrating that higher IB values correspond to better generalization and representation learning.
> > >
> > > In summary, while explicit IB methods like VIB or MINE rely on heavy MI estimators and additional loss functions, IBNorm achieves superior IB-theoretic and downstream performance without external MI estimators or explicit IB losses, making it a more practical and efficient approach. Our empirical results confirm that the compression embedded in normalization layers is sufficient to improve generalization and representation learning, validating the unique advantage of IBNorm.

---

> > > > ### Author Response · Authors · 2025-11-20
> > > > **Response to Reviewer mLmV (4/7)**
> > > >
> > > > > **W4: Lack of references for IB-guided normalization objective.**
> > > >
> > > > Thank you for your comment. We had cited the paper [1] in L194.
> > > >
> > > > The prior work [1] explores the IB theory of deep learning and empirically shows how hidden layers naturally compress task-irrelevant information. However, these analyses do not address how normalization contributes to the compression process. However, it doesn't provide a explicit form of the layer-wise IB loss. Unlike [1], which offers only empirical observations, we provide a theoretical analysis of widely used variance-centric normalization methods. This analysis motivates a principled mechanism for designing normalization layers that follow IB principle.
> > > >
> > > > ---
> > > >
> > > > > **W5: Inaccurate statement about mutual information estimation.**
> > > >
> > > > Thank you for your comment. Our statement was meant to emphasize the computational and optimization challenges associated with directly optimizing Eq.(5), rather than to claim that such estimation is entirely infeasible. We have rephrased this claim and clarified this point in Section 4.1. As we discussed in response to W3, we clarify the distinction between IBNorm and existing IB-inspired methods and provide the justifications below.
> > > >
> > > > **Distinction from Explicit IB Methods**
> > > >
> > > > Explicit IB methods, such as VIB loss or MINE loss, introduce external regularization to control the IB value by estimating mutual information during training. However, these components introduce extra hyperparameters, increase computational cost, and may lead to unstable optimization. Specifically, these approaches face two key limitations when applied to deep learning models, such as autoregressive LLMs, in practice:
> > > >
> > > > - MI Neural Estimator Architecture and Computational Overhead:
> > > > These methods require additional MI neural estimators alongside the main network, significantly increasing heavy inference computational cost and complicating optimization. In addition, designing such neural estimators is particularly difficult for autoregressive LLMs due to their sequential decoding characteristics.
> > > >
> > > > - Sample Complexity:
> > > > Accurate MI estimation, as noted in MINE [5], requires large sample sizes, which is challenging for models with long-context inputs and requires heavy training cost.
> > > >
> > > > In contrast, IBNorm integrates the IB principle directly within the normalization layer, without relying on auxiliary MI estimators or explicit IB losses. This design makes IBNorm computationally efficient and compatible with existing architectures, unlike VIB or MINE.
> > > >
> > > > For these reasons, our primary comparison baselines are widely used variance-centric normalization methods (BatchNorm, LayerNorm), which share the same computational constraints and functional role as IBNorm.

---

> ### Author Response · Authors · 2025-11-20
> **Response to Reviewer mLmV (5/7)**
>
> > **Q2 (2,3): Could the authors provide a quantitative analysis or visualization showing how these three functions differ in their information compression characteristics (e.g., in terms of mutual information, entropy, or Fisher information)? Additionally, what design principles guided the specific forms of IBNorm-S/L/T? Were they chosen empirically or derived from any theoretical insight?**
>
> Thank you for your insightful questions.
>
> **IB-Inspired Normalization**
>
> As described in Section 4.3, all three IBNorm variants are derived from a unified IB-theoretic design principle:
> a valid compression operation must satisfy the bounded-compression property to guarantee the generalization benefits formalized in Theorem 1 and Corollary 2. Based on this principle, we instantiate three representative compression functions: IBNorm-S, IBNorm-L, and IBNorm-T, chosen to span different functional shapes, different tail-compression strengths, and different sensitivities to large activations, while all remaining within the same theoretical constraints.
>
> **Distinction of IBNorm-S, IBNorm-L, and IBNorm-T**
>
> IBNorm-S, IBNorm-L, and IBNorm-T are three types of IBNorm, which satisfy the compression property. We discuss their compression performance in detail in Appendix D.5. Compression operations in IBNorm compress the tail of activations while adjusting higher-order statistics. IBNorm-L, IBNorm-T, and IBNorm-S exhibit different abilities to compress the tails of activations. These differences manifest in the induced entropy of the output distribution and in how aggressively each function compresses activation tails. Because all three satisfy IB constraints, their IB curves (predictive MI, nuisance MI, token-level IB value) follow similar trends during training, making it difficult to distinguish their own characteristics purely from the IB framework. For this reason, we provide an additional quantitative analysis using kernel density entropy, which directly reflects how much information each compression function removes.
>
> **Quantitative Analysis of IBNorm-S/L/T**
>
> To illustrate the distinct compression behaviors of IBNorm-S, IBNorm-L, and IBNorm-T, we compute the entropy of the kernel density estimate (KDE) of activations after applying each variant (with the same hyperparameter $\lambda=4$ under different input distributions. For an input Gaussian $\mathcal{N}(0,2)$ with entropy 1.768, the resulting KDE entropies are 1.695 for IBNorm-S, 1.530 for IBNorm-L, and 1.302 for IBNorm-T. Similarly, for an input Gaussian $\mathcal{N}(0,1)$ with entropy 1.418, the corresponding output entropies are 1.352 for IBNorm-S, 1.263 for IBNorm-L, and 1.192 for IBNorm-T. These results consistently show that IBNorm-S performs the mildest compression, IBNorm-L exhibits stronger compression, and IBNorm-T achieves the most aggressive entropy reduction, highlighting the distinct statistical effects of the three IBNorm variants. In conclusion, all three IBNorm variants are grounded in the same IB-theoretic design principle. They differ in tail-compression strength, statistical effect, and entropy reduction.
>
> Beyond compression strength, we observe differences in robustness. As shown in Tables 4 and 11, IBNorm-S is less sensitive to variations in the hyperparameter $\lambda$ compared to IBNorm-L and IBNorm-T. Similarly, ablation studies on the normalization structure shown in Tables 5 and 10 indicate that IBNorm-S is more robust to the ordering of the IB compression and standardization operations.
>
> At the task level, performance differences align with these compression characteristics. Across diverse medium-scale LLMs (Llama 130M–1B and GPT 350M), IBNorm-S generally achieves the worst performance among the three IBNorm variants on evaluation tasks in Tables 1 and 6, reflecting its mildest compression. For challenging tasks, such as BBH and GPQA, the stronger compression applied by IBNorm-L can lead to improved performance, suggesting that mild compression can better support reasoning and generalization in difficult tasks.
>
> We include these analysis of IBNorm-S/L/T in our revision in Appendix D.5.

---

> > ### Author Response · Authors · 2025-11-20
> > **Response to Reviewer mLmV (6/7)**
> >
> > > **Q4: Impact of $\lambda$ across models and tasks. The relationship between $\lambda$ and model performance may vary across architectures and data modalities. Could the authors provide more analysis (e.g., $\lambda$-sensitivity curves or cross-task validation) to better understand how $\lambda$ affects representation informativeness and generalization?**
> >
> > Thank you for this insightful comment. We agree that the choice of $\lambda$ may depend on both the architecture and the downstream task. We provide extensive $\lambda$-sensitivity analyses in Appendix D.3.
> >
> > **Ablation Study on $\lambda$**
> >
> > We include detailed ablation studies on both LLaMA and GPT-2 architectures:
> >
> > - LLaMA-60M results are reported in Table 11.
> >
> > - GPT-2 Small (124M) results are reported in Table 12.
> >
> > Across all experiments, we find that for IBNorm-L, IBNorm-T, and IBNorm-S, values of $[3,5]$ yield consistently strong performance, with no significant fluctuations within this range. This stability across two architectures suggests that the method is not highly sensitive to the precise choice of $\lambda$.
> >
> > **$\lambda$–Information Bottleneck Curve**
> > To further analyze how $\lambda$ affects representation compression from an IB-theoretic perspective, we provide an extended $\lambda$–IB curve analysis for LLaMA-130M in Appendix D.3. We demonstrate the evolution of predictive information $\hat I(Y;T_l)$, task–nuisance information $\hat I(T_{l-1};T_l)$, and the token-level IB value during training of Llama-130M on C4, using IBNorm-T across different hyperparameters $\lambda=0.5,4,8$.
> >
> > The results (Figure 3) reveal a clear trend:
> >
> > - Under-compression ($\lambda=0.5$) fails to sufficiently suppress task-irrelevant information, leading to suboptimal IB tradeoffs.
> >
> > - Over-compression ($\lambda=8$) starts to suppress task-relevant information, harming predictive performance.
> >
> > - Moderate compression ($\lambda=4$) achieves the best IB value, effectively removing nuisance factors while preserving task-useful information.
> >
> > These findings provide a principled explanation for the empirically observed optimal $\lambda$.
> >
> > In conclusion,
> >
> > 1. IBNorm is robust to the choice of $\lambda$, with a stable optimal range across models.
> >
> > 2. The effect of $\lambda$ is theoretically interpretable through the IB framework, which explains why moderate compression is most effective.

---

> > > ### Author Response · Authors · 2025-11-20
> > > **Response to Reviewer mLmV (7/7)**
> > >
> > > > **Q5: Scalability to larger models and practical deployment. How does IBNorm scale to larger LLMs (e.g., 7B–70B parameters)? Are there computational trade-offs (e.g., increased training time, numerical stability issues) when applying IBNorm at scale? Clarifying these practical considerations would be useful for understanding IBNorm's potential in real-world.**
> > >
> > > Thank you for the question regarding scalability.
> > >
> > > **Extended Results on LLaMA 1B**
> > >
> > > We conducted additional experiments on the LLaMA-1B model to evaluate IBNorm at a larger scale. These results confirm that IBNorm consistently improves downstream performance, demonstrating its effectiveness (see Tables 1 and 6).
> > >
> > > **Computational Overhead and Stability**
> > >
> > > IBNorm introduces only a single compression operation per normalization layer. Consequently, it does not incur significant computational overhead or cause numerical stability issues, even for larger models. Training time and memory usage remain comparable to standard variance-centric normalization methods such as LayerNorm or RMSNorm.
> > >
> > > We report the training time and total VRAM usage for LLaMA-60M under the same global batch size$=512$ trained on 4×L40-S GPUs:
> > > |Normalization|Training Time|Total VRAM|
> > > |-|-|-|
> > > |LayerNorm| 1h 54min | 95944MiB|
> > > |RMSNorm| 2h 15min |102601MiB |
> > > |NormalNorm| 3h 38min |139207MiB |
> > > |IBNorm-S| 2h 14min |102460MiB |
> > > |IBNorm-L| 2h 17min | 106371MiB |
> > > |IBNorm-T| 2h 14min | 104853MiB |
> > >
> > > Our experiments are limited to medium-scale LLMs due to computational constraints. Extending evaluations to larger foundation models is a direction for our future work.
> > >
> > > > **Q6: Under the IB framework, it is difficult to determine in advance how close we are to optimal compression, which can easily lead to the over-compression or under-compression issue of representations [4,6]. Has this work considered the problems in the compression process, which may have significant implications for layer-wise IB?**
> > >
> > > Thank you for the insightful question.
> > >
> > > It is intractable to determine the theoretically optimal IB compression, as the optimal IB value cannot be derived without access to the full population distribution. Therefore, the notion of "optimal compression" is generally intractable in practice.
> > >
> > > In IBNorm, the optimization of IB value is treated as a dynamic process within the representation space during training. Rather than aiming for a fixed target IB value, IBNorm continuously adjusts layer-wise compression through the normalization layers, allowing representations to adaptively balance task-relevant and task-irrelevant information. This dynamic approach mitigates the risk of over-compression or under-compression at each layer and aligns with the overall training objective, providing practical and effective layer-wise control of IB values.
> > >
> > > [6] The conditional entropy bottleneck. Entropy 2020.

---

> > > > ### Author Response · Authors · 2025-11-27
> > > >
> > > > Dear Reviewer mLmV,
> > > >
> > > > We believe that our response has addressed your concerns. As the discussion period concludes in seven days, we kindly ask that you review our response and, if you find it has comprehensively resolved your issues, please consider revising your scores accordingly. If you have any additional questions or require clarification, please do let us know.

---

### Author Response · Authors · 2025-11-20
**Summary of Revisions**

We thank all the reviewers for their valuable feedback and insightful suggestions. Based on the reviews, we have made the following revisions to our paper:

a. We have enhanced the clarity of our problem setting and highlighted the novelty of our proposed methodology by including a detailed comparison with prior explicit information bottleneck (IB) works [1-3], and adding a discussion about prior comparison with task-irrelevant information compression methods in representation learning [4-9]. (Sections 2, 4, 6, and Appendix B) [`Reviewer mLmV, P5Gw`]

b. We have provided a justification for the unique advantages of the compression functions in IBNorm compared to standard nonlinear activation functions. (Section 4 and Appendix B.1) [`Reviewer mLmV`]

c. We clarify the distinctions among the three types of compression functions (IBNorm-S, IBNorm-L, IBNorm-T) and provide a quantitative analysis of their performance. (Appendix D.5) [`Reviewer mLmV`]

d. We have refined the statement of Corollary 2 and improved the proofs of Theorem 1 and Corollary 2 to enhance both clarity and rigor. (Corollary 2 and Appendix F) [`Reviewer 9sfx`]

e. We have included the training time and VRAM analysis. (Appendix D.7) [`Reviewer mLmV`]

f. We have added an explanation regarding the failure cases of IBNorm language tasks. (Appendix D.1) [`Reviewer P5Gw`]

g. We include additional ablation studies on the hyperparameter $\lambda$ for GPT-2, along with a sensitivity analysis. (Appendix D.3) [`Reviewer mLmV, P5Gw`]

h. We have conducted statistical significance analyses of our results and clarified all experimental settings. (Tables 8, 9, and Appendix G) [`Reviewer P5Gw, 9sfx`]

i. We provide additional experiments on Llama-1B, GPT-2, and ViT under the ViT-S/16 and ViT-B/16 settings. (Tables 1, 2, 7, and 13) [`Reviewer mLmV`]

j. We have improved the quality of visual illustrations (e.g., Figure 1), corrected minor typographical errors, and refined wording and phrasing throughout the manuscript. [`Reviewer P5Gw`]

We have also addressed each reviewer's comments with more detailed, in-depth responses. Once again we appreciate all the suggestions made by reviewers to improve our work. It is our pleasure to hear your feedback, and we look forward to answering your follow-up questions.

[1] Deep variational information bottleneck. ICLR 2017.

[2] Mutual information neural estimation. ICML 2018.

[3] Structured probabilistic coding. AAAI 2024.

[4] Iterative normalization: Beyond standardization towards efficient whitening. CVPR 2019.

[5] Decorrelated batch normalization. CVPR, 2018.

[6] On the impact of activation and normalization in obtaining isometric embeddings at initialization. NeurIPS 2023.

[7] Impartial multi-task representation learning via variance-invariant probabilistic decoding. ACL 2025.

[8] On feature decorrelation in self-supervised learning. ICCV, 2021.

[9] Modulate your spectrum in self-supervised learning. ICLR, 2024.

---

### Comment · Area_Chair_iwst · 2025-11-24

Dear reviewers,

       The authors now have given their response to the reviews, please have a look on the rebuttal and revised PDF to make your further concerns.

      After that, please give your final rating on this submission.

Your AC
Best

---

### Meta-Review · Area_Chair_HSMQ · 2025-12-19

**Summary:**

This paper introduces IBNorm, a normalization method inspired by the Information Bottleneck (IB) principle. Unlike variance-centric methods (BatchNorm, LayerNorm, RMSNorm), IBNorm incorporates compression operations before standardization to preserve task-relevant information while suppressing nuisance variability. The authors provide theoretical analysis claiming higher IB values and tighter generalization bounds, along with empirical validation across language models (LLaMA, GPT-2) and vision models (ResNet, ViT).

The submission received mixed reviews: one marginally above acceptance (6), one marginally below (4), and one clear rejection (2). The discussion period revealed substantial concerns about theoretical rigor, experimental methodology, and incremental novelty.

**Reviewer Concerns:**

**Addressed Concerns:**

* Novelty and positioning (Reviewers mLmV, P5Gw): The authors clarified that IBNorm is the first to systematically analyze normalization through an IB-theoretic lens, distinguishing it from NormalNorm and explicit IB methods (VIB, MINE) that require auxiliary MI estimators. They added comparisons with whitening methods (IterNorm) showing IBNorm's superiority.
* Compression function justification (Reviewer mLmV): Authors explained that unlike activation functions (e.g., tanh), IBNorm's compression operates within normalization layers that are invoked throughout the model, systematically shaping representations. They provided quantitative analysis via KDE entropy showing distinct compression strengths: IBNorm-S (mild), IBNorm-L (moderate), IBNorm-T (aggressive).
* Hyperparameter sensitivity (Reviewers mLmV, P5Gw): Extensive ablation studies (Tables 11-12) demonstrate λ ∈ [1.0, 2.0] yields stable performance. λ-IB curve analysis (Figure 3) shows under-compression (λ < 1) and over-compression (λ > 3) lead to suboptimal IB tradeoffs.
* Statistical significance (Reviewers P5Gw, 9sfx): Authors added 3-run experiments for LLaMA-60M/130M and GPT-2 Small (Tables 8-9), confirming consistent improvements. They committed to reporting mean ± std for all experiments.

**Remaining Concerns:**

* Theoretical rigor: Reviewer 9sfx identified multiple flaws in reasoning and statement of the proofs. Although the authors claim to have fixed it, this is a substantial change to the manuscript and should be reviewed by more expert reviewers that the AC.

**Reviewer Scores:**

* Reviewer mLmV: 4 → 5 — Reviewer might increase their score because the authors provided a comprehensive technical responses (λ-sensitivity analysis, computational overhead, IBNorm variant comparisons) directly addressed all major concerns; likely crosses acceptance threshold given "would not mind if accepted" stance.
* Reviewer P5Gw: 6 → 6 — Numerical errors corrected and statistical significance added, but incrementality concerns persist and failure case explanations remain superficial; will likely maintain marginally positive position.
* Reviewer 9sfx: 2 → 2 — Acknowledges good-faith effort and improved rigor, but core theoretical issues remain unresolved (generalization bound structure, model complexity invariance assumptions, continuous extension validity);

---

### Decision · Program_Chairs · 2026-01-26

Reject